# PRIVATE BLIND MODEL AVERAGING –
# DISTRIBUTED, NON-INTERACTIVE, AND CONVERGENT

## ABSTRACT

Scalable distributed differentially private learning would benefit notably from re-
duced communication and synchronization overhead. The current best methods,
based on gradient averaging, inherently require many synchronization rounds. In
this work, we analyze *blind model averaging* for convex and smooth empirical
risk minimization (ERM): each user first locally finishes training a model and then
submits the model for secure averaging without any client-side online synchroniza-
tion. This setting lends itself not only to data point-level privacy but also to flexible
user-level privacy, where the combined impact of the user's trained model does not
depend on the number of data points used for training.

In detail, we analyze the utility side of blind model averaging for support vector
machines (SVMs) and the inherently multi-class Softmax regression (SoftmaxReg).
On the theory side, we use strong duality to show for SVMs that blind model
averaging converges toward centralized training performance if the task is robust
against L2-regularization, i.e. if increasing the regularization weight does not
destroy utility. Furthermore, we provide theoretical and experimental evidence that
blind averaged Softmax Regression works well: we prove strong convexity of the
dual problem by proving smoothness of the primal problem. Using this result, we
also derive the first output perturbation bounds for Softmax regression. On the
experimental side, we support our theoretical SVM convergence. Furthermore, we
observe hints of an even more fine-granular connection between good utility of
model averaging and mid-range regularization weights which lead to compelling
utility-privacy-tradeoffs for SVM and Softmax regression on 3 datasets (CIFAR-
10, CIFAR-100, and federated EMNIST embeddings). We additionally provide
ablation for an artificially extreme non-IID scenario.

## 1 INTRODUCTION

Non-interactivity in distributed learning, with just one user message sent, offers minimal commu-
nication overhead for privacy-preserving learning, which enables high scalability. Yet, Smith et al.
(2017) have shown that non-interactive learning has limitations: for any non-interactive learning
method, there are problems for which the excess risk increases exponentially with the number of
model parameters, where excess risk is the loss-difference to the optimal model for the union of all
local data. By contrast, for a class of convex empirical risk-minimization (ERM)-based learning
objectives, Jayaraman et al. (2018) have explored *blind model averaging (BlindAvg)*: each user first
locally finishes training a model and then submits the model for secure averaging non-interactively.
Their work, however, does not capture the key utility improvement of BlindAvg. Their result leaves
open whether blindly averaged ERMs converges beyond the best local model. More importantly, prior
work leaves the following question open, which we tackle in this work: *Under which circumstances
does BlindAvg achieve a compelling utility-privacy tradeoff comparable to the centralized setting?*

To achieve privacy amplification for blind averaging, prior work has identified a necessary condition:
a bound of the trained model $s$ (output sensitivity). In contrast to sensitivity bounds only for global
optima (Chaudhuri et al., 2011; Jayaraman et al., 2018), the work by Wu et al. (2017) accounts for
leakage of the optimization algorithm and provides such bounds for each training iteration. They
show that for output sensitivity with stochastic gradient descent (SGD) learning it suffices that we
have constants $\beta$, $\Lambda$, and $L$ s.t. the objective function has a lower and upper bound on the second

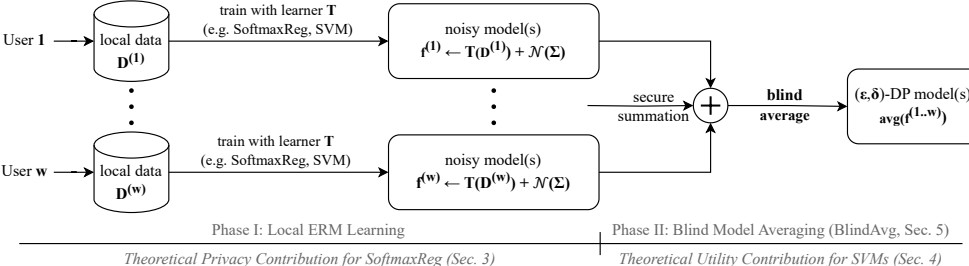

Figure 1: **Schematic overview of BlindAvg** (details: Sec. 5). (Phase I) Each user locally learns an ERM model, e.g. our SoftmaxReg (Alg. 1) or an SVM (Alg. 3), and noises it once with $\mathcal{N}(\Sigma)$. (Phase II) A single SecSum step results in an averaged $(\varepsilon, \delta)$-DP model. For hinge-loss SVMs, this blind average converges in the limit (cf. Sec. 4). Our experiments (cf. Sec. 6) use SimCLR pretraining.

derivative ($\Lambda$-strong convexity and $\beta$-smoothness respectively) and an upper bound on the first derivative ($L$-Lipschitzness). Then $s \in \mathcal{O}(L/\Lambda n)$ for $n$ many data points.

**Contributions.** We examine blind model averaging (BlindAvg) as visualized in Fig. 1. Training (Phase I) is non-interactive and local. Averaging (Phase II) is "blind" in the sense that no synchronization is needed and is private for any learning method with an output sensitivity bound.

**(1) Phase II: Robustness towards regularization drives utility and convergence of BlindAvg.** We show that for SVMs blind model averaging converges toward centralized training performance if the task is robust against L2-regularization, i.e. if increasing the regularization weight does not destroy utility. In our experiments (cf. Fig. 5), we support our theoretical SVM convergence, observe the same for blind averaged Softmax Regression, and observe hints of an even more fine-granular connection between good utility of model averaging and mid-range regularization weights.

**(2) Phase I: Blind averaging Softmax regression (SoftmaxReg) works and is output private.** A key property for proving convergence of BlindAvg for SVMs is that the dual objective is well-behaved in the sense of strong duality. We take a first step toward understanding convergence of BlindAvg for SoftmaxReg: we prove strong convexity of the dual objective by proving smoothness of the primal problem (cf. Thm. J.8). Smoothness, Lipschitzness (cf. Thm. J.7), and strong convexity (cf. Thm. J.6) conclude output perturbation bounds for SoftmaxReg Wu et al. (2017). More technically, the sensitivity of SoftmaxReg $s = 2L/\Lambda n$ (cf. Thm. 3.1) depends on its Lipschitzness $L$ which is, in contrast to SVMs, independent of the number of classes for a fixed model size.

**(3) Compelling utility-privacy-tradeoff for SVM and SoftmaxReg on 3 datasets (cf. Fig. 2).** We use CIFAR-10, CIFAR-100, and federated EMNIST, all after SimCLR pretaining Chen et al. (2020b). Supported by theoretical arguments, we observe that blindly averaged SoftmaxReg outperforms blindly averaged SVMs (cf. Sec. 3) as the number of classes increases and blindly averaged SVM outperforms gradient averaging such as FL, as the number of users increases (cf. Sec. 7).

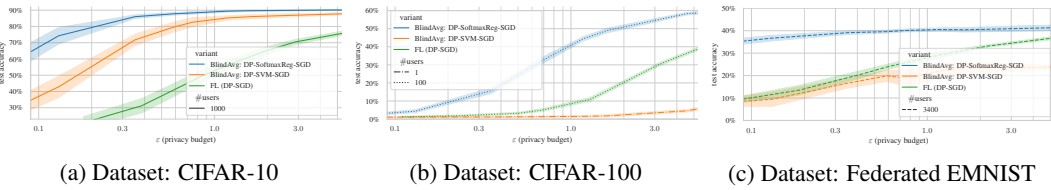

(a) Dataset: CIFAR-10          (b) Dataset: CIFAR-100          (c) Dataset: Federated EMNIST

Figure 2: **Main result** (detailed plot: Fig. 7). Classification accuracy vs. $\varepsilon$ of BlindAvg ($\delta = 10^{-5}$, $t = 50\%$ honest users) on SoftmaxReg (Alg. 1) and SVM (Alg. 3) and DP-SGD-based federated learning (FL). Only ours is non-interactive. The line of SVM-SGD for CIFAR-100 represents an upper bound for the 100 user performance. For CIFAR10/100, each user gets a random data split.

**(4) Generalizability & Ablation.** We design two heterogeneous datasets: (a) BlindAvg succeeds in an extreme non-IID scenario where each user solely holds data from one class and (b) BlindAvg fails if the data only classify well for a too-small regularization. For real-world heterogeneous data, the accuracy of the SVM variant only deteriorates by 2 percentage points on CIFAR-10 in the non-IID

scenario (a) (cf. Tbl. 2) whereas for federated EMNIST with differing local data sizes, the gap becomes larger (cf. Fig. 7 (d)). Moreover, our framework is scalable to millions of users if a scalable secure summation protocol is used, e.g. Bell et al. (2020). For user-level privacy, we extrapolate compelling utility-privacy results for 20 million users, e.g. $87\%$ accuracy for $\varepsilon = 10^{-4}$ (cf. Fig. 6).

## 2 PRELIMINARIES & NOTATION

Table 1: (left) A **configuration** $\zeta(\mathcal{U}, t, \mho, T, K, \sigma)$, short: $\zeta$, and (right) other **notation**.

| Symbol | Description | Symbol | Description |
|---|---|---|---|
| $U^{(i)} \in \mathcal{U}$ | $i$-th out of $w$-many users | $\mathrm{avg}(T)$ | blind model average: $\frac{1}{w}\sum_{i=1}^{w} T_\xi(D^{(i)})$ |
| $t$ | ratio of honest users | $V \subseteq D$ | set of support vectors |
| $D^{(i)} \in \mho$ | local dataset of the $i$-th user | $K_{\mathrm{comp}}$ | number of compositions |
| $n^{(i)}$ | number of local data points | $(x, y)$ | label $y$ and data point $x \in \mathbb{R}^p$ in $D$ |
| $\xi$ | hyperparameters of $T$ | $f$ or $\alpha$ | model parameter; dual: $f = \sum_{j=1}^{n} \alpha_j x_j$ |
| $T_\xi(D)$ | $s$-sensitivity bounded learner e.g. SVM-SGD, SoftmaxReg | $\mathcal{J}$ | objective function on inputs $(f, D)$ |
| | | $M$ | number of training iterations |
| $K$ | number of classes | $L, \beta, \Lambda$ | $L$-Lipschitz, $\beta$-smooth, $\Lambda$-regularized |
| $\sigma \in \mathbb{R}_+$ | noise scale | $c, R$ | input & model clipping bound |

Preliminaries of secure summation, pretraining, dual SVM representation via the representer theorem, and strong convexity, Lipschitzness, and smoothness definitions are available in Appxs. D.2 to D.5.

**Differential Privacy (DP)** (Dwork et al., 2006b) quantifies the protection of any individual's data within a dataset against an arbitrarily strong attacker observing the output of a computation on the said dataset. Strong protection is achieved by bounding the influence of each individual's data on the resulting ML model. For the (standard) definition of DP utilized in our proofs, we refer to Appx. D.1.

**DP ERMs via SGD Training** We consider machine learning algorithms (learners) that have a strongly convex objective function, like Support Vector Machines (SVMs), logistic regression (LR), or a softmax-activated single-layer perceptron (SoftmaxReg). These algorithms display a unique local minimum and a lower bound on the objective function's growth, making them ideal for computing tight DP bounds. If we train these algorithms with stochastic gradient descent (SGD) for a bounded learning rate and their objective function is smooth and Lipschitz, then they are uniformly stable (Hardt et al., 2016) which implies a DP bound on the learner (Wu et al., 2017). For that, the sensitivity of the model is bounded in any iteration and does not scale with the number of iterations if we choose a linearly decreasing learning rate schedule.

**Theorem 2.1** (Lemma 8 in Wu et al. (2017)). *Let $\zeta$ be a configuration as in Tbl. 1. If a learner $T_\xi$ is trained with SGD on learning rate $\tau_m = \min(^1/\beta, ^1/\Lambda m)$ for iteration $m$ and has a $\Lambda$-strongly convex, $\beta$-smooth, and $L$-Lipschitz objective function $\mathcal{J}$, then the output model $f = T_\xi(D)$ has the sensitivity bound $s = ^{2L}/\Lambda n$. A mini-batch SGD extension follows by Wu et al. (2017).*

For a finite data-independent Lipschitzness $L$, we use a $c$-bounded input space $\mathcal{X}$ ensured via norm-clipping, i.e. $\forall x \in \mathcal{X}\colon \|x\| \leq c$, as well as an $R$-bounded model parameter space $\mathcal{F}$ ensured via projected SGD, i.e. rescaling $f \in \mathcal{F}$ after each iteration $m$ s.t. $\|f_m\| \leq R$. Thus, the parameters $c$ and $R$ influence the Lipschitzness $L$ and the sensitivity for most objective functions.

Adding Gaussian noise to the output of a sensitivity-bounded function archives DP if the noise is calibrated to the sensitivity. We assume that for each class, one model is trained and noised. For a binary-class model, set $K = 1$.

**Definition 2.2** (Delta-Gaussian). Given noise scale $\tilde{\sigma} \in \mathbb{R}_+$, complementary error function erfc, $\sigma_{\mathrm{p}} := ^1/\tilde{\sigma}$, and $\mu_{\mathrm{p}} := \sigma_{\mathrm{p}}^2/2$, we define $\delta(\varepsilon, K_{\mathrm{comp}}) = 0.5 \cdot (\mathrm{erfc}(\frac{\varepsilon - K_{\mathrm{comp}}\mu_{\mathrm{p}}}{\sqrt{2K_{\mathrm{comp}}}\sigma_{\mathrm{p}}}) - e^\varepsilon \mathrm{erfc}(\frac{\varepsilon + K_{\mathrm{comp}}\mu_{\mathrm{p}}}{\sqrt{2K_{\mathrm{comp}}}\sigma_{\mathrm{p}}}))$.

**Lemma 2.3** (Gaussian mechanism is DP, Theorem 5 in Sommer et al. (2019) & Lem. J.5). *Let $q_k$ be $s$-sensitivity-bounded functions on data $D$ and $\mathcal{N}$ a multivariate Gaussian. The Gaussian mechanism $D \mapsto \{q_k(D) + \mathcal{N}(0, \tilde{\sigma}^2 s^2 I)\}_{k \in \{1, \dots, K_{comp}\}}$ is tightly $(\varepsilon, \delta)$-DP with $\delta(\varepsilon, K_{comp})$ as in Def. 2.2.*

**Corollary 2.4** (Gaussian mechanism on $T_\xi$ is DP). *For configuration $\zeta$ as in Tbl. 1 and learner $T_\xi$ of Thm. 2.1 with an output model $f \in \mathbb{R}^{p+1}$, $D \mapsto T_\xi(D) + \mathcal{N}(0, \tilde{\sigma}^2 s^2 I_{(p+1) \times K})$ is tightly*

$(\varepsilon, \delta)$-*DP with $\delta(\varepsilon, K_{comp})$ as in Def. 2.2. $K_{comp} = K$ when $s$ is defined on a per-class learner like in Example 2.5/2.6 (SVM/LR) and $K_{comp} = 1$ on the all-classes-included learner in Alg. 1 (SoftmaxReg).*

In essence, $\varepsilon \in \mathcal{O}(s \cdot \sqrt{K_{\text{comp}}})$. Balle et al. (2020a) have shown a similar tight composition result. By Thm. J.3 in Appx. J.3.1 we can apply the Gaussian mechanism for a deterministic sensitivity (cf. Lem. 2.3) to learner $T_\xi$ of Thm. 2.1 that has a randomized sensitivity as in Def. D.3.

*Example* 2.5 ($T = $ SVM-SGD). Alg. 3 describes SVM training in the one-vs-rest (OVR) scheme where we train an SVM for each class against all other classes. The objective function is $\Lambda$-strongly convex, $(\Lambda R + c)$-Lipschitz, and $((c^2/2h + \Lambda)^2 + p\Lambda^2)^{1/2}$-smooth if we use a Huber loss (cf. Appx. I.1) which is a smoothed hinge-loss. Thus, $T$ has a per-class sensitivity of $s = 2(\Lambda R + c)/\Lambda n$.

*Example* 2.6 ($T = $ LR-SGD). For logistic regression (LR), we adapt Alg. 3 with the objective function $\mathcal{J}'(f, D) \coloneqq \frac{\Lambda}{2}\langle f, f\rangle + \frac{1}{n}\sum_{(x,y)\in D} \ln(1 + \exp(-y\langle f, clipped(x)\rangle))$ and $\beta = ((c^2/4 + \Lambda)^2 + p\Lambda^2)^{1/2}$.

# 3 PHASE I: DIFFERENTIALLY PRIVATE SOFTMAXREG

---

**Algorithm 1** Our $T_\xi = \texttt{SoftmaxReg-SGD}_\xi(D)$ with hyperparameters $\xi \coloneqq (K, c, M, \Lambda, R)$

---

**Input:** dataset $D \coloneqq \{(x_j, y_j)\}_{j=1}^n$ where data point $x_j$ is structured as $[1, x_{j,1}, \ldots, x_{j,p}]$; #classes $K$; input clipping bound: $c \in \mathbb{R}_+$; #iterations $M$; regularization parameter: $\Lambda \in \mathbb{R}_+$; model clipping bound: $R \in \mathbb{R}_+$

**Result:** $f_M \in \mathbb{R}^{(p+1)\times K}$: a model with hyperplane $\in \mathbb{R}^{p\times K}$ and intercept $\in \mathbb{R}^K$

$clipped(x) \coloneqq c \cdot x / \max(c, \|x\|)$

$\mathcal{J}_{\text{softmax}}(f, D) \coloneqq \frac{\Lambda}{2}\sum_{k=1}^K \langle f_k, f_k\rangle + \frac{1}{n}\sum_{(clipped(x),y)\in D} - \sum_{k=1}^K y_k \log \frac{\exp\langle f_k, x\rangle}{\sum_{j=1}^K \exp\langle f_j, x\rangle}$

**for** $m$ **in** $1, \ldots, M$ **do**

    $f_m \leftarrow f_{m-1} - \tau_m \nabla \mathcal{J}_{\text{softmax}}(f_{m-1}, (x_j, y_j), k)$, with learning rate $\tau_m \coloneqq \min(\frac{1}{\beta}, \frac{1}{\Lambda m})$,

    $\beta = \sqrt{(d+1)K\Lambda^2 + 0.5(\Lambda + c^2)^2}$, and index $j = m \bmod n$.

    $f_m \coloneqq R \cdot f_m / \|f_m\|$                                            {projected SGD}

---

For Phase I (cf. Fig. 1), we show that Softmax regression satisfies an output sensitivity bound. The differentially private variant of the Huber-loss SVM (SVM-SGD) (Wu et al., 2017) in Alg. 3 falls short for multi-class classification. We address this shortcoming by showing DP for a softmax-activated single-layer perceptron (SoftmaxReg-SGD) in Alg. 1. It has (1) a utility and (2) a privacy boost over SVM-SGD: (1) Since SoftmaxReg-SGD uses the Softmax loss, we also optimize the selection of the most dominant class. In contrast, SVM-SGD uses the one-vs-rest (OVR) scheme where each of the $K$ classes is trained against each other, resulting in $K$ many SVMs. Then, the most dominant class is selected by the argmax of the prediction of all SVMs which is static and thus not optimized. (2) The privacy guarantee $\varepsilon_{\text{softmax}}$ of SoftmaxReg-SGD does not scale with the number of classes $K$ if we keep the hypersphere of trainable parameters $R$ constant. In contrast, $\varepsilon_{\text{svm}}$ of SVM-SGD scales with $\sqrt{K}$ in the same scenario since it does require a $K$-fold sequential composition. Overall, when scaling with the number of trainable parameters, SoftmaxReg-SGD saves a privacy budget of $\frac{\varepsilon_{\text{svm}}}{\varepsilon_{\text{softmax}}} \propto \frac{\Lambda R + \sqrt{K}c}{\Lambda R + \sqrt{2}c}$ with input clipping bound $c$ and regularization parameter $\Lambda$.

We show DP for SoftmaxReg-SGD by showing that its objective $\mathcal{J}_{\text{softmax}}$ is $\Lambda$-strongly convex (cf. Thm. J.6), $(\beta = \sqrt{(p+1)K\Lambda^2 + 0.5(\Lambda + c^2)^2})$-smooth (cf. Thm. J.8), and $(L = \Lambda R + \sqrt{2}c)$-Lipschitz (cf. Thm. J.7). Then, a sensitivity directly follows from Wu et al. (2017, Lemma 8).

**Theorem 3.1** (SoftmaxReg-SGD sensitivity). *For configuration $\zeta$ as in Tbl. 1, the learning algorithm $T = $ SoftmaxReg-SGD of Alg. 1 has a sensitivity of $s = 2(\Lambda R + \sqrt{2}c)/\Lambda n$ for the output model.*

**Corollary 3.2** (Gaussian mechanism on SoftmaxReg-SGD is DP). *For configuration $\zeta$ as in Tbl. 1 and $s$-sensitivity-bounded $T = $ SoftmaxReg-SGD with an output model $f \in \mathbb{R}^{(p+1)\times K}$ (cf. Thm. 3.1), DP-SoftmaxReg-SGD$_\xi(D, \tilde{\sigma}) \coloneqq$ SoftmaxReg-SGD$_\xi(D) + \mathcal{N}(0, \tilde{\sigma}^2 s^2 I_{(p+1)\times K})$ is tightly $(\varepsilon, \delta)$-DP with $\delta(\varepsilon, K_{comp} = 1)$ as in Def. 2.2.*

Cor. 3.2 directly follows from a bounded sensitivity (cf. Thm. 3.1) and the Gaussian mechanism (cf. Cor. 2.4). Note that, $\varepsilon$ is in $\mathcal{O}(s)$ as $s$ bounds the sensitivity of all classes thus we do not need to compose any per-class models sequentially.

**Lipschitzness.** The sensitivity of SoftmaxReg-SGD and thus the privacy budget $\varepsilon$ is directly proportional to its Lipschitzness $L = \Lambda R + \sqrt{2}c$. The constant $L$ is proven in Das et al. (2023, Appendix D) for a non-regularized objective and we prove it in Appx. J.4.2 for regularized one which shows to be independent of the number of classes for a fixed $R$ ($\forall f\colon \langle f, f \rangle \leq R$). The proof bounds the Jacobian of the objective $\mathcal{J}_{\text{softmax}}$, i.e. $\sup_{z \in D, f} \|\nabla_f \mathcal{J}_{\text{softmax}}(f, z)\| \leq L$. The "$\Lambda R$"-part of the bound $L$ originates from the l2-regularization term in the objective, i.e. $\Lambda/2 \langle f, f \rangle$, which is influenced by the size of model $f$ whereas the "$\sqrt{2}c$"-part originates from the softmax loss for which we use the characteristic of the softmax function that the probabilities of each class add up to 1. In particular, we use the fact that for the $K$ softmax probabilities $s_1, \ldots, s_K$ and class label $y$: $\max_{s_1, \ldots, s_K}\big\{\big(\sum_{k=1}^{K}(s_k - 1_{[y=k]})^2\big)^{1/2} \mid \sum_{k=1}^{K} s_k = 1 \wedge \forall k\colon s_k \geq 0\big\} = \sqrt{2}$. $L$ contains this $\sqrt{2}$.

**Smoothness.** By (Zhou, 2018, Theorem 1) a smoothness and convexity of the primal problem implies a strongly convex dual problem. The smoothness is also used as the upper bound of the learning rate in SoftmaxReg-SGD. We prove in Appx. J.4.3 smoothness $\beta = \sqrt{(p+1)K\Lambda^2 + 0.5(\Lambda + c^2)^2}$ by bounding the Hessian of the objective: $\sup_{z \in D, f}\|\mathbf{H}_f(\mathcal{J}_{\text{softmax}}(f, z))\| \leq \beta$. The first part of the smoothness bound $\beta$, "$(p+1)K\Lambda^2$", stems from the l2-regularization term of the objective function $\Lambda/2\langle f, f \rangle$ which is influenced by the size of model $f \in \mathbb{R}^{(p+1) \times K}$. In particular, the second derivative of the regularization term is constant in each direction of the derivative thus marking the dependence on the number of model parameters $(p+1)K$. The second part of $\beta$, "$0.5(\Lambda + c^2)^2$", stems from the softmax loss function for which we use the characteristic of the softmax function that the probabilities of each class add up to 1. In particular, we also use the fact that $\max_{s_1, \ldots, s_K}\big\{\sum_{k=1}^{K} s_k(1 - s_k)(C + s_k) \mid \sum_{k=1}^{K} s_p = 1 \wedge \forall k\colon s_k \geq 0\big\} \leq 0.25(C+1)^2$ for $C \propto \Lambda/c^2$ which we proof in Lem. J.9 using the KKT conditions. The bound "$0.25(C+1)^2$" scales proportional to $2c^4$ (cf. Thm. J.8) which directly corresponds to the "$0.5(\Lambda + c^2)^2$" term in $\beta$.

**Strong convexity.** The strong convexity $\Lambda$ stems from the regularization term $\Lambda/2 \langle f, f \rangle$ and we use and show in Appx. J.4.1 that the objective function without the regularization is convex. In particular, the Hessian of the softmax-based cross-entropy loss function $\mathcal{L}_{\text{CE}}(y, z) := -\sum_{k=1}^{K} y_k \log \frac{\exp z_k}{\sum_{j=1}^{K} \exp z_j}$ is convex if it is positive semi-definite: $\nabla^2 \mathcal{L}_{\text{CE}} \succeq 0$.

## 4 PHASE II: NON-INTERACTIVE BLIND MODEL AVERAGING (BLINDAVG)

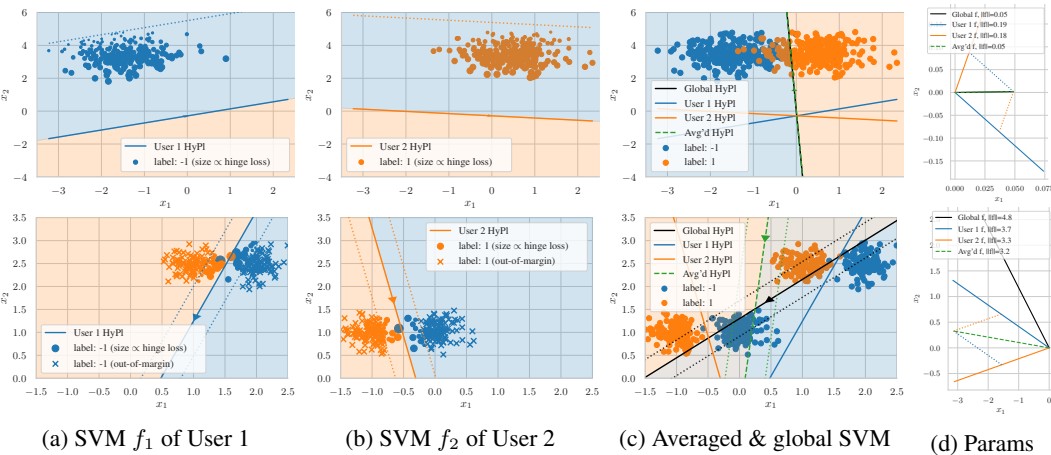

(a) SVM $f_1$ of User 1    (b) SVM $f_2$ of User 2    (c) Averaged & global SVM    (d) Params

Figure 3: **Putting blind averaging to the extreme:** (top: `SynNonIID`) an strongly biased non-IID scenario where BlindAvg works and (bottom: `SynFail`) a deliberately chosen scenario where it fails. Both scenarios differ in the strength of regularization: (`SynNonIID`) strong $\Lambda = 20$ vs. (`SynFail`) weak $\Lambda = 0.05$. (a+b) Local SVM hyperplanes (solid), their direction (arrows), and their margins (dotted) on a point cloud $x$ for users 1 and 2. (c) After averaging (green), they approximate the global SVM (black) trained on the combined point cloud, i.e. both SVMs (`SynNonIID`) overlap or (`SynFail`) fail to overlap. (d) The parameters $f$ (normal vectors) of each SVM illustrate the average $\text{avg}(f) = 0.5 \cdot (f_1 + f_2)$. Hyperparameters: R = 5, c = 5, $n^{(1)} = n^{(2)} = 250$, bs = 25, epochs = 1500.

In Phase II (cf. Fig. 1), the core idea of BlindAvg is to locally train models $f^{(i)} := T_\xi(D^{(i)})$ and then average them blind, i.e. without further synchronizing or model fine-tuning: we output $\mathrm{avg}(T) = \frac{1}{w} \sum_{i=1}^{w} f^{(i)}$. We identify robustness towards regularization as the utility driver and provide a utility bound: we (1) reduce the utility requirement of BlindAvg on a regularized ERM $T$ to the coefficients $\alpha$ of the dual problem of $T$, (2) leverage the dual problem to show that there exists a regularization parameter $\Lambda$ for a hinge-loss linear SVM, $T = \mathrm{HingeSVM}$, such that $\mathrm{avg}(T)$ gracefully convergences to the best model for the combined local datasets $\mho$: $\mathbb{E}[\mathcal{J}(\mathrm{avg}(\mathrm{HingeSVM}), \mho) - \inf_f \mathcal{J}(f, \mho)] \in \mathcal{O}(1/M)$ for $M$ many local train rounds and objective function $\mathcal{J}$. Thus, convergence holds in a strongly non-IID setting for large regularization ($\Lambda = 20$) as illustrated in Fig. 3 (top) where each user only has access to one class. We also designed a heterogeneous dataset in Fig. 3 (bottom) where BlindAvg fails to converge as the data only classifies well for a too-small regularization ($\Lambda = 0.05$).

Cor. 4.1 shows that we can describe the effect of blind averaging by analyzing the dual coefficients $\alpha_j$ of a model $f$. By the representer theorem (cf. Thm. D.7 (Argyriou et al., 2009)), ERMs like SVMs and SoftmaxReg admit the dual form $f = \sum_{j=1}^{n} \alpha_j x_j$. Our Cor. 4.1 (proof: Appx. J.2.1) shows that the average of ERMs has the union of the local dual coefficients as dual coefficients. Thus, if only a few $\alpha_j$'s differ between the averaged local models and the global model, in the worst case the error of BlindAvg is significantly smaller as if a lot $\alpha_j$'s differ. If all $\alpha_j$ are the same, BlindAvg converges.

**Corollary 4.1** (Averaged Representer theorem). *For a configuration $\zeta$ as in Tbl. 1, if a local learner $T_\xi$ admits a solution of the form $f^{(i)} = T_\xi(D^{(i)}) = \sum_{j=1}^{n} \alpha_j^{(i)} x_j^{(i)}$ (cf. Thm. D.7) then the average $\mathrm{avg}(f^{(i)})$ admits a solution of the form $f = T_\xi(\mho) = \frac{1}{w} \sum_{i=1}^{w} \sum_{j=1}^{n} \alpha_j^{(i)} x_j^{(i)}$.*

For hinge-loss SVMs, Cor. 4.1 implies Lem. 4.2 (proof: Appx. J.2.2): if each converged local HingeSVM with the same local data sizes, i.e. $\forall i, i': n^{(i)} = n^{(i')}$, has support vectors $V^{(i)}$ then its average has support vectors $V = \bigcup_{i=1}^{w} V^{(i)}$. This follows as support vectors can be defined with $\alpha_j$.

**Lemma 4.2** (Support Vectors of averaged SVMs). *For configuration $\zeta$ as in Tbl. 1, a locally trained hinge-loss linear SVM $T$, $f^{(i)} := T_\xi(D^{(i)}) = \mathrm{argmin}_f \frac{1}{n} \sum_{(x,y) \in D^{(i)}} \max(0, 1 - y \langle f, x \rangle) + \Lambda \langle f, f \rangle$, has the support vectors $V^{(i)} := \{(x, y) \in D^{(i)} \mid y \langle f^{(i)}, x \rangle \leq \langle f^{(i)}, f^{(i)} \rangle^{-1}\}$. Then, the average of these locally trained models $\mathrm{avg}(T) = \frac{1}{w} \sum_{i=1}^{w} f^{(i)}$ has the support vectors $V = \bigcup_{i=1}^{w} V^{(i)}$.*

If the support vectors and thus all $\alpha_j$ of an averaged and a global SVM are the same, we converge (cf. Thm. 4.3, proof: Appx. J.2.3). Such a scenario occurs e.g. if the regularization $\Lambda$ is high and thus the margin is large enough such that all data points are within the margin, i.e. support vectors: $V = \mho$.

**Theorem 4.3** (Averaging locally trained SVMs converges to a global SVM). *Given a configuration $\zeta$ as in Tbl. 1 and the same local data sizes $\forall i, i': n^{(i)} = n^{(i')}$, there exists a regularization parameter $\Lambda$ such that the average of locally trained models $\mathrm{avg}(T)$ with a hinge-loss linear SVM as an objective function $\mathcal{J}$ trained with projected subgradient descent using weighted averaging, $T = \mathrm{HingeSVM}\text{-}SGDWA$, converges with the number of local iterations $M$ to the best model for the combined local datasets $\mho$, i.e. $\mathbb{E}[\mathcal{J}(\mathrm{avg}(\mathrm{HingeSVM}\text{-}SGDWA), \mho) - \inf_f \mathcal{J}(f, \mho)] \in \mathcal{O}(1/M)$.*

With differential privacy, BlindAvg does not converge. The error introduced by additive Gaussian noise $\mathcal{N}(0, \tilde{\sigma})$ on a trained model $\tilde{f}$ amounts to $\mathbb{E}[|(\tilde{f} + \mathcal{N}(0, \tilde{\sigma})) - f|] = \mathbb{E}[|\mathcal{N}(0, \tilde{\sigma})|] = \tilde{\sigma}\sqrt{2}/\sqrt{\pi}$.

## 5 PHASE II: SYSTEM DESIGN OF BLINDAVG

We present the system design of BlindAvg (cf. Alg. 2 and schematically in Fig. 1) including its privacy guarantees. Each user holds a small dataset while all users jointly learn a model. There are two scenarios: first (DP, see Fig. 7), each person contributes one data point to a user who is a local aggregator, e.g. a hospital; second (user-level privacy, see Fig. 6 for $\Upsilon = n^{(i)} = 50$), each user is a person and contributes its dataset. $\Upsilon$ notates the group size in group-DP.

Consider a set of users $\mathcal{U}$, each with a local dataset $D$ of size $n^{(i)} = |D^{(i)}|$ that already is in a sufficiently simplified representation by the SimCLR pretaining feature extractor (Chen et al., 2020a;b). The users collectively train an $(\varepsilon, \delta)$-DP model using a learner $T_\xi$ that is $s$-sensitivity-bounded as in Def. D.3. An example for $T$ is SoftmaxReg-SGD (cf. Alg. 1) with $s = 2(\Lambda R + \sqrt{2}c)/\Lambda n^{(i)}$ or SVM-SGD (cf. Alg. 3) with $s = 2(\Lambda R + c)/\Lambda n^{(i)}$.

**Algorithm 2** BlindAvg($\zeta$). SoftmaxReg-SGD (Alg. 1) and SVM-SGD (Alg. 3) have $s = \frac{2L}{\Lambda n^{(i)}}$.

---

    **Function** Client BlindAvg(D, $w$, t, $\sigma$, T, $\xi$)
        **Input:** local dataset $D^{(i)}$ with $n^{(i)} = |D^{(i)}|$;
            #users $w$; ratio $t$ of honest users; noise scale $\sigma$;
            learner $T$; hyperparameters $\xi$ incl. #classes $K$
        **Result:** $f_{\text{priv}} \in \mathbb{R}^{(p+1) \times K}$: DP-models
        $f_{\text{np}} \leftarrow T_\xi(D)$         {$T$ is $s$-sensitivity-bounded}
        $f_{\text{priv}} \leftarrow f_{\text{np}} + \mathcal{N}(0, \tilde{\sigma}^2 s^2 I_{(p+1) \times K})$
            with $\tilde{\sigma} := \sigma \cdot \frac{1}{\sqrt{t \cdot w}}$
        Run client code of $\pi_{SecSum}$ on input $\frac{n^{(i)}}{w} \cdot f_{\text{priv}}$
    **Function** Server BlindAvg($\mathcal{U}$)
        **Input:** users $\mathcal{U}$;
        **Result:** empty string
        Run the server protocol of $\pi_{SecSum}$ as in Def. D.5.

---

Alg. 2 follows the scheme of Jayaraman et al. (2018) with an extension to handle differing local data sizes $n^{(i)}$: First, each user separately trains a non-private model $f_{\text{np}}$, using $T$ and the hyperparameters $\xi$, e.g. $\xi := (K, c, M, \Lambda, R)$ for SoftmaxReg-SGD. Next, each user adds to $f_{\text{np}}$, Gaussian noise scaled with $s$ and $1/\sqrt{t \cdot w}$, where $t \cdot w$ is the number of honest users in the system. Together, the users then run a secure summation protocol $\pi_{SecSum}$ as in Def. D.5 where the input of each user is the noisy model, which is scaled down by the number of users to yield the average model and scaled up by the local data size $n^{(i)}$. This upscaling keeps all local sensitivities constant and independent of $n^{(i)}$ and thus allows differentially private blind averaging with differing $n^{(i)}$. Utility-wise, the prediction of SVMs and SoftmaxReg are scale-invariant, i.e. the prediction is the same if we scale the model by any constant. If all $n^{(i)}$ are the same, this upscaling corresponds to an averaged model scaled by a constant $n$ which has no utility implications due to this scale-invariant nature. Thanks to secure summation, we show centralized-DP guarantees with noise in the order of $\mathcal{O}(w^{-1})$ after upscaling by $n$ within a threat model akin to that of federated learning with DP. For privacy accounting, we use tight composition bounds (Meiser & Mohammadi, 2018; Sommer et al., 2019; Balle et al., 2020a).

**Security Guarantee.** We provide a detailed security description in Appx. E. We prove DP for BlindAvg (Alg. 2) where $\varepsilon \in \mathcal{O}(s'/\sqrt{t \cdot w} \cdot \sqrt{K_{\text{comp}}})$ with a full statement and proof in Appx. J.1.3. Simplified, the proof follows by applying the sensitivity (cf. Lem. E.1) to the Gaussian mechanism (cf. Lem. 2.3) where the noise is applied per user (cf. Lem. E.2).

**Theorem 5.1** (Main Theorem, simplified). *For a configuration $\zeta$ as in Tbl. 1, BlindAvg($\zeta$) of Alg. 2 satisfies computational $(\varepsilon, \delta + \nu)$-DP with $\delta(\varepsilon, K_{comp})$ as in Def. 2.2 and a function $\nu$ negligible in the security parameter used in $\pi_{SecSum}$.*

BlindAvg uses local aggregators but can also protect all data points from a user with our user-level privacy generalization in Cor. 5.2 based on Cor. E.3: a $\Upsilon$-group-private variant of Thm. 5.1 where the neighboring datasets differ not by 1 but $\Upsilon$-many elements. For user-level privacy ($\Upsilon = n^{(i)}$), it suffices that the norm of each model is bounded by $R$: then an averaged model has a sensitivity of $\frac{2R}{w}$. This method enables non-SGD optimizers, varying local data sizes $n^{(i)}$, and can bound the sensitivity tighter than SVM-SGD or SoftmaxReg for certain $\Upsilon$ or $n^{(i)}$. The proof is in Appx. J.1.4.

**Corollary 5.2** (User-level sensitivity). *Given a configuration $\zeta$ as in Tbl. 1 and a learner $T_\xi$, we say that $T_\xi$ is $R$-norm bounded if for any local dataset $D^{(i)}$: $\left\| T_\xi(D^{(i)}) \right\| \leq R$. Any $R$-norm bounded learner $T_\xi$ has a deterministic sensitivity $s = 2R$. In particular, $D^{(i)} \mapsto T_\xi(D^{(i)}) + \mathcal{N}(0, \tilde{\sigma}^2 s^2 I_{p \times K})$ satisfies $(\Upsilon \varepsilon, \delta)$, $\Upsilon$-group differential privacy with $\delta(\varepsilon, K_{comp})$ as in Def. 2.2 and $\Upsilon = n^{(i)} = |D^{(i)}|$.*

**Threat model & security goals.** We assume that a fraction of at least $t$ users are honest (say $t = 50\%$), i.e. they follow the protocol including honestly generated noise and don't collude with the adversary. In contrast, untrustworthy users can collude with a passive, collaborating adversary by exchanging information about the randomness used in their local computation. The adversary is assumed to have full knowledge about each user's dataset, except for one data point of one user. Our privacy goals are $(\varepsilon, \delta)$-differential privacy (protecting single samples) and $(\varepsilon, \delta)$-$\Upsilon$-group DP (protecting all samples of a user at once) respectively, depending on whether each user is a local aggregator or a person. To compensate for untrustworthy users, we adjust the noise added by each user by $t$; e.g., if $t = 50\%$, then we double the noise to satisfy our guarantees.

**Non-interactive protocol.** BlindAvg is agnostic to a specific secure summation protocol $\pi_{\text{SecSum}}$ (cf. Alg. 2): e.g. we can use a non-interactive extension of the SecAgg protocol (Bell et al., 2020). SecAgg requires 4 communication rounds, but the user only shares their local model in the third

round. Following Bogetoft et al. (2009), we introduce $J$ computation servers that aggregate the model parameters on behalf of the users. Specifically, each user $i$ sends their model $f^{(i)}$ in fixed-point arithmetic in shares $r^{(j,i)}$ to server $j$, where for $j < J$, each $r^{(j,i)}$ is drawn randomly from $\{1, \ldots, B\}$ for a sufficiently large $B$ and where $r^{(J,i)} = (f^{(i)} - \sum_{j<J} r^{(j,i)}) \mod B$. The computation servers then run SecAgg among each other, yielding the sum of all inputs. All $r^{(j,i)}$ cancel out and the sum over all models $f^{(i)}$ remains. The secret sharing technique is information-theoretically secure if at least one computation server is honest. Security assumptions of SecAgg apply to the computation servers instead of the users. Although secure, this protocol is not robust against active attacks.

## 6 EXPERIMENTAL RESULTS

We next elaborate on our evaluation and ablations; the experimental setup is postponed to Appx. F.1.

**Pretraining enables linear classification head.** As proposed by Tramèr & Boneh (2021), we used a SimCLR pretrained model[1] (Chen et al., 2020b) on ImageNet ILSVRC-2012 (Russakovsky et al., 2015) to get an embedding of the local data (cf. Fig. 10 in Appx. H for an embedding view). Their evaluation shows that a linear classification head on the embedding already performs better than a supervised-only model of the same size: $83.1\%$ vs $80.5\%$ accuracy (ImageNet, 1000 classes). The pretrained model is a ResNet152 with selective kernels (Li et al., 2019) and a width multiplier of 3 (overall: 795M parameters) in the fine-tuned SimCLR variant. For EMNIST, we inverted the image.

**Sensitive datasets.** CIFAR-10, CIFAR-100 (Krizhevsky, 2009), and federated EMNIST[2] (Cohen et al., 2017; Caldas et al., 2018) act as our sensitive datasets; all after SimCLR pretraining. CIFAR is frequently used as a benchmark dataset in DP and EMNIST in distributed learning literature. Both CIFAR datasets consist of 60,000 thumbnail-sized, colored images of 10 or 100 classes. Federated EMNIST consists of $\approx 750,000$ thumbnail-sized, grayscale images of 62 classes and is annotated with 3,400 user-partitions based on the author of the images: users have between 19 and 465 data points, on average $220 \pm 85$.

**Evaluation.** We compare BlindAvg with the DP-SVM-SGD and DP-SoftmaxReg-SGD variants to DP-SGD-based 1-layer federated learning (FL) and analyze four research questions:

*(RQ1) What is the privacy-utility tradeoff of blind averaging compared to gradient averaging like FL for a varying number of users?* Supported by theoretical arguments, we expect a natural ordering where (a) blindly averaged SoftmaxReg outperforms blindly averaged SVMs (cf. Sec. 3) as the number of classes increases and (b) blindly averaged SVM outperforms gradient averaging, such as FL, as the number of users increases (cf. Sec. 7): (a) Our experiments in a high-user scenario (cf. Fig. 2) and in a centralized setting (cf. Fig. 9) support the theory where the gap of DP-SoftmaxReg-SGD towards DP-SVM-SGD becomes wider the more classes the dataset has. (b) Our experiments support the hypothesis that FL performs worse the more users $w$ partake for the same data size (cf. Fig. 7) or gains less with an increasing data

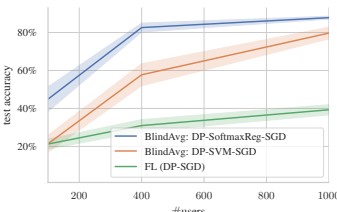

Figure 4: CIFAR-10 accuracy vs. #users with 50 data points per user for $(\varepsilon, \delta) = (0.6, 10^{-5})$. FL values are interpolated.

and user base (cf. Fig. 4). This effect is not unique to FL but resembles an inherent disadvantage of non-MPC-based averaging like commonly used gradient averaging which scales the noise with $\mathcal{O}(\sqrt{w})$. In contrast, blind averaging SVM-SGD or DP-SoftmaxReg-SGD loses little performance with an increasing number of users (cf. Fig. 7). Note that, the performance gap in blind averaging between 1 and 100 users is largely due to our assumption of $t = 50\%$ dishonest users which scales the noise by $\sqrt{2}$. For $t = 1$, we would deactivate this disadvantage.

*(RQ2) How well does non-private blind averaging work with varying regularization weights?* Fig. 5 presents this connection.

---

[1] accessible at `https://github.com/google-research/simclr`, Apache-2.0 license
[2] ref: `https://tensorflow.org/federated/api_docs/python/tff/simulation/datasets/emnist`

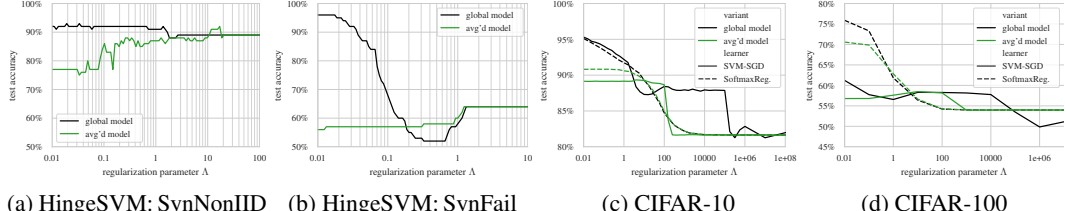

(a) HingeSVM: SynNonIID    (b) HingeSVM: SynFail    (c) CIFAR-10    (d) CIFAR-100

Figure 5: **BlindAvg vs. a global model for varying regularization** $\Lambda$**:** For HingeSVM and SoftmaxReg, BlindAvg converges to the global non-private model with increasing $\Lambda$ and all models maintain close task-performance for a mid-range regularization. BlindAvg fails for tasks not robust under regularization (i.e. those that only perform well with a small $\Lambda$) like the deliberately designed SynFail (cf. Fig. 3) but succeeds at robust tasks like SynNonIID (cf. Fig. 3), CIFAR-10, or CIFAR-100. In contrast to HingeSVM, SVM-SGD does not converge without error as it uses with a relaxation of the hinge loss: the Huber loss where non-support-vectors can influence the model independent of the regularization as there is no hard cutoff at the margin like in a hinge-loss SVM.

*(RQ3) How robust is BlindAvg's utility if the local data is non-IID?* We observe for strongly biased non-IID data (cf. Tbl. 2) that on CIFAR-10 the utility decline of DP-SVM-SGD is small whereas DP-SoftmaxReg-SGD needs more users for a similar utility preservation since it is more sensitive to noise. For real-world federated EMNIST with unbalanced local data sizes $n^{(i)}$, we observe a notable performance gap between the averaged (3,400 users) and global variant (1 user) in both learners (cf. Fig. 7 (d)). We weigh each local SVM by $n^{(i)}$ to achieve the constant sensitivity per user. This has a utility disadvantage against non-weighted averaging visible in the experiments. Still, SoftmaxReg surpasses the performance of FL.

Table 2: **Strongly biased non-IID experiments** ($\varepsilon = 1.2$): each user has exclusive access to only one class. We report the non-IID accuracy and compare it to our regular experiments in percentage points (pp). As in Fig. 6, we extrapolate the accuracy on datasets 67 times larger using less noise.

| | | ACCURACY ON DATASET | |
|---|---|---|---|
| BLINDLY AVERAGING VARIANTS | DATASET MULTIPLIER | CIFAR-10 | CIFAR-100 |
| BLINDAVG: DP-SVM-SGD | 1X | 85 % ($-2$ PP) | – |
| BLINDAVG: DP-SVM-SGD | 67X | 87 % ($-3$ PP) | – |
| BLINDAVG: DP-SOFTMAXREG-SGD | 1X | 42 % ($-49$ PP) | 1.6 % ($-43$ PP) |
| BLINDAVG: DP-SOFTMAXREG-SGD | 67X | 88 % ($-4$ PP) | 58 % ($-4$ PP) |

*(RQ4) How does BlindAvg perform for user-level privacy if scaled with significantly more users.* Fig. 6 presents this connection. Most notably, for ($\Upsilon \geq 2$)-group DP, we observe a user-level sensitivity of $^{2R}/_w$ (cf. Cor. 5.2) to be mostly tighter than the data point dependent one by Wu et al. (2017).

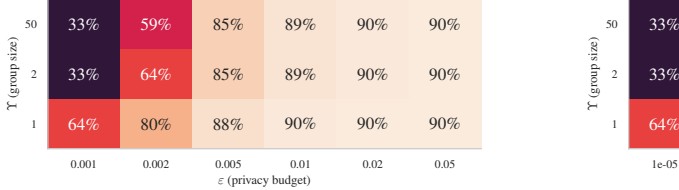 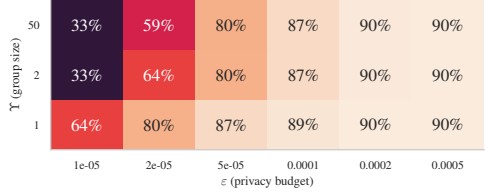

Figure 6: **Accuracy for local aggregators (group sizes** $\Upsilon < 50$**, cf. Cor. E.3) and user-level privacy (**$\Upsilon = 50$**, cf. Cor. 5.2) in BlindAvg** for the DP-SoftmaxReg-SGD learner on CIFAR-10 data (left: $\delta = 10^{-10}$; right: $\delta = 10^{-12}$) with roughly 200,000 (left) and 20,000,000 (right) users. We take the best accuracy for $\Upsilon$-group DP (protects $\Upsilon$ out of $n = 50$ local data points) and user-level privacy (protects the entire user) which already for $\Upsilon = 2$ pivots to the latter. To emulate more users and larger datasets, we interpolated the accuracy of 1,000 users on 50 data points each to a rescaled $\varepsilon$-value ($\varepsilon' := {}^{1000 \cdot \varepsilon \cdot \Upsilon}/_{n_{users}}$, approximates the actual $\varepsilon'$). Thus, we report pessimistic accuracies as the accuracy does not increase with the users; actually averaging over all users should perform better.

**Computation costs.** Extrapolating Bell et al. (2020, Table 2), we need for BlindAvg with model size $\ell \approx 100{,}000$ (CIFAR-10) and $1{,}000$ users $\leq 0.2s$ user time and $40s$ server time.

# 7 RELATED WORK

Table 3: **Comparison to related work** for $w$ users with $n$ data points each and $M$ training iterations: utility guarantee, DP noise scale, and number of secure summation (SecSum) invocations. 'Utility: CC' denotes whether blind model averaging converges to the centralized setting before noise. The convergence rate of Jayaraman et al. (2018, Output perturbation) depends on the dataset size $nw$ whereas we truly converge with the number of iterations (cf. Thm. 4.3). FL + SecSum is by Agarwal et al. (2018); Kairouz et al. (2021). ($\checkmark$) denotes experimental evidence without formal proof.
*: In FL, an untrusted aggregator combines the differentially private updates (users add noise and norm-clip those); it does not invoke SecSum but needs a communication round per training iteration.

| SVM Algorithms | Utility: CC | Noise | SecSum Invoc. |
|---|---|---|---|
| Jayaraman et al. (2018), gradient perturb. | $\checkmark$ | $\mathcal{O}(\sqrt{M}/nw)$ | $\mathcal{O}(\log(nw))$ |
| Jayaraman et al. (2018), output perturb. | ($\checkmark$) | $\mathcal{O}(1/nw)$ | 1 |
| **BlindAvg: SVM (ours)** | $\checkmark$: Thm. 4.3 | $\mathcal{O}(1/nw)$ | 1 |
| SoftmaxReg Algorithms | | | |
| DP federated learning (FL) | $\checkmark$ | $\mathcal{O}(\sqrt{M}/(n\sqrt{w}))$ | $M^*$ |
| FL + SecSum | $\checkmark$ | $\mathcal{O}(\sqrt{M}/nw)$ | $M$ |
| **BlindAvg: SoftmaxReg (ours)** | ($\checkmark$) | $\mathcal{O}(1/nw)$: Thm. 3.1 | 1 |
| Baseline: Centralized training | $\checkmark$ | $\mathcal{O}(1/nw)$ | 0 |

Here, and in Tbl. 3 we discuss the most related work and continue the discussion in Appx. C. For interactive ERM via gradient perturbation, Jayaraman et al. (2018) show strong utility-privacy tradeoffs. It requires $\mathcal{O}(\log(nw))$ invocations of secure summation (SecSum) for $w$ users with $n$ data points each. For non-interactive learning, however, Jayaraman et al. (2018) show output perturbation results using Chaudhuri et al. (2011) for which they only show that the convergence bounds from the local training are preserved, thus leaving a gap of $1/w$ to centralized learning. We prove that SVM learning converges in the limit, closing the $1/w$ gap, and we experimentally show that SoftmaxReg leads to strong results. Moreover, their work does not exclude leakage from the learner (only from the optimum), while we use bounds on the learner SGD (Wu et al., 2017).

In differentially private federated learning (FL) (McMahan et al., 2017; Abadi et al., 2016) and also in follow-up work (Chen et al., 2022; Choquette-Choo et al., 2024), each user protects its local data by only submitting noisy local model updates protected via DP guarantees (DP-SGD). A central server then aggregates all incoming updates. However, DP-FL is interactive as its communication rounds increase with the number of training iterations. Moreover, FL achieves a weaker utility-privacy tradeoff than centralized learning, already for hundreds of users. FL's noise scales with $\mathcal{O}(\sqrt{w})$ and has a prohibitively high number of communication rounds.

# 8 LIMITATIONS & DISCUSSION

For datasets that work best with small regularization $\Lambda$ like the deliberately designed SYNFAIL, BlindAvg fails to capture a strong task-performance. Yet, with privacy even the global model works best with mid-range $\Lambda \geq 1$ as the noise scales with $\mathcal{O}(1/\Lambda)$. This points to a more general limitation of small $\Lambda$'s for DP ERMs. For unfavorable like non-IID datasets, blind averaging leads to a reduced signal-to-noise ratio, i.e. model parameters smaller than in the centralized setting. This may explain why blind averaged SoftmaxReg works better for less noise (cf. Tbl. 2). For unbalanced datasets, increasing $\Lambda$ too much to help convergence can lead to poor accuracy as the margin grows so much that not the same number of support vectors is chosen per label. For a detailed discussion, cf. Appx. B.

**Future work** can theoretically capture the utility-link of BlindAvg to the global model for any $\Lambda$ or extend our insight into BlindAvg convergence via the dual problem to other learners like SoftmaxReg.

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

# A    EXTENDED EVALUATION

(a) Dataset: CIFAR-10

(b) Dataset: CIFAR-10

(c) Dataset: CIFAR-100

(d) Dataset: Federated EMNIST

Figure 7: **Classification accuracy** compared to privacy budget $\varepsilon$ (in log-scale) of BlindAvg (cf. Sec. 5) and DP-SGD-based federated learning (FL) ($\delta = 10^{-5}$). (b) Different numbers of users with 50 data points per user. (a,c,d) We use all available data points of the dataset for each line, spreading them among a differing number of users. For BlindAvg and more than 1 user, we assume honest noise of only $t = 50\%$ of the users, thus scaling the noise by a factor of $\sqrt{2}$.

# B    EXTENDED LIMITATIONS

**Distributional shift between the public and sensitive dataset.**    For pretraining, we leverage contrastive learning. While very effective generally, it is susceptible to performance loss if the shape of the sensitive data used to train an SVM or SoftmaxReg is significantly different from the shape of the initial public data.

**Input Clipping.**    We require bounded input data for a bounded sensitivity. In many pretraining methods like SimCLR, no natural bound exists thus we norm-clip the input data by a constant $c$. To provide a data-independent $c$ on CIFAR-10 and EMNIST, $c$ is based on CIFAR-100 (here: $34.854$); its similar data distribution encompasses the output distribution of the pretraining reasonably well. For CIFAR-100, $c$ is based on CIFAR-10 (here: $34.157$).

**Hyperparameter Search.**    In SVM-SGD, we have two important hyperparameters that influence the noise scale: the regularization weight $\Lambda$ and the predictor radius $R$. In the noise scale subterm "$c/\Lambda + R$", the maximal predictor radius is naturally significantly smaller than $c/\Lambda$ due to the regularization penalty. Thus, a bad-tuned $R$ often does not have as large of a utility impact as a bad-tuned $\Lambda$. Estimating parameters for a fixed $\varepsilon$ from public data is called hyperparameter freeness in prior work (Iyengar et al., 2019). For the other $\varepsilon$ values, we can estimate $\Lambda$ by fitting a (linear) curve on related public data (proposed by Chaudhuri et al. (2011)) or synthetic data (proposed by AMP-NT (Iyengar et al., 2019)) as smaller $\varepsilon$ prefer a higher $\Lambda$ and vice versa.

**Blind averaging – Signal-to-noise ratio.**    For unfavorable yet balanced local datasets, we identify as a main limitation of blind averaging a reduced signal-to-noise ratio: the model is not as large as in the centralized setting w.r.t. the sensitivity analysis. This effect would explain why our experiments show that blind averaged SoftmaxReg works better with very little noise (cf. Tbl. 2).

For SVMs, our formal characterization of the effect of blind averaging enables us to describe its limitations more precisely. In summary, we see two effects that reduce the signal-to-noise ratio. One effect comes from the requirement of the SVM training that the model with the smallest norm shall be found that satisfies the soft-margin constraints of the training data points. The local SVM training has fewer data points and, thus, fewer constraints. Hence, unfavorable local data sets will lead to a smaller model. Another effect comes from the averaging itself. Unfavorable local data sets can lead to local models that point in very different directions. When averaging these models, their norm naturally decreases as for any two vectors $a, b \in \mathbb{R}^p$ we have $0.5\|a + b\|_2 \leq 0.5(\|a\|_2 + \|b\|_2)$, and this discrepancy is larger the smaller the inner product is.

**Blind averaging – Unbalanced data.** Convergence holds if all data points are support vectors (SV) which implies a large margin. Yet a regularly trained SVM chooses roughly equally many SVs per class: by the dual problem, we have the constraint $y^T \alpha = 0$ for labels $y_j \in \{-1, 1\}$ and dual coefficients $\alpha$. If we have an SV inside the margin then $\alpha_j = \Lambda^{-1}$. Hence, enlarging the margin such that all data points are SVs can lead to poor utility performance. Moreover, unbalanced local data can deteriorate the performance of blind averaging as observed in the EMNIST experiments (cf. Fig. 7 (d)) as we favor privacy, i.e. a constant sensitivity per user, above utility, i.e. optimal local scaling.

**Active attacks.** Active attackers may deviate from the protocol or send maliciously construed local models. If the used secure summation protocol is resilient against active adversaries and can still guarantee that only the sum of the inputs is leaked, privacy is preserved. This follows from analyzing our algorithm for just the honest users and then leveraging the post-processing property of differential privacy. Secure summation protocols such as Bell et al. (2020) leak partial sums under active attacks and will diminish the privacy offered by our work against such adversaries as well.

## C    EXTENDED RELATED WORK

**Other Privacy-preserving Distributed Learning Protocols.** The noise overhead of FL can be completely avoided by protocols that rely on cryptographic methods to hide intermediary training updates from a central aggregator. Several secure distributed learning methods protect the contributions during training but do not come with privacy guarantees for the model such as DP: an attacker, e.g. a curious training party, can potentially extract information about the training data from the model. As we focus on differentially private distributed learning methods, we will neglect those methods.

cpSGD (Agarwal et al., 2018) is a protocol that utilizes secure multi-party computation (MPC) methods to honestly generate noise and compute DP-SGD. While cpSGD provides the full flexibility of SGD, it does not scale to millions of users as it relies on expensive MPC methods. Truex et al. (2019) relies on a combination of MPC and DP methods which also does not scale to millions of users.

Another line of research aims for the stronger privacy goal of protecting a user's entire input, called local DP, during distributed learning (Balle et al., 2020b; Girgis et al., 2021). Due to the strong privacy goal, federated learning with local DP tends to achieve weaker accuracy. With Cor. E.3, evaluated in Fig. 6, we show how BlindAvg achieves a comparable guarantee via group privacy and MPC-based aggregation overhead: given enough users, any user can protect their entire dataset at once while we still reach good accuracy.

For DP SVM training, other methods besides output perturbation Chaudhuri et al. (2011); Wu et al. (2017) such as objective perturbation (Chaudhuri et al., 2011; Kifer et al., 2012; Iyengar et al., 2019; Bassily et al., 2019) and gradient perturbation (Bassily et al., 2014; Wang et al., 2017; Feldman et al., 2018; Bassily et al., 2019; Feldman et al., 2020; Yu et al., 2021) exist. Output perturbation noises a trained model once calibrated to the model's sensitivity which marks it an ideal candidate for non-interactive learning. In contrast, objective perturbation noises the objective function while gradient perturbation noises the intermediary gradient updates. Yet, DP requires objective perturbation and the privacy amplification by iteration variant of gradient perturbation Feldman et al. (2018; 2020) to leak no intermediary model updates which marks in the convex SVM setting their amplification above DP-SGD. To protect all of these updates in a distributed setting, the training has to be performed in expensive protocols like MPC or homomorphic encryption. Other gradient perturbation methods that leak intermediary gradient Bassily et al. (2014); Abadi et al. (2016); Yu et al. (2021) need one MPC

invocation per iteration. All these variants have a large communication overhead and are thus highly interactive.

**Trustworthy Distributed Noise Generation.** One core requirement of MPC-based distributed learning is honestly generated and unleakable noise, otherwise our privacy guarantees would not hold anymore. There is a rich body of work on distributed noise generation (Moran et al., 2009; Dwork et al., 2006a; Kairouz et al., 2015b; 2021; Goryczka & Xiong, 2015). So far, however, no distributed noise generation protocol scales to millions of users. Thus, we use a simple, yet effective technique: we add enough noise if at least a fraction of them (say $t = 50\%$) are not colluding to violate privacy by sharing the noise they generate with each other.

## D  EXTENDED PRELIMINARIES

---

**Algorithm 3** Example 2.5: DP SVM $T_\xi$ =SVM-SGD($D$) with hyperparameters $\xi :=$ $(K, c, M, h, \Lambda, R)$

---

**Input:** dataset $D := \{(x_j, y_j)\}_{j=1}^n$ where data point $x_j$ is structured as $[1, x_{j,1}, \ldots, x_{j,p}]$; #classes $K$; input clipping bound: $c \in \mathbb{R}_+$; #iterations $M$; Huber loss relaxation $h \in \mathbb{R}_+$; regularization parameter: $\Lambda \in \mathbb{R}_+$; model clipping bound: $R \in \mathbb{R}_+$

**Result:** $\{f_M^{(k)}\}_{k \in \{1,\ldots,K\}} \in \mathbb{R}^{(p+1) \times K}$: models with hyperplanes $\in \mathbb{R}^p$ and intercepts $\in \mathbb{R}$

$clipped(x) := c \cdot x / \max(c, \|x\|)$

$\mathcal{J}(f, D, k) := \frac{1}{n} \sum_{(x,y) \in D} \ell_{\text{huber}}(h, y\langle f, clipped(x)\rangle \cdot (1[y = k] - 1[y \neq k])) + \frac{\Lambda}{2}\langle f, f\rangle$

**for** $k$ **to** $1, \ldots, K$ **do**

  **for** $m$ **to** $1, \ldots, M$ **do**

  $f_m^{(k)} \leftarrow f_{m-1}^{(k)} - \tau_m \nabla \mathcal{J}(f_{m-1}^{(k)}, (x_j, y_j), k)$, with learning rate $\tau_m = \min(\frac{1}{\beta}, \frac{1}{\Lambda m})$,

  $\beta = \sqrt{(c^2/2h + \Lambda)^2 + p\Lambda^2}$, and index $j = m \bmod n$.

  $f_m^{(k)} := R \cdot f_m^{(k)} / \|f_m^{(k)}\|$ ⠀⠀⠀⠀⠀⠀⠀⠀⠀⠀⠀⠀⠀⠀⠀⠀⠀⠀⠀⠀⠀ {projected SGD}

---

### D.1  DIFFERENTIAL PRIVACY

To ease our analysis, we consider a randomized mechanism $M$ to be a function translating a database to a random variable over possible outputs. Running the mechanism then is reduced to sampling from the random variable. With that in mind, the standard definition of differential privacy looks as follows.

**Definition D.1** ($\approx_{\varepsilon,\delta}$ relation). Let $Obs$ be a set of observations, and $\text{RV}(Obs)$ be the set of random variables over $Obs$, and $\mathcal{D}$ be the set of all databases. A randomized algorithm $M : \mathcal{D} \to \text{RV}(Obs)$ for a pair of datasets $D, D' \in \mathcal{D}$, we write $M(D) \approx_{\varepsilon,\delta} M(D')$ if for all tests $S \subseteq Obs$ we have $\Pr[M(D) \in S] \leq \exp(\varepsilon) \Pr[M(D') \in S] + \delta$ with $\varepsilon, \delta \in \mathbb{R}_+$.

**Definition D.2** (Differential Privacy). Let $Obs$ be a set of observations, and $\text{RV}(Obs)$ be the set of random variables over $Obs$, and $\mathcal{D}$ be the set of all databases. A randomized algorithm $M : \mathcal{D} \to \text{RV}(Obs)$ for all pairs of databases $D, D' \in \mathcal{D}$ that differ in at most 1 element is a $(\varepsilon, \delta)$-DP mechanism if we have $M(D) \approx_{\varepsilon,\delta} M(D')$.

In our proofs, we utilize a randomized variant of the sensitivity as proposed by Wu et al. (2017) to achieve DP ERMs via SGD Training.

**Definition D.3** (Randomized Sensitivity). Let $q : (D, r) \to \mathbb{R}$ be a randomized function on dataset $D$ and randomness $r$. The *sensitivity* of $q$ is defined as $s = \max_{D \sim_1 D'} \max_r \|q(D, r) - q(D', r)\|$, where $D \sim_1 D'$ denotes that the datasets $D$ and $D'$ differ in at most one element. We say that $q$ is an $s$-sensitivity-bounded function.

In the context of machine learning, the randomized algorithm represents the training procedure of a predictor. Our distinguishing element is one data record of the database.

**Computational Differential Privacy** Note that because of the secure summation, we technically require the computational version of differential privacy (Mironov et al., 2009), where the differential

privacy guarantees are defined against computationally bounded attackers; the resulting increase in $\delta$ is negligible and arguments about computationally bounded attackers are omitted to simplify readability.

**Definition D.4** (Computational $\approx^c_{\varepsilon,\delta}$ Differential Privacy)**.** Let $\mathcal{D}$ be the set of all databases and $\eta$ a security parameter. A randomized algorithm $M : \mathcal{D} \to \mathrm{RV}(Obs)$ for a pair of datasets $D, D' \in \mathcal{D}$, we write $M(D) \approx^c_{\varepsilon,\delta} M(D')$ if for any polynomial-time probabilistic attacker $\Pr[A(M(D)) = 0] \leq \exp(\varepsilon) \Pr[A(M(D')) = 1] + \delta(\eta)$. For all pairs of databases $D, D'$ that differ in at most 1 element $M$ is a computational $(\varepsilon, \delta(\eta))$-DP mechanism if we have $M(D) \approx^c_{\varepsilon,\delta} M(D')$.

## D.2 Secure Summation

Hiding intermediary local training results as well as ensuring their integrity is provided by an instance of secure multi-party computation (MPC) called secure summation (Bonawitz et al., 2017; Bell et al., 2020). It is targeted to comply with distributed summations across a huge number of parties. In fact, Bell et al. (2020) has a computational complexity for $w$ users on an $l$-sized input of $\mathcal{O}(\log^2 w + l \log w)$ for the client and $\mathcal{O}(w(\log^2 w + l \log w))$ for the server as well as a communication complexity of $\mathcal{O}(\log^2 w + l)$ for the client and $\mathcal{O}(w(\log w + l))$ for the server thus enabling an efficient run-through of roughly $10^9$ users without biasing towards computationally equipped users. Additionally, it offers resilience against client dropouts and colluding adversaries, both of which are substantial features for our distributed setting.

**Definition D.5** (Secure Summation)**.** Let $\mathcal{F}(s_1, \ldots, s_n) := \sum_{i=1}^w s_i$. We say that $\pi_{SecSum}$ is secure summation if there is a probabilistic polynomial-time simulator $\mathrm{Sim}_{\mathcal{F}}$ such that if a fraction of clients is corrupted ($C \subseteq \left\{ U^{(1)}, \ldots, U^{(w)} \right\}$, $|C| = \gamma w$), $\mathrm{Real}_{\pi_{SecSum}}(s_1, \ldots, s_w)$ is statistically indistinguishable from $\mathrm{Sim}_{\mathcal{F}}(C, \mathcal{F}(s_1, \ldots, s_w))$, i.e., for an unbounded attacker $\mathcal{A}$ there is a negligible function $\nu$ such that

$$\mathrm{Advantage}(\mathcal{A}) = |\Pr[\langle \mathcal{A}, \mathrm{Real}_{\pi_{SecAgg}}(s_1, \ldots, s_w)\rangle = 1] -$$
$$\Pr[\langle \mathcal{A}, \mathrm{Sim}_{\mathcal{F}}(C, \mathcal{F}(s_1, \ldots, s_w))\rangle = 1]| \leq \nu(\eta).$$

Here, $\mathrm{Sim}_{\mathcal{F}}$ is a potentially interactive simulator that only has access to the sum of all elements and the (sub)-set of corrupted clients. The adversary is unable to distinguish interactions and outputs of the simulator from those of the real protocol. For a detailed definition of the network execution $\mathrm{Real}_{\pi}$ using the notion of interactive machines we refer to Appx. I.2.

The following theorem is proven for global network attackers that are passive and statically compromised parties. Formally, the theorem holds for all attackers $(\mathcal{A}', \mathcal{A}'')$ of the following form. $\mathcal{A}'$ internally runs $A''$ and ensures that only static compromization is possible and that the attacker remains passive.

**Theorem D.6** (Secure Aggregation $\pi_{SecAgg}$ in the semi-honest setting exists (Bell et al., 2020))**.** *Let $s_1, \ldots, s_n$ be the $d$-dimensional inputs of the clients $U^{(1)}, \ldots, U^{(w)}$. Let $\mathcal{F}$ be the ideal secure summation function: $\mathcal{F}(s_1, \ldots, s_n) := \sum_{i=1}^w s_i$. If secure authentication encryption schemes and authenticated key agreement protocol exist, the fraction of dropouts (i.e., clients that abort the protocol) is at most $\rho \in [0, 1]$, at most a $\gamma \in [0, 1]$ fraction of clients is corrupted ($C \subseteq \left\{ U^{(1)}, \ldots, U^{(w)} \right\}$, $|C| = \gamma w$), and the aggregator is honest-but-curious, then there is a secure summation protocol $\pi_{SecAgg}$ for a central aggregator and $w$ users that securely emulates $\mathcal{F}$ as in Def. D.5.*

## D.3 Pretraining to boost DP Performance

Recent work (Tramèr & Boneh, 2021; De et al., 2022) has shown that strong feature extractors (such as SimCLR (Chen et al., 2020a;b)), trained in an unsupervised manner, can be combined with simple learners to achieve strong utility-privacy tradeoffs for high-dimensional data sources like images. As a variation to transfer learning, it delineates a two-step process (cf. Fig. 8), where a simplified representation of the high-dimensional data is learned first before a tight privacy algorithm like DP-SVM-SGD or DP-SoftmaxReg-SGD conducts the prediction process on these simplified representations. For that, two data sources are compulsory: a public data source used for a framework that learns a pertinent simplified representation and our sensitive data source that conducts the

Figure 8: Pretraining: Schematic overview. Dashed lines denote data flow during training and solid lines during inference.

prediction process in a differentially private manner. Thereby, the sensitive dataset is protected while strong expressiveness is assured through the feature reduction network. Note that a homogeneous data distribution of the public and the sensitive data is not necessarily required.

Recent work has shown that for several applications such representation reduction frameworks can be found, such as SimCLR for pictures, FaceNet for face images, UNet for segmentation, or GPT for language data. Without loss of generality, we focus in this work on the unsupervised SimCLR feature reduction network (Chen et al., 2020a;b). SimCLR uses contrastive loss and image transformations to align the embeddings of similar images while keeping those of dissimilar images separate (Chen et al., 2020a). It is based upon a self-supervised training scheme called contrastive loss where no labeled data is required. Labelless data is especially useful as it exhibits possibilities to include large-scale datasets that would otherwise be unattainable due to the labeling efforts needed.

### D.4 DUAL SVM REPRESENTATION

With the representer theorem, we can completely describe a converged SVM $T_\xi(D)$ on $n$-sized dataset $D$ using a sum of the dual coefficients and the data points: $T_\xi(D) = \sum_{j=1}^n \alpha_j x_j$. The requirements for the representer theorem to hold are listed in the theorem below and include most notably an L2-regularized ERM objective.

**Theorem D.7** (Representer theorem, cf. Argyriou et al. (2009) Lem 3, Thm 8)**.** *Given a configuration $\zeta$, a local dataset $D^{(i)} := \{ (x_j, y_j) \}_{j=1}^n \subseteq \mathcal{H} \times \mathcal{Y}$ on a Hilbert space $\mathcal{H}$ with $\dim(\mathcal{H}) \geq 2$ and label space $\mathcal{Y}$, and a locally trained model $f^{(i)}$ of a learner $T_\xi$ such that there exists a solution that belongs to $\mathrm{span}(\{ x_j \}_{j=1}^n)$, where $f^{(i)} = T_\xi(D^{(i)}) = \mathrm{argmin}_{f \in \mathcal{H}} E(\{ \langle f, x_j \rangle, y_j \}_{j=1}^n) + \Lambda \Omega(f)$ for some arbitrary error function $E\colon (\mathbb{R} \times \mathcal{Y})^n \to \mathbb{R}$ and differentiable regularizer $\Omega\colon \mathcal{H} \to \mathbb{R}$. Then $T_\xi$ admits a solution of the form $f^{(i)} = \sum_{j=1}^n \alpha_j x_j$ for some $\alpha_j \in \mathbb{R}$ if and only if $\forall f \in \mathcal{H}\colon \Omega(f) = h(\langle f, f \rangle)$ with $h\colon \mathbb{R}_+ \to \mathbb{R}$ as a non-descreasing function.*

In the case of SVM-SGD and SoftmaxReg-SGD, we have $\Omega = \langle f, f \rangle$ which fulfills the requirements of the representer theorem since $h(z) = z$ is a linear function and the learner $T$ follows the definitions after convergence: $E(\{ \langle f, x_j \rangle, y_j \}_{j=1}^n) = \frac{1}{n} \sum_{(x,y) \in D^{(i)}} \ell_{\mathrm{huber}}(y \langle f, x \rangle)$ is the error function of SVM-SGD and $E(\{ \langle f, x_j \rangle, y_j \}_{j=1}^n) = \frac{1}{n} \sum_{(x,y) \in D^{(i)}} \ell_{\mathrm{softmax}}(y \langle f, x \rangle)$ the one of SoftmaxReg-SGD.

### D.5 STRONG CONVEXITY, LIPSCHITZNESS, AND SMOOTHNESS

Let $\langle \cdot, \cdot \rangle$ denote the inner product, $\|\cdot\|$ the L2-norm, and $\nabla$ the derivative operator.

**Definition D.8** (Strong convexity)**.** For all parameters $f$ and $f'$, function $q$ is $\Lambda$-strongly convex if

$$q(f) \leq q(f') + \langle \nabla q(f'), f - f' \rangle + \frac{\Lambda}{2} \|f - f'\|^2.$$

**Definition D.9** (Lipschitzness)**.** For all parameters $f$ and $f'$, function $q$ is $L$-Lipschitz continuous if

$$\frac{\|q(f) - q(f')\|}{\|f - f'\|} \leq L.$$

**Definition D.10** (Smoothness)**.** For all parameters $f$ and $f'$, function $q$ is $\beta$-smooth if

$$\frac{\|\nabla_f q(f) - \nabla_{f'} q(f')\|}{\|f - f'\|} \leq \beta.$$

# E    SECURITY OF BLINDAVG

First, we derive a tight output sensitivity bound. A naïve approach would be to release each individual predictor, determine the noise scale proportionally to $\tilde{\sigma} := \sigma$ (cf. Cor. 2.4), showing $(\varepsilon, \delta)$-DP for every user. We can save a factor of $\sqrt{w}$ by leveraging that $w$ is known to the adversary and we have at least $t = 50\%$. Consequently, local noise of scale $\tilde{\sigma} := \sigma \cdot 1/\sqrt{t \cdot w}$ is sufficient for $(\varepsilon, \delta)$-DP.

**Lemma E.1** (Privacy amplification via averaging). *For a configuration $\zeta$ as in Tbl. 1, BlindAvg($\zeta$) of Alg. 2 without noise, $avg(n^{(i)} \cdot T_\xi(D^{(i)}))$, has a sensitivity of $s' \cdot 1/w$ for each model if $s = s'/n$.*

The proof is in Appx. J.1.1. The sensitivity of the aggregate is bounded to $s' \cdot 1/w$ by rescaling the local models by $n^{(i)}$ which leads to local sensitivities independent of $n^{(i)}$ and allows blind averaging with varying local data sizes $n^{(i)}$. The sensitivities of $T = $ SVM-SGD (cf. Example 2.5) and $T = $ SoftmaxReg-SGD (cf. Thm. 3.1) fulfill the condition in the Lemma as they are proportional to $n^{-1}$, thus: $s' = s \cdot n$. Next, we show that locally adding noise per user $\tilde{\sigma}$ proportional to $\sigma \cdot n^{(i)}/\sqrt{w}$ and taking the mean over the users is equivalent to centrally adding noise $\tilde{\sigma}$ proportional to $\sigma \cdot n^{(i)}/w$. Adding dishonest noise can be treated as post-processing and does not impact privacy.

**Lemma E.2.** *For a configuration $\zeta$ as in Tbl. 1 and noise scale $\tilde{\sigma}$: $\frac{1}{w} \sum_{i=1}^{w} \mathcal{N}(0, (\tilde{\sigma} \cdot 1/\sqrt{w})^2) = \mathcal{N}(0, (\tilde{\sigma} \cdot 1/w)^2)$.*

The proof is in Appx. J.1.2. We now prove differential privacy for BlindAvg of Alg. 2 with noise scale $\tilde{\sigma} := \sigma \cdot 1/\sqrt{t \cdot w}$ and thus $\varepsilon \in \mathcal{O}(s'/\sqrt{t \cdot w} \cdot \sqrt{K_{\text{comp}}})$.

**Theorem 5.1** (Main Theorem, simplified). *For a configuration $\zeta$ as in Tbl. 1, BlindAvg($\zeta$) of Alg. 2 satisfies computational $(\varepsilon, \delta + \nu)$-DP with $\delta(\varepsilon, K_{comp})$ as in Def. 2.2 and a function $\nu$ negligible in the security parameter used in $\pi_{SecSum}$.*

The full statement and proof are in Appx. J.1.3. Simplified, the proof follows by applying the sensitivity (cf. Lem. E.1) to the Gaussian mechanism (cf. Lem. 2.3) where the noise is applied per user (cf. Lem. E.2). Next, we show how to protect the entire dataset of a single user (e.g., for distributed training via smartphones). The sensitivity-based bound on the Gaussian mechanism implies strong $\Upsilon$-group privacy results (see Appx. J.3.2), leading to security guarantees as in local DP with the overhead of secure summation.

**Corollary E.3** (Group-private variant). *For a configuration $\zeta$ as in Tbl. 1, BlindAvg($\zeta$) of Alg. 2 satisfies computational $(\Upsilon\varepsilon, \delta + \nu)$, $\Upsilon$-group DP with $\delta(\varepsilon, K_{comp})$ as in Def. 2.2 and $\nu$ negligible in the security parameter used in $\pi_{SecSum}$: for any pair of datasets $\mho, \mho'$ that differ at most $\Upsilon$ many data points, BlindAvg($\zeta(\ldots, \mho, \ldots)$) $\approx_{\varepsilon, \delta}$ BlindAvg($\zeta(\ldots, \mho', \ldots)$).*

# F    EXPERIMENTAL SETUP

## F.1    EXPERIMENTAL SETUP

Unless stated differently, we leveraged 5-repeated 6-fold stratified cross-validation for all differential privacy CIFAR experiments and 10-repeated cross-validation on the pre-defined split for differential privacy EMNIST ones. We conducted a hyperparameter search across $\Lambda$ and $R$ for each evaluation setting and $\varepsilon$ and reported the highest mean accuracy. Privacy Accounting has been done with the privacy bucket (Meiser & Mohammadi, 2018; Sommer et al., 2019) toolbox[3] or, for Gaussians without subsampling, with Sommer et al. (2019, Theorem 5) where both can be extended to multivariate Gaussians (cf. Appx. J.3.3). We set DP $\delta = 10^{-5}$ if not stated otherwise which is for CIFAR below $1/nw$ where $nw$ is the size of the combined local data.

Concerning computation resources, our Python implementation of the EMNIST experiments with 3,400 users took under a minute per user on a machine with 2x *Intel Xeon Platinum 8168* @24 Cores.

For DP-SVM-SGD-based experiments, we utilized projected SGD (PSGD) as used by Wu et al. (2017) and chose a batch size of 20 and the Huber loss with relaxation parameter $h = 0.1$. For CIFAR-10, we chose a hypothesis space radius $R \in \{0.04, 0.05, 0.06, 0.07, 0.08\}$, a regularization parameter $\Lambda \in \{10, 100, 200\}$, and trained for 500 epochs; for the variant where we protect the

---

[3]accessible at `https://github.com/sommerda/privacybuckets`, MIT license

whole local dataset, we chose $\Lambda \in \{\,0.5, 1, 2, 5\,\}$ and $R \in \{\,0.06, 0.07\,\}$ instead. For CIFAR-100, we chose $R \in \{\,0.04, 0.06, 0.08\,\}$, $\Lambda \in \{\,3, 10, 30, 100\,\}$, and trained for 150 epochs. For EMNIST, we chose in the 1 user setting $R = 0.08$ and $\Lambda \in \{\,3, 10, 100\,\}$ and in the 3,400 user setting $R = 0.04$ with $\Lambda \in \{\,30, 100\,\}$, and trained for 150 epochs.

For DP-SoftmaxReg-SGD-based experiments, we utilized PSGD with a batch size of 20 and trained for 150 epochs. For CIFAR-10, we chose $R \in \{\,0.1, 0.4, 0.6, 1.0\,\}$ and $\Lambda \in \{\,1, 3, 10, 30\,\}$; for the variant where we protect the whole local dataset, we chose $\Lambda \in \{\,0.5, 1\,\}$ and $R \in \{\,1, 3\,\}$ instead. For CIFAR-100, we chose the parameter combination $(R, \Lambda) \in \{\,(0.01, 100), (0.03, 30), (0.03, 100),$ $(0.1, 10), (0.1, 30), (0.3, 3), (0.3, 10), (1, 1), (1, 3)\,\}$. For EMNIST, we chose in the 1 user setting $R = 6$ and $\Lambda \in \{\,0.03, 0.1, 0.3\,\}$ and in the 3,400 user setting $R = 2$ with $\Lambda \in \{\,0.3, 1, 3\,\}$ and $R = 6$ with $\Lambda \in \{\,0.03, 0.1\,\}$.

For the experiments of Tbl. 2, we reported the results of the following hyperparameters: (CIFAR10, DP-SVM-SGD, regular & non-iid) $R = 0.06, \Lambda = 100$ for the dataset multiplier 1x and $R = 0.06, \Lambda = 10$ for the dataset multiplier 67x; (CIFAR10, DP-SoftmaxReg-SGD, regular) $R = 1.0, \Lambda = 1$ for both dataset multipliers 1x, 67x; (CIFAR10, DP-SoftmaxReg-SGD, non-iid) $R = 0.6, \Lambda = 3$ for both dataset multipliers 1x, 67x; (CIFAR100, DP-SoftmaxReg-SGD, regular) $R = 1.0, \Lambda = 3$ for dataset multiplier 1x and $R = 1.0, \Lambda = 1$ for both dataset multiplier 67x; (CIFAR100, DP-SoftmaxReg-SGD, non-iid) $R = 1.0, \Lambda = 3$ for both dataset multipliers 1x, 67x.

For the federated learning (FL) experiments, we utilized the *opacus*[4] PyTorch library (Yousefpour et al., 2021), which implements DP-SGD (Abadi et al., 2016). For a fair comparison, we halved the noise scale for the privacy accounting to comply with bounded DP as *opacus* currently uses unbounded DP: it now uses a noise scale proportional to $2C$ instead of $C$ with $C$ as the clipping bound. We loosely adapted our hyperparameters to the ones reported by Tramèr & Boneh (2021) who evaluated DP-SGD on SimCLR's embeddings for the CIFAR-10 dataset. In detail, the neural network is a single-layer perceptron with a $6\,144\,\mathrm{d}$ input and has the following configuration: (CIFAR-10) $61{,}450$ trainable parameters on a $10\,\mathrm{d}$ output, (CIFAR-100) $614{,}500$ trainable parameters on a $100\,\mathrm{d}$ output, and (EMNIST) $380{,}990$ trainable parameters on a $62\,\mathrm{d}$ output. The loss function is the categorical cross-entropy on a softmax activation function and training has been performed with SGD. We set the learning rate to 4, the Poisson sample rate (CIFAR) $q = {}^{1024}/_{50000}$ (EMNIST) $q = {}^{1024}/_{671585}$ which in expectation samples a batch size of 1024, trained for 40 epochs, and norm-clipped the gradients with a clipping bound $C = 0.1$.

In the distributed training scenario, instead of running an end-to-end experiment with full SecAgg clients, we evaluate a functionally equivalent abstraction without cryptographic overhead. In our CIFAR experiments, we randomly split the available data points among the users and emulated scenarios where not all data points were needed by taking the first training data points. The validation size remained constant. For FL, we kept a constant expected batch size: $q' = {}^{1024}/_{20000}$ for 20000 and $q'' = {}^{1024}/_{5000}$ for 5000 available data points $(wn)$. For FL, we emulated a larger number of users by dividing the noise scale $\sigma$ by $\sqrt{w}$ to the benefit of FL. Here, the model performance is not expected to differ as the mean of the gradients of one user is the same as the mean of gradients from different users: SGD computes, just as FL, the mean of the gradients. Yet, the noise will increase by a factor of $\sqrt{w}$.

# G  EXTENDED ABLATION STUDY (CENTRALIZED SETTING)

## G.1  SETUP OF THE ABLATION STUDY

For DP-SVM-SMO-based experiments, we used the *liblinear* (Fan et al., 2008) library via the Scikit-Learn method *LinearSVC*[5] for classification. *Liblinear* is a fast C++ implementation that uses the SVM-agnostic sequential minimal optimization (SMO) procedure. However, it does not offer a guaranteed and private convergence bound.

More specifically, we used the $L_2$-regularized hinge loss, an SMO convergence tolerance of $tol := 2 \cdot 10^{-12}$ with a maximum of 10,000 iterations which were seldom reached, and a log-

---

[4]accessible at `https://github.com/pytorch/opacus/`, Apache-2.0 license

[5]`https://scikit-learn.org/stable/modules/generated/sklearn.svm.LinearSVC.html`, BSD-3-Clause license

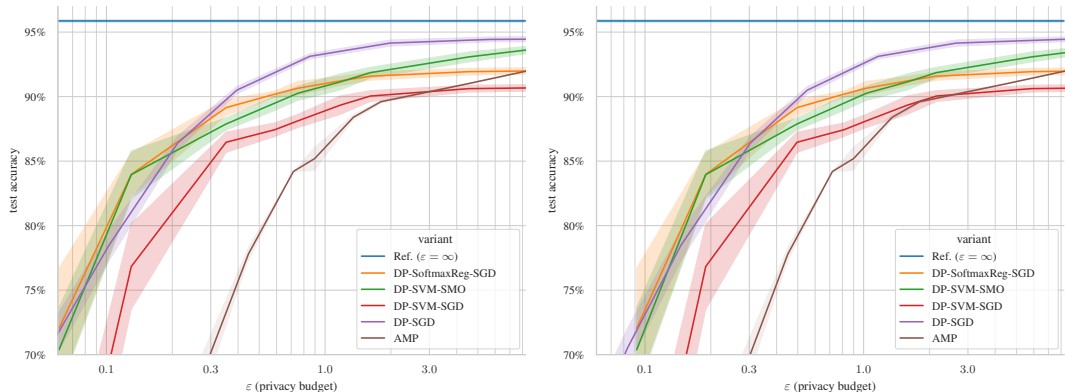

Figure 9: CIFAR-10 Accuracy vs. $\varepsilon$ budget of DP-SVM-SGD (cf. Example 2.5), DP-SoftmaxReg-SGD (cf. Alg. 1), DP-SVM-SMO where only the optima are perturbed, DP-SGD (single-layer) (Abadi et al., 2016), and AMP (SVM with objective perturbation) (Iyengar et al., 2019) (top: $\delta = 10^{-5}$, bottom: $\delta = 2 \cdot 10^{-8} \ll 1/\text{data\_size}$). For comparison, we report a non-private SVM baseline.

arithmically spaced inverse regularization parameter $C \in \left\{ \left\{ 3, 6 \right\} \cdot 10^{-8}, \left\{ 1, 2, 3, 6 \right\} \cdot 10^{-7}, \left\{ 1, 2, 3, 6 \right\} \cdot 10^{-6}, \left\{ 1, 2, 3, 6 \right\} \cdot 10^{-5}, \left\{ 1, 2 \right\} \cdot 10^{-4} \right\}$. To better fit with the *LinearSVC* implementation, the original loss function is rescaled by $1/\Lambda$ and $C$ is set to $1/\Lambda \cdot n$ with $n$ as the number of data points. Furthermore, for distributed DP-SVM-SMO training we extended the range of the hyperparameter $C$ – whenever appropriate – up to $3 \cdot 10^{-3}$ which becomes relevant in a scenario with many users and few data points per user. Similar to DP-SVM-SGD-based experiments, the best-performing regularization parameter $C$ was selected for each parameter combination.

The non-private reference baseline uses a linear SVM optimized via SMO with the hinge loss and an inverse regularization parameter $C = 2$ (best performing of $C \in \left\{ \leq 5 \cdot 10^{-5}, 0.5, 1, 2 \right\}$).

For the ablation study, we also included the Approximate Minima Perturbation (AMP) algorithm[6] (Iyengar et al., 2019) which resembles an instance of objective perturbation. There, we used a (80–20)-train-test split with 10 repeats and the following hyperparameters: $L \in \left\{ 0.1, 1.0, 34.854 \right\}$, eps_frac $\in \left\{ .9, .95, .98, .99 \right\}$, eps_out_frac $\in \left\{ .001, .01, .1, .5 \right\}$. We selected ($L = 1$, $eps\_out\_frac = 0.001$, $eps\_frac = 0.99$) as a good performing parameter combination for AMP. For better performance, we resembled the GPU-capable *bfgs_minimize* from the Tensorflow Probability package. To provide better privacy guarantees, we leveraged the results of Kairouz et al. (2015a); Murtagh & Vadhan (2016) for tighter composition bounds on arbitrary DP mechanisms.

## G.2 RESULTS OF THE ABLATION STUDY

For the extended ablation study, we considered the centralized setting (only 1 user) and compared different algorithms and different values for the privacy parameter $\delta$. The results are depicted in Fig. 9 and display five algorithms: firstly, the differentially private Support Vector Machine with SGD-based training DP-SVM-SGD (cf. Example 2.5), secondly, the differentially private Softmax-activated single-layer perceptron with SGD-based training DP-SoftmaxReg-SGD (cf. Alg. 1), thirdly, a similar differentially private SVM but with SMO-based training which does not offer a guaranteed and private convergence bound, fourthly, differentially private Stochastic Gradient descent (DP-SGD) (Abadi et al., 2016) applied on a 1-layer perceptron with the cross-entropy loss, and fifthly, approximate minima perturbation (AMP) (Iyengar et al., 2019) which is based upon an SVM with objective perturbation. Note that, only DP-SVM-SMO, DP-SVM-SGD, and DP-SoftmaxReg-SGD have an output sensitivity and are thus suited for this efficient BlindAvg scheme.

While all algorithms come close to the non-private baseline with rising privacy budgets $\varepsilon$, we observe that although DP-SGD performs best, DP-SVM-SMO and DP-SoftmaxReg-SGD come considerably close, DP-SVM-SGD has a disadvantage above DP-SVM-SMO of about a factor of

---

[6]reference implementation by the authors: `https://github.com/sunblaze-ucb/dpml-benchmark`, MIT license

2, and AMP a disadvantage of about a factor of $4$. We suspect that DP-SGD is able to outperform the variants other than DP-SoftmaxReg-SGD as it directly optimizes for the multi-class objective via the cross-entropy loss while others are only able to simulate it via the one-vs-rest (ovr) SVM training scheme. Additionally, DP-SGD has a noise-correcting property from its iterative noise application. The inherently multi-class DP-SoftmaxReg-SGD performs better than ovr-based DP-SVM-SGD indicating that a joint learning of all classes can boost performance. DP-SoftmaxReg-SGD additionally has a privacy advantage as it does not need to rely on sequential composition as it has an output sensitivity for all classes which is another factor that can lead to the boost of DP-SoftmaxReg-SGD above DP-SVM-SGD. Although DP-SVM-SMO also has an output sensitivity and renders better than DP-SVM-SGD, it does not offer a privacy guarantee when convergence is not reached. In the case of AMP, we have an inherent disadvantage of about a factor of $3$ due to an unknown output distribution, and thus bad composition results in the multi-class SVM. Here, the privacy budget of AMP roughly scales linearly with the number of classes.

For DP-SGD, DP-SVM-SGD, DP-SoftmaxReg-SGD, and DP-SVM-SMO, Fig. 9 shows that a smaller and considerably more secure privacy parameter $\delta \ll {}^1/nw$ where $nw$ is the sum of the size of all local datasets is supported although reflecting on the reported privacy budget $\varepsilon$.

# H  PRETRAINING VISUALIZATION

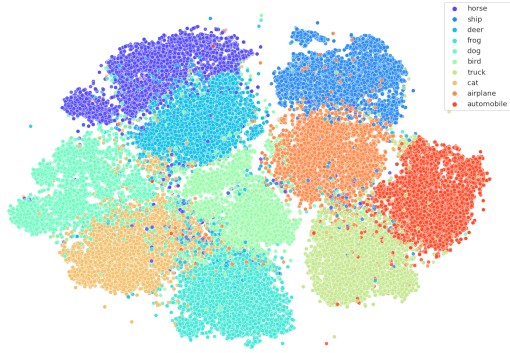

Figure 10: 2-d projection of the CIFAR-10 dataset via t-SNE (Van der Maaten & Hinton, 2008) with colored labels. t-SNE is defined on the local neighborhood thus global structures may be arbitrary.

# I  POSTPONED DEFINITIONS

## I.1  DP-SVM-SGD

**Definition I.1.** The Huber loss according to Chaudhuri et al. (2011, Equation 7) is with a relaxation parameter $h$ defined as

$$\ell_{\text{huber}}(h, z) := \begin{cases} 0 & \text{if } z > 1 + h \\ \frac{1}{4h}(1 + h - z)^2 & \text{if } |1 - z| \le h \\ 1 - z & \text{if } z < 1 - h \end{cases}.$$

## I.2  SECURE SUMMATION

Before formulating the security of the secure summation protocol, we define a network execution against a global network attacker that is active and adaptive. For self-containedness, we briefly present the notion of interactive machines and a sequential activation network execution. More general frameworks for such a setting include, e.g., the universal composability framework (Canetti, 2000).

We rely on the notion of interactive machines. For two interactive machines $X, Y$, we write $\langle X, Y \rangle$ for the interaction between $X$ and $Y$. We write $\langle X, Y \rangle = b$ to state that the machine $X$ terminates and outputs $b$.

**The network execution $\text{Real}_\pi$.** Next, we define a network execution against a global network attacker that is active and adaptive. Given a protocol $\pi$ with client and server code, we define an interactive machine $\text{Real}_\pi$ that lets each client party run the client code, lets the servers run the server code, and emulates a (sequential-activation-based) network execution, and interacts with another machine, called the attacker $\mathcal{A}$. The interaction is written as $\langle \mathcal{A}, \text{Real}_\pi \rangle$. Whenever within this network execution a party $B$ sends a message $m$ over the network to a party $C$, the interactive machine $\text{Real}_\pi$, sends this message $m$ to the attacker, activates the attacker, and waits for a response $m'$ from the attacker. $\text{Real}_\pi$ then lets this response $m'$ be delivered to party $C$, and activates party $C$. Moreover, the attacker $\mathcal{A}$ can send a dedicated message $(\texttt{compromise}, P)$ for compromising a party $P$ within the protocol execution. Whenever the attacker sends the message $(\texttt{compromise}, P)$ to the network execution $\text{Real}_\pi$, the network execution marks this party $P$ as compromised and sends the internal state of this party to the attacker $\mathcal{A}$. For each compromised party $P$, the attacker decides how $P$ acts. Formally, the network execution redirects each message $m$ that is sent to $P$ to the attacker $\mathcal{A}$ and awaits a response message $(m', P')$ from the attacker $\mathcal{A}$. Upon receiving the response $(m', P')$, the network execution $\text{Real}_\pi$ sends on behalf of $P$ the message $m'$ to the party $P'$.

For convenience, we write that a party $P$ runs *the client code of a protocol $\pi$ on input $m$* when the network execution runs for party $P$ the client code of $\pi$ on input $m$.

## J PROOFS

**Overview of All Statements with Proofs**

### J.1 SECURITY GUARANTEE OF BLINDAVG

#### J.1.1 PROOF OF LEM. E.1 (PRIVACY AMPLIFICATION VIA AVERAGING)

**Lemma E.1** (Privacy amplification via averaging). *For a configuration $\zeta$ as in Tbl. 1, BlindAvg($\zeta$) of Alg. 2 without noise, $\text{avg}(n^{(i)} \cdot T_\xi(D^{(i)}))$, has a sensitivity of $s' \cdot {}^1/w$ for each model if $s = {}^{s'}/n$.*

*Proof.* Without loss of generality, we consider one arbitrary model which corresponds to one class $k \in \{1, \ldots, K\}$ for SVM-SGD and all classes $k := K$ for SoftmaxReg-SGD. We know that $T_\xi$ is

an $s$-sensitivity bounded algorithm thus

$$s = \max_{D_0^{(i)} \sim D_1^{(i)}} \max_r \left| T_\xi(D_0^{(i)}, r) - T_\xi(D_1^{(i)}, r) \right|$$

with $D_0^{(i)}$ and $D_1^{(i)}$ as 1-neighboring datasets and $r$ as the randomness of $T$ (cf. Thm. J.3 for details about a randomized sensitivity involving $r$). For instance, for $T = \text{SVM-SGD}$ we have $s = \frac{2(\Lambda R + c)}{\Lambda n^{(i)}}$ (cf. Example 2.5) and for $T = \text{SoftmaxReg-SGD}$ we have $s = \frac{2(\Lambda R + \sqrt{2}c)}{\Lambda n^{(i)}}$ (cf. Thm. 3.1) which fulfill the condition $s \propto s'/n^{(i)}$.

By Alg. 2, we take the average of multiple local models, i.e. $\text{avg}(n^{(i)} \cdot T_\xi(D^{(i)})) = \frac{1}{w} \sum_{i=1}^{w} n^{(i)} \cdot T_\xi(D^{(i)}, r)$. The challenge element – i.e. the element that differs between $D_0^{(i)}$ and $D_1^{(i)}$ – is only contained in one of the $w$ models. By the application of the parallel composition theorem, we know that the sensitivity reduces to

$$\max_{\forall i=0,\ldots,w:\; D_0^{(i)} \sim D_1^{(i)}} \max_r \left| \frac{1}{w} \sum_{i=1}^{w} n^{(i)} \cdot T_\xi(D_0^{(i)}, r) - \frac{1}{w} \sum_{i=1}^{w} n^{(i)} \cdot T_\xi(D_1^{(i)}, r) \right|$$

$$= \max_{\forall i=0,\ldots,w:\; D_0^{(i)} \sim D_1^{(i)}} \max_r \left| \frac{n^{(i)}}{w} (T_\xi(D_0^{(i)}, r) - T_\xi(D_1^{(i)}, r)) \right|$$

$$\leq \max_{\forall i=0,\ldots,w:\; D_0^{(i)} \sim D_1^{(i)}} \max_r \left( \frac{n^{(i)}}{w} s \right) = s' \cdot \frac{1}{w}.$$

Hence, the constant $n^{(i)}/w$ factor reduces the sensitivity by a factor of $n^{(i)}/w$. $\qquad\square$

### J.1.2    PROOF OF LEM. E.2 (DISTRIBUTE NOISE TO USERS)

We recall Lem. E.2:

**Lemma E.2.** *For a configuration $\zeta$ as in Tbl. 1 and noise scale $\tilde{\sigma}$: $\frac{1}{w} \sum_{i=1}^{w} \mathcal{N}(0, (\tilde{\sigma} \cdot {}^1/\sqrt{w})^2) = \mathcal{N}(0, (\tilde{\sigma} \cdot {}^1/w)^2)$.*

*Proof.* We have to show that

$$\frac{1}{w} \sum_{i=1}^{w} \mathcal{N}(0, (\tilde{\sigma} \cdot \tfrac{1}{\sqrt{w}})^2) = \mathcal{N}(0, (\tilde{\sigma} \cdot \tfrac{1}{w})^2).$$

It can be shown that the sum of normally distributed random variables behaves as follows: Let $X \sim \mathcal{N}(\mu_X, \sigma_X^2)$ and $Y \sim \mathcal{N}(\mu_Y, \sigma_Y^2)$ two independent normally-distributed random variables, then their sum $Z = X + Y$ equals $Z \sim \mathcal{N}(\mu_X + \mu_Y, \sigma_X^2 + \sigma_Y^2)$ in the expectation.

Thus, in this case, we have

$$\frac{1}{w} \sum_{i=1}^{w} \mathcal{N}(0, (\tilde{\sigma} \cdot \tfrac{1}{\sqrt{w}})^2) = \frac{1}{w} \mathcal{N}(0, w \cdot (\tilde{\sigma} \cdot {}^1/\sqrt{w})^2) = \frac{1}{w} \mathcal{N}(0, \tilde{\sigma}^2).$$

As the normal distribution belongs to the location-scale family, we get $\mathcal{N}(0, (\tilde{\sigma} \cdot {}^1/w)^2)$. $\qquad\square$

### J.1.3    PROOF OF THM. 5.1 (MAIN THEOREM)

We state the full version of Thm. 5.1:

**Theorem 5.1** (Main Theorem, full). *For a configuration $\zeta$ as in Tbl. 1, a maximum fraction of dropouts $\rho \in [0, 1]$, and a maximum fraction of corrupted clients $\gamma \in [0, 1]$. Assume that secure summation $\pi_{SecSum}$ exists as in Def. D.5. Then BlindAvg$(\zeta)$ (cf. Alg. 2) satisfies computational $(\varepsilon, \delta + \nu_1)$-DP with $\delta(\varepsilon, K_{comp})$ as in Def. 2.2, for $\nu_1 := (1 + \exp(\varepsilon)) \cdot \nu(\eta)$ and a function $\nu$ negligible in the security parameter $\eta$ used in $\pi_{SecSum}$.*

*Proof.* We first show $(\varepsilon, \delta)$-DP for a variant $M_1$ of BlindAvg that uses the ideal summation protocol $\mathcal{F}$ instead of $\pi_{SecSum}$. We conclude that for BlindAvg (abbreviated as $M_2$) which uses the real secure summation protocol $\pi_{SecSum}$ for some negligible function $\nu_1$ $(\varepsilon, \delta + \nu_1)$-DP holds.

Recall that we assume at least $t \cdot w$ many honest users. As we solely rely on the honest $t \cdot w$ to contribute correctly distributed noise to the learner $T$, we have for each output model similar to Lem. E.2

$$\frac{1}{w} \sum_{i=1}^{t \cdot w} \mathcal{N}(0, (\tilde{\sigma} \cdot \frac{1}{\sqrt{w}})^2) = \sum_{i=1}^{t \cdot w} \mathcal{N}(0, (\tilde{\sigma} \cdot \frac{1}{w\sqrt{w}})^2)$$

$$= \mathcal{N}(0, (\tilde{\sigma} \cdot \frac{\sqrt{t \cdot w}}{w\sqrt{w}})^2) = \mathcal{N}(0, (\tilde{\sigma} \cdot \frac{\sqrt{t}}{w})^2).$$

Hence, we scale the noise parameter $\tilde{\sigma}$ with $1/\sqrt{t}$ and get

$$\frac{1}{w} \sum_{i=1}^{tw} \mathcal{N}(0, (\tilde{\sigma} \cdot \frac{1}{\sqrt{t}} \cdot \frac{1}{\sqrt{w}})^2) = \mathcal{N}(0, (\tilde{\sigma} \cdot \frac{1}{w})^2).$$

By Lem. E.1, Lem. E.2, and Lem. 2.3, we know that $M_1$ satisfies $(\varepsilon, \delta)$-DP (with the parameters as described above).

Considering an unbounded attacker $\mathcal{A}$, we know that for any pair of neighboring data sets $D, D'$ the following holds

$$\Pr[\mathcal{A}(\mathcal{M}_1(D)) = 1] \leq \exp(\varepsilon) \Pr[\mathcal{A}(\mathcal{M}_1(D')) = 1] + \delta$$

If $\pi_{SecSum}$ is a secure summation protocol, there is a negligible function $\nu$ such that for any neighboring data sets $D, D'$ (differing in at most one element) the following holds w.l.o.g.:

$$\Pr[\mathcal{A}(\mathcal{M}_2(D)) = 1] - \nu(\eta) \leq \Pr[\mathcal{A}(Sim_{\mathcal{F}}(\mathcal{M}_1(D))) = 1].$$

For the attacker $\mathcal{A}'$ that first applies $Sim$ and then $\mathcal{A}$, we get:

$$\Pr[\mathcal{A}(\mathcal{M}_2(D)) = 1] - \nu(\eta) \leq \exp(\varepsilon) \Pr[\mathcal{A}(Sim_{\mathcal{F}}(\mathcal{M}_1(D'))) = 1] + \delta$$
$$\leq \exp(\varepsilon)(\Pr[\mathcal{A}(\mathcal{M}_2(D')) = 1] + \nu(\eta)) + \delta$$

thus we have

$$\Pr[\mathcal{A}(\mathcal{M}_2(D)) = 1] \leq \exp(\varepsilon) \Pr[\mathcal{A}(\mathcal{M}_2(D')) = 1] + \delta + (1 + \exp(\varepsilon)) \cdot \nu(\eta).$$

From a similar argumentation it follows that

$$\Pr[\mathcal{A}(\mathcal{M}_2(D')) = 1] \leq \exp(\varepsilon) \Pr[\mathcal{A}(\mathcal{M}_2(D)) = 1] + \delta + (1 + \exp(\varepsilon)) \cdot \nu(\eta)$$

holds.

Hence, with $\nu_1 := (1 + \exp(\varepsilon)) \cdot \nu(\eta)$ the mechanism BlindAvg mechanism $\mathcal{M}_2$ which uses $\pi_{SecSum}$ is $(\varepsilon, \delta + \nu_1)$-DP. As $\nu$ is negligible and $\varepsilon$ is constant, $\nu_1$ is negligible as well.

$\square$

**Corollary J.1.** *Given a configuration $\zeta$, a maximum fraction of dropouts $\rho \in [0, 1]$, and a maximum fraction of corrupted clients $\gamma \in [0, 1]$, if secure authentication encryption schemes and authenticated key agreement protocol exist, then BlindAvg($\zeta$) (cf. Alg. 2) instantiated with $\pi_{SecSum} = \pi_{SecAgg}$ (Bell et al., 2020) satisfies computational $(\varepsilon, \delta + \nu_1)$-DP with $\delta(\varepsilon, K_{comp})$ as in Def. 2.2, for $\nu_1 := (1 + \exp(\varepsilon)) \cdot \nu(\eta)$ and a function $\nu$ negligible in the security parameter $\eta$ used in secure summation.*

This follows directly from Thm. 5.1, as by Thm. D.6, we know that $\pi_{SecAgg}(s_1, \ldots, s_n)$ securely emulates $\mathcal{F}$ (w.r.t. an unbounded attacker).

### J.1.4   PROOF OF COR. 5.2 (USER-LEVEL SENSITIVITY)

We recall Cor. 5.2:

**Corollary 5.2** (User-level sensitivity)**.** *Given a configuration $\zeta$ as in Tbl. 1 and a learner $T_\xi$, we say that $T_\xi$ is R-norm bounded if for any local dataset $D^{(i)}$: $\left\| T_\xi(D^{(i)}) \right\| \leq R$. Any R-norm bounded learner $T_\xi$ has a deterministic sensitivity $s = 2R$. In particular, $D^{(i)} \mapsto T_\xi(D^{(i)}) + \mathcal{N}(0, \tilde{\sigma}^2 s^2 I_{p \times K})$ satisfies $(\Upsilon\varepsilon, \delta)$, $\Upsilon$-group differential privacy with $\delta(\varepsilon, K_{comp})$ as in Def. 2.2 and $\Upsilon = n^{(i)} = |D^{(i)}|$.*

*Proof.* We know that the deterministic sensitivity of the learner $T_\xi$ is defined as $s = \max_{D \sim D'} \|T_\xi(D^{(i)}) - T_\xi(D'^{(i)})\|$ for $\Upsilon$-neighboring datasets $D^{(i)}, D'^{(i)}$ of the $i$-th user. Thus, in our case we have $s = 2R$ since for any dataset $\tilde{D}$, we have $T_\xi(\tilde{D}) \in [-R, R]$. As this holds independent on the dataset $\tilde{D}$ and by Lem. 2.3 and by Lem. J.4, we can protect any arbitrary number of data points per user, i.e. we have $\Upsilon$-group DP. $\qquad\square$

## J.2 NON-INTERACTIVE BLIND MODEL AVERAGING (BLINDAVG)

### J.2.1 PROOF OF COR. 4.1 (AVERAGED REPRESENTER THEOREM)

We recall Cor. 4.1:

**Corollary 4.1** (Averaged Representer theore)**.** *For a configuration $\zeta$ as in Tbl. 1, if a local learner $T_\xi$ admits a solution of the form $f^{(i)} = T_\xi(D^{(i)}) = \sum_{j=1}^n \alpha_j^{(i)} x_j^{(i)}$ (cf. Thm. D.7) then the average $avg(f^{(i)})$ admits a solution of the form $f = T_\xi(\mho) = \frac{1}{w} \sum_{i=1}^w \sum_{j=1}^n \alpha_j^{(i)} x_j^{(i)}$.*

*Proof.*
$$avg(f^{(i)}) = \tfrac{1}{w} \sum_{i=1}^w T_\xi(D^{(i)}) = \tfrac{1}{w} \sum_{i=1}^w \sum_{j=1}^N \alpha_j^{(i)} x_j^{(i)} = T_\xi(\mho) = f.$$
$\qquad\square$

### J.2.2 PROOF OF LEM. 4.2 (SUPPORT VECTORS OF AVERAGED SVMS)

We recall Lem. 4.2:

**Lemma 4.2** (Support Vectors of averaged SVMs)**.** *For configuration $\zeta$ as in Tbl. 1, a locally trained hinge-loss linear SVM $T$, $f^{(i)} := T_\xi(D^{(i)}) = \operatorname{argmin}_f \frac{1}{n} \sum_{(x,y) \in D^{(i)}} \max(0, 1 - y\langle f, x\rangle) + \Lambda \langle f, f\rangle$, has the support vectors $V^{(i)} := \{(x,y) \in D^{(i)} \mid y\langle f^{(i)}, x\rangle \le \langle f^{(i)}, f^{(i)}\rangle^{-1}\}$. Then, the average of these locally trained models $avg(T) = \frac{1}{w} \sum_{i=1}^w f^{(i)}$ has the support vectors $V = \bigcup_{i=1}^w V^{(i)}$.*

*Proof.* A learning problem that is based on a hinge-loss SVM fulfills the representer theorem requirements due to the L2-regularized ERM objective function. In fact, if a data point $x_j$ is a support vector, i.e. $x_j \in V$, then after successful training its corresponding $\alpha_j$ is restricted by $0 < \alpha_j \le \Lambda \wedge y_j = 1$ or $0 > \alpha_j \ge -\Lambda \wedge y_j = -1$, or $\alpha_j = 0$ (Ma & Ng, 2020, Equation 28-30). Thus, we denote $V^{(i)} = \left\{ x_j^{(i)} \in D^{(i)} \mid \alpha_j^{(i)} \ne 0 \right\}$. By Cor. 4.1, we have that the average of locally trained models $avg(T_\xi(D^{(i)})) = \frac{1}{w} \sum_{i=1}^w \sum_{j=1}^n \alpha_j^{(i)} x_j^{(i)}$. Since the local datasets are disjoint we simplify $\frac{1}{w} \sum_{i=1}^w \sum_{j=1}^n \alpha_j^{(i)} x_j^{(i)} = \frac{1}{w} \sum_{j=1}^{|\mho|} \alpha_j x_j$ for the combined local datasets $\mho = \bigcup_{i=1}^w D^{(i)}$ and a flattened $\alpha = \begin{bmatrix} \alpha_1^{(1)} & \dots & \alpha_n^{(1)} & \alpha_1^{(2)} & \dots & \alpha_n^{(w)} \end{bmatrix}$. A model which is represented by $\frac{1}{w} \sum_{j=1}^{|\mho|} \alpha_j x_j$ has the support vectors $V = \{ x_j \in \mho \mid \alpha_j \ne 0 \} = \bigcup_{i=1}^w V^{(i)}$, as the support vector characteristic is uniquely determined by $\alpha$ and each local $\alpha_j^{(i)}$ is element of $\alpha$ and responsible for the same data point. $\qquad\square$

### J.2.3 PROOF OF THM. 4.3 (AVERAGING LOCALLY TRAINED SVMS CONVERGES)

We recall Thm. 4.3:

**Theorem 4.3** (Averaging locally trained SVMs converges to a global SVM)**.** *Given a configuration $\zeta$ as in Tbl. 1 and the same local data sizes $\forall i, i' \colon n^{(i)} = n^{(i')}$, there exists a regularization parameter $\Lambda$ such that the average of locally trained models $avg(T)$ with a hinge-loss linear SVM as an objective function $\mathcal{J}$ trained with projected subgradient descent using weighted averaging, $T = HingeSVM\text{-}SGDWA$, converges with the number of local iterations $M$ to the best model for the combined local datasets $\mho$, i.e. $\mathbb{E}[\mathcal{J}(avg(HingeSVM\text{-}SGDWA), \mho) - \inf_f \mathcal{J}(f, \mho)] \in \mathcal{O}(1/M)$.*

*Proof.* First (1), we show that there exists a regularization parameter $\Lambda$ for which the converged global model equals the average of the converged locally trained models: $T_\xi(\mho) = avg(T\_xi(D^{(i)}))$. Second (2), we show that both the global and the local models converge with rate $\mathcal{O}(1/M)$.

Note that we assume that each data point $x_j$ is structured as $[1, x_{j,1}, \ldots, x_{j,p}]$ to include the intercept. We also denote the flattened $\alpha^{(\text{avg\_loc})} = \begin{bmatrix} \alpha_1^{(1)} & \ldots & \alpha_n^{(1)} & \alpha_1^{(2)} & \ldots & \alpha_n^{(w)} \end{bmatrix}$ as the dual coefficients of the averaged local SVM and $\alpha^{(\text{glob})}$ as the dual coefficients of the global SVM.

(1) By Lem. 4.2 we know for the combined local datasets $\mho = \bigcup_{i=1}^w D^{(i)}$ that

$$\text{avg}(T_\xi(D^{(i)})) = \frac{1}{wN} \sum_{j=1}^{|\mho|} \alpha_j^{(\text{avg\_loc})} x_j = \frac{1}{|\mho|} \sum_{j=1}^{|\mho|} \alpha_j^{(\text{avg\_loc})} x_j.$$

Note that we assume a scaled parameter per local SVM: $T_\xi(D^{(i)}) = \frac{1}{n} \sum_{j=1}^n \alpha_j x_j$. Without this assumption, we would not average the local SVMs but instead compute their sum.

For the global model, we write by the representer theorem

$$T(\mho) = \frac{1}{|\mho|} \sum_{j=1}^{|\mho|} \alpha_j^{(\text{glob})} x_j.$$

Thus, by parameter comparison we have that $T_\xi(\mho) = \text{avg}(T_\xi(D^{(i)}))$ if $\forall j\colon \alpha_j^{(\text{glob})} = \alpha_j^{(\text{avg\_loc})}$. By the characteristic of a hinge-loss linear SVM, we know that any $\alpha_j$ has the value $\alpha_j = \Lambda y_j$ if a data point is a support vector inside the margin (Ma & Ng, 2020, Equation 28-30). Hence, $\forall j\colon \alpha_j^{(\text{glob})} = \alpha_j^{(\text{avg\_loc})}$ if the margin is large enough that for both SVMs all data points are inside the margin. Since the margin of a hinge-loss linear SVM is the inverse of the parameter norm, $\|T(\mho)\|^{-1}$, and the parameter norm gets smaller with an increased regularization parameter $\Lambda$ by the definition of the objective function $\frac{1}{n} \sum_{(x,y) \in D^{(i)}} \max(0, 1 - y \langle f, x \rangle) + \Lambda \langle f, f \rangle$, we derive that there exists a regularization parameter $\Lambda$ which is large enough s.t. all data points are within the margin.

(2) By Lacoste-Julien et al. (2012), we know that a hinge-loss linear SVM converges to the optima with rate $\mathcal{O}(M^{-1})$, if we use projected subgradient descent using weighted averaging (SGDWA) as an optimization algorithm, i.e.

$$\mathbb{E}\left[ \mathcal{J}(\text{avg}(\text{HingeSVM-SGDWA}(D^{(i)})), \mho) - \inf_f \mathcal{J}(f, \mho) \right] \in \mathcal{O}(1/M).$$

$\square$

## J.3 Additional Privacy Proofs

### J.3.1 Proof of Thm. J.3 (Learner $T$ with a Randomized Sensitivity is DP)

**Lemma J.2.** *Let $T_{priv} : (D, r, \kappa) \to U$ be a randomized mechanism on dataset $D$ with two independent randomnesses $r$ and $\kappa$ and universe $U$. We define $T_{priv}^r(D, \kappa) := T_{priv}(D, r, \kappa)$, i.e., $T_{priv}^r : (D, \kappa) \to U$. If $\forall r$, $T_{priv}^r$ is $(\varepsilon, \delta)$-DP, then $T_{priv}$ is $(\varepsilon, \delta)$-DP.*

*Proof.* Let $D, D'$ be neighboring datasets and $S \subseteq U$ be defined over some universe $U$ as required. Let $R$ denote a distribution of randomness $r$ which is independent of the data $D$ as $r$ and $D$ are separate inputs. We show that if $\forall r\colon \Pr_\kappa[T_{priv}^r(D, \kappa) \in S] \leq \exp(\varepsilon) \Pr_\kappa[T_{priv}^r(D', \kappa) \in S] + \delta$ then $\Pr_{r,\kappa}[T_{priv}(D, r, \kappa) \in S] \leq \exp(\varepsilon) \Pr_{r,\kappa}[T_{priv}(D', r, \kappa) \in S] + \delta$. The proof is similar to that of Wu et al. (2017, Lemma 5).

By the law of total probability, we have

$$\Pr_{r,\kappa}[T_{\text{priv}}(D, r, \kappa) \in S] = \sum_r \Pr[R = r] \Pr_\kappa[T_{\text{priv}}(D, r, \kappa) \in S \mid R = r]$$

$$= \sum_r \Pr[R = r] \Pr_\kappa[T^r_{\text{priv}}(D, \kappa) \in S]$$

$$\leq \sum_r \Pr[R = r] \left( \exp(\varepsilon) \Pr_\kappa[T^r_{\text{priv}}(D', \kappa) \in S] + \delta \right)$$

$$= \exp(\varepsilon) \sum_r \Pr[R = r] \Pr_\kappa[T_{\text{priv}}(D', r, \kappa) \in S \mid R = r] + \sum_r \Pr[R = r]\delta$$

$$= \exp(\varepsilon) \Pr_{r,\kappa}[T_{\text{priv}}(D', r, \kappa) \in S] + \delta.$$

$\square$

**Theorem J.3.** *Let $T_{priv} : (D, r) \mapsto T(D, r) + \kappa$ be an additive mechanism with a Gaussian randomness $\kappa \in \text{pdf}_{\mathcal{N}(0,\sigma^2)}$ and noise scale $\sigma$ where $T$ is a randomized mechanism with randomness $r$ and dataset $D$. $T$ has a randomized sensitivity $\max_{D,D'} \max_r \|T(D, r) - T(D', r)\| \leq s$ where $D, D'$ are 1-neighboring datasets. Then $T_{priv}$ is $(\varepsilon, \delta)$-DP.*

*Proof.* Let $R$ denote the distribution of randomness $r$ which by construction does not depend on data $D$ or randomness $\kappa$. We define $T^r(D) := T(D, r)$, i.e., $T^r : D \mapsto T(D, r)$. We make a case distinction over each $r \in R$:

For each $r \in R$, we have the mechanism $T^r_{\text{priv}} : D \mapsto T^r(D) + \kappa$ with a deterministic sensitivity $\max_{D,D'} \max_{r \in R} \|T(D, r) - T(D', r)\| = \max_{D,D'} \|T^r(D) - T^r(D')\|$, where $D, D'$ are 1-neighboring datasets. By construction, $T^r_{\text{priv}}$ is a Gaussian mechanism which is $(\varepsilon, \delta)$-DP by Lem. 2.3. By Lem. J.2, since $T^r_{\text{priv}}$ is $(\varepsilon, \delta)$-DP for all $r$, $T_{\text{priv}}$ is $(\varepsilon, \delta)$-DP. $\square$

The same holds if we use the Gaussian mechanism in the group privacy extension (cf. Lem. J.4) or in the distributed setting (cf. Lem. E.1. In each case, we divide the algorithm output by a constant factor *const* which scales both the deterministic and the randomized sensitivity by *const*:

$$\max_{D,D'} \max_{r \in R} \left\| \frac{T(D, r)}{const} - \frac{T(D', r)}{const} \right\| = \frac{1}{const} \max_{D,D'} \max_{r \in R} \|T(D, r) - T(D', r)\| = \frac{s_{\text{rand}}}{const}$$

$$\max_{D,D'} \left\| \frac{T^r(D)}{const} - \frac{T^r(D')}{const} \right\| = \frac{1}{const} \max_{D,D'} \|T^r(D) - T^r(D')\| = \frac{s_{\text{det}}}{const}.$$

### J.3.2 PROOF OF LEM. J.4 (GROUP PRIVACY REDUCTION OF A MULTIVARIATE GAUSSIAN)

**Lemma J.4.** *Let $\text{pdf}_{\mathcal{N}(A,B)}[x]$ denote the probability density function of the multivariate Gaussian distribution with location and scale parameters $A, B$ which is evaluated on an atomic event $x$. For any atomic event $x$, any covariance matrix $\Sigma$, any group size $k \in \mathbb{N}$, and any mean $\mu$, we get*

$$\frac{\text{pdf}_{\mathcal{N}(0,k^2\Sigma)}[x]}{\text{pdf}_{\mathcal{N}(\mu,k^2\Sigma)}[x]} = \frac{\text{pdf}_{\mathcal{N}(0,\Sigma)}[x/k]}{\text{pdf}_{\mathcal{N}(\mu/k,\Sigma)}[x/k]}.$$

*Proof.*

$$\frac{\text{pdf}_{\mathcal{N}(0,k^2\Sigma)}[x]}{\text{pdf}_{\mathcal{N}(\mu,k^2\Sigma)}[x]} = \frac{\frac{1}{det(2\pi k^2\Sigma)} \exp(-\frac{1}{2}x^T k^2 \Sigma^{-1} x)}{\frac{1}{det(2\pi k^2\Sigma)} \exp(-\frac{1}{2} \underbrace{(x-\mu)^T k^2 \Sigma^{-1}(x-\mu)}_{=x^T k^2\Sigma^{-1}x - \mu^T k^2\Sigma^{-1}x - x^T k^2\Sigma^{-1}\mu + \mu^T k^2\Sigma^{-1}\mu})}$$

$$= \exp(-\frac{1}{2}(-\mu^T k^2 \Sigma^{-1} x - x^T k^2 \Sigma^{-1} \mu + \mu^T k^2 \Sigma^{-1} \mu))$$

$$= \exp(-\frac{1}{2}k^2 \cdot (-\mu^T \Sigma^{-1} x - x^T \Sigma^{-1} \mu + \mu^T \Sigma^{-1} \mu))$$

for $\mu_1 := \mu/k$

$$= \exp(-\frac{1}{2} \cdot k(-\mu_1^T \Sigma^{-1} x - x^T \Sigma^{-1} \mu_1 + \mu_1^T \Sigma^{-1} \mu_1/k))$$

for $x_1 := x/k$

$$= \exp(-\frac{1}{2} \cdot (-\mu_1^T \Sigma^{-1} x_1 - x_1^T \Sigma^{-1} \mu_1 + \mu_1^T \Sigma^{-1} \mu_1))$$

$$= \exp(-\frac{1}{2} \cdot (-\mu_1^T \Sigma^{-1} x_1 - x_1^T \Sigma^{-1} \mu_1 + \mu_1^T \Sigma^{-1} \mu_1))$$

$$= \frac{\frac{1}{det(2\pi\Sigma)} \exp(-\frac{1}{2} x_1^T \Sigma^{-1} x_1)}{\frac{1}{det(2\pi\Sigma)} \exp(-\frac{1}{2}(x_1 - \mu_1)^T k^2 \Sigma^{-1}(x_1 - \mu_1))}$$

$$= \frac{\text{pdf}_{\mathcal{N}(0,\Sigma)}[x/k]}{\text{pdf}_{\mathcal{N}(\mu/k,\Sigma)}[x/k]}$$

$\square$

As the Gaussian distribution belongs to the location-scale family, Lem. J.4 directly implies that the $(\varepsilon, \delta)$-DP guarantees of using $\mathcal{N}(0, k^2 \Sigma)$ noise for sensitivity $k$ and using $\mathcal{N}(0, \Sigma)$ for sensitivity 1 are the same.

### J.3.3 PROOF OF LEM. J.5 (REPRESENTING A MULTIVARIATE GAUSSIAN AS UNIVARIATE ONES)

For completeness, we rephrase a proof that we first saw in Abadi et al. (2016) that argues that sometimes the multivariate Gauss mechanism can be reduced to the univariate Gauss mechanism.

**Lemma J.5.** *Let* $\text{pdf}_{\mathcal{N}(\mu, \text{diag}(\sigma^2))}$ *denote the probability density function of a multivariate* $(p \geq 1)$ *spherical Gaussian distribution with location and scale parameters* $\mu \in \mathbb{R}^p, \sigma \in \mathbb{R}^p_+$. *Let* $M_{gauss,p,q}$ *be the* $p$ *dimensional Gaussian mechanism* $D \mapsto q(D) + \mathcal{N}(0, \sigma^2 \cdot I_p)$ *for* $\sigma^2 > 0$ *of a function* $q : \mathcal{D} \to \mathbb{R}^p$, *where* $\mathcal{D}$ *is the set of datasets. Then, for any* $p \geq 1$, *if* $q$ *is* $s$*-sensitivity-bounded, then for any* $p \geq 1$, *there is another* $s$*-sensitivity-bounded function* $q' : \mathcal{D} \to \mathbb{R}$ *such that the following holds: for all* $\varepsilon \geq 0, \delta \in [0,1]$ *if* $M_{gauss,1,q'}$ *satisfies* $(\varepsilon, \delta)$*-DP, then* $M_{gauss,p,q}$ *satisfies* $(\varepsilon, \delta)$*-DP.*

*Proof.* First observe that for any $s$-sensitivity-bounded function $q''$, two adjacent inputs $D, D'$ (differing in one element) with $\|q''(D) - q''(D')\|_2 = s$ are worst-case inputs. As a spherical Gaussian distribution (covariance matrix $\Sigma = \sigma^2 \cdot I_{p \times n}$) is rotation invariant, there is a rotation such that the difference only occurs in one dimension and has length $s$. Hence, it suffices to analyze a univariate Gaussian distribution with sensitivity $s$. Hence, the privacy loss distribution of both mechanisms (for the worst-case inputs) is the same. As a result, for all $\varepsilon \geq 0, \delta \in [0,1]$ (i.e. the privacy profile is the same) if $(\varepsilon, \delta)$-DP holds for the univariate Gaussian mechanism it also holds for the multivariate Gaussian mechanism. $\square$

### J.4 SOFTMAXREG

### J.4.1 PROOF OF THM. J.6 (STRONG CONVEXITY OF SOFTMAXREG)

**Theorem J.6.** *Let* $\mathcal{J}$ *denote the objective function* $\mathcal{J}(f, D) := \frac{\Lambda}{2} \sum_{k=1}^K \langle f_k, f_k \rangle + \frac{1}{n} \sum_{(x,y) \in D} \mathcal{L}_{CE}(y, \langle f, x \rangle)$ *with the cross-entropy loss* $\mathcal{L}_{CE}(y, z) := -\sum_{k=1}^K y_k \log \frac{\exp z_k}{\sum_{j=1}^K \exp z_j}$ *and parameters* $f \in \mathbb{R}^{d+1,K}$, *dataset* $D$ *where* $(x, y) \in D$ *with data points* $x \in \mathbb{R}^{d+1}$ *structured as* $[1 \ \ x_1 \ \ \ldots \ \ x_d]$ *and labels* $y \in \{0, 1\}^K$, *number of classes* $K$, *and regularization parameter* $\Lambda$. $\mathcal{J}$ *is* $\Lambda$*-strongly convex.*

*Proof.* $\mathcal{J}$ is $\mu$-strongly convex if $\mathcal{J} - \frac{\mu}{2} \langle f, f \rangle$ is convex. In our case, with $\mu = \Lambda$, it remains to be shown show that the cross entropy loss $\mathcal{L}_{CE}(y, z)$ is convex since a linear layer like $\langle f, x \rangle$ represents an affine map which preserves convexity (Bertsekas, 2009).

It is known that the cross entropy loss is convex by a simple argumentation: If the Hessian is positive semidefinite $\nabla^2 \mathcal{L}_{\mathrm{CE}}(y,z) \succeq 0$ then $\mathcal{L}_{\mathrm{CE}}$ is convex. By the Gershgorin circle theorem, a symmetric diagonally dominant matrix is positive semi-definite if the diagonals are non-real.

Since the second derivative of the cross-entropy loss is $\frac{\partial^2}{\partial z_p \partial z_q} \mathcal{L}_{\mathrm{CE}} = s_p(1_{[p=q]} - s_q)$ for the softmax probabilities $s_p = \frac{\exp z_p}{\sum_{j=1}^{K} \exp z_j}$, we conclude that the diagonals are non-negative since $s_p(1 - s_p)$ for $0 \leq s_p \leq 1$ is always non-negative. The Hessian is diagonally dominant if for every row $p$ the absolute value of the diagonal entry is larger or equal to the sum of the absolute values of all other row entries. In our case, we have

$$\forall p\colon |s_p(1-s_q)| \geq \sum_{q=1,q\neq p}^{K} |s_p(-s_q)| \iff \forall p\colon (1-s_q) \geq \sum_{q=1,q\neq p}^{K} s_q \iff \forall p\colon (1-s_q) \geq (1-s_p)$$

$\square$

### J.4.2 PROOF OF THM. J.7 (LIPSCHITZNESS OF SOFTMAXREG)

**Theorem J.7.** *Let $\mathcal{J}$ denote the objective function $\mathcal{J}(f, D) := \frac{\Lambda}{2} \sum_{k=1}^{K} \langle f_k, f_k \rangle + \frac{1}{n} \sum_{(x,y)\in D} \mathcal{L}_{CE}(y, \langle f, x \rangle)$ with the cross-entropy loss $\mathcal{L}_{CE}(y,z) := -\sum_{k=1}^{K} y_k \log \frac{\exp z_k}{\sum_{j=1}^{K} \exp z_j}$ and parameters $f \in \mathbb{R}^{d+1,K}$, dataset $D$ where $(x,y) \in D$ with data points $x \in \mathbb{R}^{d+1}$ structured as $\begin{bmatrix} 1 & x_1 & \dots & x_d \end{bmatrix}$ and labels $y \in \{0,1\}^K$, number of classes $K$, and regularization parameter $\Lambda$. $\mathcal{J}$ is L-Lipschitz with $L = \Lambda R + \sqrt{2}c$ where $\|x\| \leq c$ and $\|f\| \leq R$.*

*Proof.* In the following, we abbreviate $d' := d + 1$, flatten $f \in \mathbb{R}^{d'K}$ and notate $z := (x, y)$.

The Lipschitz continuity is defined as:

$$\sup_{z\in D, f, f'} \frac{\|\mathcal{J}(f,z) - \mathcal{J}(f',z)\|}{\|f - f'\|} \leq L.$$

We first (1) show

$$\sup_{z\in D, f, f'} \frac{\|\mathcal{J}(f,z) - \mathcal{J}(f',z)\|}{\|f - f'\|} \leq \sup_{z\in D, f} \|\nabla_f \mathcal{J}(f,z)\|$$

using the mean value theorem and subsequently (2) bound $\sup_{z\in D, f} \|\nabla_f \mathcal{J}(f,z)\| \leq L$.

(1) Recall that the multivariate mean value theorem states that for some function $g\colon G \mapsto \mathbb{R}$ on an open subset $G \in \mathbb{R}^n$, some $x, y \in G$ and some $c \in [0,1]$, we have

$$g(y) - g(x) = \langle \nabla g((1-c)x + cy), y - x \rangle.$$

In our case, we write

$$\sup_{z\in D, f, f'} \frac{\|\mathcal{J}(f,z) - \mathcal{J}(f',z)\|}{\|f - f'\|}$$

by the multivariate mean value theorem for some $c \in [0,1]$

$$= \sup_{z\in D, f, f'} \frac{|\langle \nabla \mathcal{J}((1-c)f' - cf, z), f - f' \rangle|}{\|f - f'\|}$$

for $f'' := (1-c)f' - cf$ and by the Cauchy-Schwarz inequality $|\langle \nabla_{f''} \mathcal{J}(f'', z), f - f' \rangle| \leq \|\nabla_{f''} \mathcal{J}(f'', z)\| \cdot \|f - f'\|$

$$\leq \sup_{z\in D, f''} \|\nabla_{f''} \mathcal{J}(f'', z)\|.$$

(2) We know that for $1 \leq j \leq d', 1 \leq p \leq K$ the partial derivative of $\mathcal{J}$ is $\frac{\partial}{\partial f_p} \mathcal{J}(f, (x, y)) = \Lambda f_{lp} + x_l \cdot (s_p - 1_{[y=p]})$ with $s_p := \frac{\exp\langle f_p, x \rangle}{\sum_{j=1}^K \exp\langle f_j, x \rangle}$. Thus, we have

$$\|\nabla_f \mathcal{J}(f, z)\| = \sqrt{\sum_{lp=1}^{d'K} \left( \Lambda f_{lp} + x_l(s_p - 1_{[y=p]}) \right)^2}$$

$$= \sqrt{\sum_{lp=1}^{d'K} \left( \Lambda^2 f_{lp}^2 + 2\Lambda f_{lp} x_l(s_p - 1_{[y=p]}) + x_l^2(s_p - 1_{[y=p]})^2 \right)}$$

$$= \sqrt{\Lambda^2 \|f\|^2 + 2\Lambda \sum_{l=1}^{d'} x_l \sum_{p=1}^{K} f_{lp}(s_p - 1_{[y=p]}) + \sum_{l=1}^{d'} x_l^2 \sum_{p=1}^{K} (s_p - 1_{[y=p]})^2}$$

due to the Cauchy-Schwarz inequality, we have $\sum_{p=1}^K f_{lp}(s_p - 1_{[y=p]}) \leq \sqrt{\sum_{p=1}^K f_{lp}^2} \sqrt{\sum_{p=1}^K (s_p - 1_{[y=p]})^2}$ and $\sum_{l=1}^{d'} x_l \sqrt{\sum_{p=1}^K f_{lp}^2} \leq \sqrt{\sum_{l=1}^{d'} x_l^2} \sqrt{\sum_{lp=1}^{d'K} f_{lp}^2} = \|x\|^2 \|f\|^2$

$$\leq \sqrt{\Lambda^2 \|f\|^2 + 2\Lambda \|x\| \|f\| \sqrt{\sum_{p=1}^{K} (s_p - 1_{[y=p]})^2} + (\sum_{l=1}^{d'} x_l^2)(\sum_{p=1}^{K} (s_p - 1_{[y=p]})^2)}$$

since $\max_{s_1, \ldots, s_K} \left\{ (s_p - 1)^2 + \sum_{q=1, q \neq p}^K s_q^2 \mid \sum_{k=1}^K s_k = 1 \wedge \forall k \colon s_k \geq 0 \right\} = 2$ with $s_q = 1 \wedge s_p = 0 \bigwedge_{k=1, k \neq q}^K s_k = 0$ where $q \neq p$

$$\leq \sqrt{\Lambda^2 \|f\|^2 + 2\sqrt{2}\Lambda \|x\| \|f\| + 2 \|x\|^2} = \Lambda \|f\| + \sqrt{2} \|x\|$$

Thus, with $\|x\| \leq c, \|f\| \leq R$ we conclude that

$$\sup_{z \in D, f, f'} \frac{\|\mathcal{J}(f, z) - \mathcal{J}(f', z)\|}{\|f - f'\|} \leq \sup_{z \in D, f} \|\nabla_f \mathcal{J}(f, z)\| \leq \Lambda R + \sqrt{2}c = L$$

$\square$

### J.4.3 Proof of Thm. J.8 (Smoothness of SoftmaxReg)

**Theorem J.8.** *Let $\mathcal{J}$ denote the objective function $\mathcal{J}(f, D) := \frac{\Lambda}{2} \sum_{k=1}^K \langle f_k, f_k \rangle + \frac{1}{n} \sum_{(x,y) \in D} \mathcal{L}_{CE}(y, \langle f, x \rangle)$ with the cross-entropy loss $\mathcal{L}_{CE}(y, z) := -\sum_{k=1}^K y_k \log \frac{\exp z_k}{\sum_{j=1}^K \exp z_j}$ and parameters $f \in \mathbb{R}^{d+1, K}$, dataset $D$ where $(x, y) \in D$ with data points $x \in \mathbb{R}^{d+1}$ structured as $[1 \quad x_1 \quad \ldots \quad x_d]$ and labels $y \in \{0, 1\}^K$, number of classes $K$, and regularization parameter $\Lambda$. $\mathcal{J}$ is $\beta$-smooth with $\beta = \sqrt{(d+1)K\Lambda^2 + 0.5(\Lambda + c^2)^2}$ where $\|x\| \leq c$.*

*Proof.* In the following, we abbreviate $d' := d + 1$, flatten $f \in \mathbb{R}^{d'K}$ and notate $z := (x, y)$.

$\beta$-Smoothness is defined as:

$$\sup_{z \in D, f, f'} \frac{\left\| \nabla_f \mathcal{J}(f, z) - \nabla_{f'} \mathcal{J}(f', z) \right\|}{\|f - f'\|} \leq \beta.$$

We first (1) show

$$\sup_{z \in D, f, f'} \frac{\left\| \nabla_f \mathcal{J}(f, z) - \nabla_{f'} \mathcal{J}(f', z) \right\|}{\|f - f'\|} \leq \sup_{z \in D, f} \|\mathbf{H}_f(\mathcal{J}(f, z))\|$$

using the mean value theorem and subsequently (2) bound $\sup_{z \in D, f} \|\mathbf{H}_f(\mathcal{J}(f, z))\| \leq \beta$.

(1) Recall that the multivariate mean value theorem states that for some function $g \colon G \mapsto \mathbb{R}$ on an open subset $G \in \mathbb{R}^n$, some $x, y \in G$ and some $c \in [0, 1]$, we have

$$g(y) - g(x) = \langle \nabla g((1-c)x + cy), y - x \rangle.$$

In our case, we write

$$\sup_{z \in D, f, f'} \frac{\left\| \nabla_f \mathcal{J}(f, z) - \nabla_{f'} \mathcal{J}(f', z) \right\|}{\| f - f' \|}$$

$$= \sup_{z \in D, f, f'} \frac{\sqrt{\sum_{i=0}^{d'K} (\nabla_{f_i} \mathcal{J}(f, z) - \nabla_{f_i'} \mathcal{J}(f', z))^2}}{\| f - f' \|}$$

by the multivariate mean value theorem for some $c \in [0, 1]$ and $g_i(f, z) \coloneqq \nabla_{f_i} \mathcal{J}(f, z)$

$$= \sup_{z \in D, f, f'} \frac{\sqrt{\sum_{i=0}^{d'K} \langle \nabla g_i((1-c)f' - cf, z), f - f' \rangle^2}}{\| f - f' \|}$$

for $f'' \coloneqq (1-c)f' - cf$ and by the Cauchy-Schwarz inequality $|\langle \nabla g_i(f'', z), f - f' \rangle|^2 \le \| \nabla g_i(f'', z) \|^2 \cdot \| f - f' \|^2$

$$\le \sup_{z \in D, f''} \sqrt{\sum_{i=0}^{d'K} \sum_{j=0}^{d'K} (\nabla^2_{f_i'', f_j''} \mathcal{J}(f'', z))^2}$$

$$= \sup_{z \in D, f} \| \mathbf{H}_f(\mathcal{J}(f, z)) \|.$$

(2) We know that with $1 \le l \le d', 1 \le p \le K$ the first-order partial derivative of $\mathcal{J}$ is $\frac{\partial}{\partial f_{lp}} \mathcal{J}(f, (x, y)) = \Lambda f_{lp} + x_l \cdot (s_p - 1_{[y=p]})$ with $s_p \coloneqq \frac{\exp\langle f_p, x \rangle}{\sum_{i=1}^K \exp\langle f_i, x \rangle}$.

With $1 \le j \le d', 1 \le q \le K$ we know that the second-order partial derivative of $\mathcal{J}$ is $\frac{\partial^2}{\partial f_{lp} \partial f_{jq}} \mathcal{J}(f, (x, y)) = 1_{[lp=jq]} \cdot \Lambda + x_l \cdot x_j \cdot s_p(1_{[p=q]} - s_q)$. Thus, we have

$$\| \mathbf{H}_f(\mathcal{J}(f, z)) \| = \sqrt{\sum_{lp=1}^{d'K} \sum_{jq=1}^{d'K} \left( 1_{[lp=jq]} \cdot \Lambda + x_l x_j s_p(1_{[p=q]} - s_q) \right)^2}$$

$$= \sqrt{\sum_{lp=1}^{d'K} \left( (\Lambda + x_l^2 s_p(1-s_p))^2 + \sum_{\substack{jq=1 \\ j \ne l}}^{d'K} x_l^2 x_j^2 s_p^2 (1-s_p)^2 + \sum_{\substack{jq=1 \\ j \ne l \\ q \ne p}}^{d'K} x_l^2 x_j^2 s_p^2 s_q^2 \right)}$$

$$= \sqrt{\sum_{lp=1}^{d'K} \left( (\Lambda + x_l^2 s_p(1-s_p))^2 + x_l^2 s_p^2 \sum_{\substack{j=1 \\ j \ne l}}^{d'} \left( x_j^2 (1-s_p)^2 + x_j^2 \sum_{\substack{q=1 \\ q \ne p}}^{K} s_q^2 \right) \right)}$$

since we have $\max_{s_1, \dots, s_K} \left\{ \sum_{q=1, q \ne p}^{K} s_q^2 \mid \sum_{q=1, q \ne p}^{K} s_q = 1 - s_p \wedge \forall i : s_i \ge 0 \right\} = (1 - s_p)^2$ due to the maximal L2-distance given a bounded L1-distance is the maximal L2-distance in one dimension, we conclude

$$= \sqrt{\sum_{lp=1}^{d'K} \left( \Lambda^2 + 2\Lambda x_l^2 s_p(1-s_p) + x_l^4 s_p^2 (1-s_p)^2 + 2x_l^2 s_p^2 (1-s_p)^2 \sum_{\substack{j=1 \\ j \ne l}}^{d'} x_j^2 \right)}$$

$$\le \sqrt{d'K\Lambda^2 + \sum_{lp=1}^{d'K} x_l^2 s_p(1-s_p) \left( 2\Lambda + 2x_l^2 s_p(1-s_p) + 2s_p(1-s_p) \sum_{\substack{j=1 \\ j \ne l}}^{d'} x_j^2 \right)}$$

$$= \sqrt{d'K\Lambda^2 + 2 \sum_{lp=1}^{d'K} x_l^2 s_p(1-s_p) \left( \Lambda + s_p(1-s_p) \| x \|^2 \right)}$$

$$\le \sqrt{d'K\Lambda^2 + 2 \| x \|^2 \sum_{l=1}^{d'} x_l^2 \sum_{p=1}^{K} s_p(1-s_p)(\Lambda \| x \|^{-2} + s_p)}$$

following Lem. J.9 (presented and shown below) we simplify with $C := \Lambda \|x\|^{-2}$: $\sum_{p=1}^{K} s_p (1 - s_p)(C + s_p) \leq 0.25(C + 1)^2$

$$\leq \sqrt{d'K\Lambda^2 + 0.5 \|x\|^4 (\Lambda \|x\|^{-2} + 1)^2} = \sqrt{d'K\Lambda^2 + 0.5(\Lambda + \|x\|^2)^2}.$$

Thus, with $\|x\| \leq c$ we conclude that

$$\sup_{z \in D, f, f'} \frac{\|\nabla_f \mathcal{J}(f, z) - \nabla_{f'} \mathcal{J}(f', z)\|}{\|f - f'\|} \leq \sup_{z \in D, f} \|\mathbf{H}_f(\mathcal{J}(f, z))\| \leq \sqrt{d'K\Lambda^2 + 0.5(\Lambda + c^2)^2} = \beta.$$

$\square$

**Lemma J.9.** *Let $\{ s_p \}_{p=1}^{K}$ denote probabilities such that $\sum_{p=1}^{K} s_p = 1$, and $C \in \mathbb{R}_+$ a constant, then we have*

$$\max_{\{s_p\}_{p=1}^{K}} \left\{ \sum_{p=1}^{K} s_p (1 - s_p)(C + s_p) \mid \sum_{p=1}^{K} s_p = 1 \land \forall p \colon s_p \geq 0 \right\}$$

$$\leq 0.25(C + 1)^2$$

*with $\forall p \in \cup_{i=1}^{k} P_i, p' \in \cup_{i=k+1}^{K} P_i, P \in \mathrm{Sym}(K) \colon (s_p = \frac{1}{k} \land s_{p'} = 0)$, i.e. for some arbitrary but fixed dimensions $k : 1 \leq k \leq K$, the solution has $k$-times $s_p = \frac{1}{k}$ and $(K - k)$-times $s_p = 0$.*

*Proof.* We show this Lemma as follows: First, we use the Karush–Kuhn–Tucker (KKT) conditions to find the $s_p$'s which maximize the maximization term. Thereby, we obtain a set of four solution candidates where we encode all $s_p$'s in closed form and introduce two new variables $k, j$ which serve as a solution counter. Second, we insert the solution candidates into the maximization term and show that the result is always bounded by $0.25(C + 1)^2$ by calculating the optimal front across all possible values of the solution counters $k, j$.

Let $f(s) := \sum_{p=1}^{K} s_p (1 - s_p)(C + s_p)$ denote the function to maximize, $h(s) := \sum_{p=1}^{K} s_p - 1$ the equality constraint, and $\forall p \colon g_p(s) := -s_p$ the inequality constraints. To find the constrained maximum, we maximize the Lagrangian function $\mathcal{L}_{agrange}(s) = f(s) + \mu_p g_p(s) + \lambda h(s)$ with $\mu_p, \lambda$ as slack variables. This suffices since $s_p$ does not have unbounded border cases: the only valid configuration of all $s_p$'s is on a hyperplane ($\sum_p s_p = 1$) bounded in all dimensions ($s_p \geq 0$). Using the slack variable $\mu_p$, we already cover whether its corresponding $s_p$ is on the border ($\mu_p > 0$) or not ($\mu = 0$). Following the KKT conditions, the following conditions have to hold for the maximum:

(1) Stationarity: $\forall p \colon \nabla_{s_p} \mathcal{L}_{agrange}(s) = C + 2s_p - 2Cs_p - 3s_p^2 + \mu_p - \lambda = 0$

(2) Primal feasibility: $\forall p \colon h(s) = 0$ and $g_p(s) \leq 0$

(3) Dual feasibility: $\forall p \colon \mu_p \geq 0$

(4) Complementary slackness: $\forall p \colon \mu_p g_p(s) = 0$

Informally, it suffices for the solution of the KKT conditions to analyze the cases where $\forall p, 1 \leq p \leq k \colon s_p > 0$ for all fixed number of dimensions $k : 1 \leq k \leq K$ since if $s_p = 0$ then we have already proved the same result for one less dimension.

Formally and without loss of generality[7], we show for all fixed numbers of dimensions $k : 1 \leq k \leq K$ that for the solution of the KKT conditions it suffices to analyze the cases where $\forall p, 1 \leq p \leq k \colon s_p > 0$. For the induction base case ($k = 1$ dimensional), we have $s_1 > 0$ and thus by condition (4) $\mu_1 = 0$. If and only if $s_1 = 1$, we satisfy conditions (2) and (1) with $\lambda = -C - 1$. With $s_1 = 0$ we would not be able to satisfy the equality constraint of condition (2), i.e. '$s_1 = 1$'.

For the $k \mapsto k+1$ induction case, we know that $\forall p, 1 \leq p \leq k \colon s_p > 0$. If $s_{p+1} > 0$, by the induction hypothesis we know that $\forall p, 1 \leq p \leq k + 1 \colon s_p = 0$. If $s_{p+1} = 0$ then by conditions (3) and (4) we have $\mu_{p+1} > 0$ and thus by condition (1), $\mu_{p+1} = \lambda - C$. Inserting $s_{p+1} = 0, \mu_{p+1} = \lambda - C$ into

---

[7]The same argumentation holds for situations where the dimensions are permuted.

conditions (1) to (4), we obtain the same set of equations and inequalities as for the $k$-dimensional case which already holds by the induction hypothesis.

We solve the KKT conditions (1) to (4) as follows: First, we solve the system of equations of condition (1) for $s_p$ via the quadratic formula:

$$s_p^{\pm} = \frac{-(2-2C)\pm\sqrt{(2-2C)^2-4(-3)(C-\lambda)}}{2(-3)} = \frac{1}{3} \cdot \left(\pm\sqrt{C^2 + C - 3\lambda + 1} - C + 1\right). \qquad (1)$$

Second, we plug $s_p^{\pm}$ into the equality constraint '$h(s) = 0$' of condition (2) and solve for $\lambda$ which gives us for some solution counter $j \in \mathbb{N}, 0 \leq j \leq k$ with $2j \neq k$:

$$h(s^{\pm}) = 0$$
$$\iff \left(\textstyle\sum_{i=1}^{j} s_i^+\right) + \left(\textstyle\sum_{i=j+1}^{k} s_i^-\right) = 1$$
$$\iff j\left(\sqrt{C^2 + C - 3\lambda + 1} - C + 1\right) + (k-j)\left(-\sqrt{C^2 + C - 3\lambda + 1} - C + 1\right) = 3$$
$$\iff (2j-k)\sqrt{C^2 + C - 3\lambda + 1} = Ck - k + 3$$
$$\Rightarrow C^2 + C - 3\lambda + 1 = \frac{(Ck-k+3)^2}{(2j-k)^2}$$
$$\iff \lambda = \frac{(2j-k)^2(C^2+C+1)-(Ck-k+3)^2}{3(2j-k)^2}.$$

The solution counter $j$ quantifies how often we plug the 'positive' variant of $s_p^{\pm}$ into $h(s^{\pm})$:

$$s^{\pm} := \begin{bmatrix} s_1^+ & \cdots & s_j^+ & s_{j+1}^- & \cdots & s_k^- \end{bmatrix}$$

or any permutation of the dimensions of $s^{\pm}$.

Note that at $2j = k$, we have a special case and by the equality constraint '$h(s) = 0$' of condition (2)

$$h(s^{\pm}) = 0 \land 2j = k \iff \left(\textstyle\sum_{p=1}^{\frac{k}{2}} s_p^+\right)+\left(\textstyle\sum_{p=\frac{k}{2}+1}^{k} s_p^-\right) = 1 \iff k(1-C) = 3 \iff C = \frac{k-3}{k}.$$

Thus, at $2j = k, C = \frac{k-3}{k}$ we simplify the solution in Eq. (1) to

$$s_p^{\pm,\, C=k\,-\,3/k} = \frac{1}{3} \cdot (\pm \underbrace{\sqrt{\tfrac{(k-3)^2}{k^2} + \tfrac{k-3}{k} - 3\lambda + 1}}_{=:\, 3Q} - \tfrac{k-3}{k} + 1) = \pm Q + \tfrac{1}{k}.$$

If we now insert $s_p^{\pm,\, C=k\,-\,3/k}$ into $f(\cdot)$ and maximize for all remaining variables, we find the maximum at

$$\max_{k,j,\lambda} \left\{ f(s_p^{\pm,\, C=k\,-\,3/k}) \mid 2j = k \land C = \tfrac{k-3}{k} \land s_p^{\pm,\, C=k\,-\,3/k} \geq 0 \right\}$$
$$\leq \max_{k,j,\lambda} \left\{ \textstyle\sum_{p=1}^{\frac{k}{2}} (\tfrac{1}{k} + Q)(1 - (\tfrac{1}{k} + Q))(C + (\tfrac{1}{k} + Q)) \right.$$
$$\left. + \textstyle\sum_{p=\frac{k}{2}+1}^{k} (\tfrac{1}{k} - Q)(1 - (\tfrac{1}{k} - Q))(C + (\tfrac{1}{k} - Q)) \mid 2j = k \land C = \tfrac{k-3}{k} \right\}$$
$$= \max_{k} \left\{ \tfrac{k}{2}\tfrac{1}{k}(1 - \tfrac{1}{k})(\tfrac{k-3}{k} + \tfrac{1}{k}) + \tfrac{k}{2}\tfrac{1}{k}(1 - \tfrac{1}{k})(\tfrac{k-3}{k} + \tfrac{1}{k}) \right\}$$
$$= \max_{k} \left\{ (1 - \tfrac{1}{k})(\tfrac{k-3}{k} + \tfrac{1}{k}) \right\} = \max_{k} \left\{ \tfrac{k-2}{k} - \tfrac{k-2}{k^2} \right\} = \max_{k} \left\{ \underbrace{1 - \tfrac{3}{k} + \tfrac{2}{k^2}}_{\leq\, 0.25(\frac{k-3}{k}+1)^2\, =\, 1-\frac{3}{k}+\frac{9}{4k^2}} \right\}.$$

Thus, at $2j = k$, $\mathcal{L}_{\text{agrange}}$ is maximal at $C = \frac{k-3}{k}$ which is always strictly below the maximum we will show in this lemma if $C = \frac{k-3}{k}$. In the following, we continue the proof for $2j \neq k$.

Third, by plugging $\lambda$ into Eq. (1) which is derived from the system of equations in condition (1) and solving for $s_p$ we obtain the following two solution candidates for $2j \neq k$

$$s_p^{(+,-)} = \tfrac{1}{3}\left(\pm\sqrt{C^2 + C - 3\tfrac{(2j-k)^2(C^2+C+1)-(Ck-k+3)^2}{3(2j-k)^2} + 1} - C + 1\right)$$
$$= \tfrac{1}{3}\left(1 - C \pm \tfrac{Ck-k+3}{2j-k}\right) = \tfrac{(2j-k)(1-C)\pm(Ck-k+3)}{3(2j-k)} = \tfrac{-2Cj+Ck+2j-k\pm(Ck-k+3)}{6j-3k}$$
$$s_p^{(+)} = \tfrac{-2(k-j)C+2(k-j)-3}{6(k-j)-3k}, s_p^{(-)} = \tfrac{-2jC+2j-3}{6j-3k}.$$

Observe that if we replace $\tilde{j} := k - j$ in $s_p^{(+)}$ we get $s_p^{(-)}$ with $\tilde{j}$ instead of $j$. To abbreviate, we write

$$s_p^{(j')} = \frac{-2j'C + 2j' - 3}{6j' - 3k}$$

for $j' \in \{j, k-j\}$. Because of the similar structure of $s_p^{(j)}$ and $s_p^{(k-j)}$, restricting $j$ by $0 \le 2j < k$ suffices since we would otherwise count the same maximum twice. With $s_p^{(j')}$ as our solution candidate, the equality constraint '$h(s) = 0$' in condition (2) holds when we have $(k - j)$ times $s_p^{(j)}$ and $j$ times $s_p^{(k-j)}$:

$$s^{sol} := \begin{bmatrix} s_1^{(k-j)} & \dots & s_j^{(k-j)} & s_{j+1}^{(j)} & \dots & s_k^{(j)} \end{bmatrix}$$

or any permutations of the dimensions of $s^{sol}$. This goes by construction of $s^{\pm}$ where the solution counter $j$ quantifies how often we plug in $s_p^{(+)}$ into $h(s^{(+,-)})$.

We next compute the second partial derivative test to determine for which parameters the solution candidate $s^{sol}$ is a local maximum or minimum: We have a maximum if the Hessian of $\mathcal{L}_{agrange}$ is positive definite and a minimum if the Hessian of $\mathcal{L}_{agrange}$ is negative definite. In our case, the second partial derivatives of $\mathcal{L}_{agrange}$ are $\nabla_{s_p}^2 \mathcal{L}_{agrange}(s) = 2 - 2C - 6s$ and $\nabla_{s_p}\nabla_{s_q}\mathcal{L}_{agrange}(s) = 0$ with $p \ne q$. Thus, we have a diagonal Hessian matrix. Hence, if $2 - 2C - 6s^{sol} < \mathbf{0}$ we have a maximum and if $2 - 2C - 6s^{sol} > \mathbf{0}$ we have a minimum. Because of the second partial derivative test, we also know that if the Hessian has both positive and negative eigenvalues then we have a saddle point. This holds in our case when we have both positive and negative values on the diagonals of the Hessian, i.e. for some $p$ we have $2 - 2C - 6s_p^{sol} < 0$ and for some $q$ we have $2 - 2C - 6s_q^{sol} > 0$. Furthermore, if we have a zero eigenvalue this test is indecisive.

We rearrange the maximum condition for any entry of $s^{sol}$ (here: $s_p^{(j')}$) as follows:

$$2 - 2C - 6\frac{-2j'C + 2j' - 3}{6j' - 3k} < 0 \iff \begin{cases} kC - k + 3 > 0 & \text{if } 0 \le 2j' < k \\ kC - k + 3 < 0 & \text{if } 2j' > k \end{cases}$$

$$\iff \begin{cases} C > \frac{k-3}{k} & \text{if } 0 \le 2j' < k \\ C < \frac{k-3}{k} & \text{if } 2j' > k \end{cases}.$$

Similarly, we rearrange the minimum condition, such that

$$2 - 2C - 6\frac{-2j'C + 2j' - 3}{6j' - 3k} > 0 \iff \begin{cases} C < \frac{k-3}{k} & \text{if } 0 \le 2j' < k \\ C > \frac{k-3}{k} & \text{if } 2j' > k \end{cases}.$$

Recall that at this point we only consider $2j \ne k$. We now distinguish three cases for the second partial derivative test for the vector $s^{sol}$: $C < \frac{k-3}{k}, C > \frac{k-3}{k}, C = \frac{k-3}{k}$.

At $C < \frac{k-3}{k}$, we write

$$\begin{bmatrix} 2 - 2C - 6s_1^{(k-j)} < 0 \\ \dots \\ 2 - 2C - 6s_j^{(k-j)} < 0 \\ 2 - 2C - 6s_{j+1}^{(j)} > 0 \\ \dots \\ 2 - 2C - 6s_k^{(j)} > 0 \end{bmatrix}$$

and at $C > \frac{k-3}{k}$, we write similarly

$$\begin{bmatrix} 2 - 2C - 6s_1^{(k-j)} > 0 \\ \dots \\ 2 - 2C - 6s_j^{(k-j)} > 0 \\ 2 - 2C - 6s_{j+1}^{(j)} < 0 \\ \dots \\ 2 - 2C - 6s_k^{(j)} < 0 \end{bmatrix}.$$

Recall the saddle point criteria as $\exists_p \exists_q\, 2 - 2C - 6s_p^{sol} < 0 \wedge 2 - 2C - 6s_q^{sol} > 0$ and the maximum criteria as $2 - 2C - 6s^{sol} < \mathbf{0}$. By the above test criteria, for $C \neq \frac{k-3}{k}$, we have a saddle point for all $j \in [1, k-1]$ as well as a maximum for $j = k \wedge C < \frac{k-3}{k}$ and for $j = 0 \wedge C > \frac{k-3}{k}$ at

$$s^{max} := \begin{bmatrix} s_1^{(k-k)} & \dots & s_k^{(k-k)} \end{bmatrix} = \begin{bmatrix} s_1^{(0)} & \dots & s_k^{(0)} \end{bmatrix} = \begin{bmatrix} 1/k & \dots & 1/k \end{bmatrix} \wedge C \neq \frac{k-3}{k}$$

since only at $j \in \{0, k\}$ do we have the case that either $s_p^{(j)}$ or $s_p^{(k-j)}$ is present in the solution $s^{sol}$.

At $C = \frac{k-3}{k}$, we have for any entry of $s^{sol}$ (here: $s_p^{(j')}$)

$$s_p^{(j', C = k-3/k)} = \frac{-2(k-3)j'/k + 2j' - 3}{6j' - 3k} = \frac{6j'/k - 3}{6j' - 3k} = \frac{1}{k}.$$

Thus, although the second partial derivative test is indecisive since $2 - 2C - 6 1/k = 0$, we have at $C = \frac{k-3}{k}$ always the same solution as in $s^{max}$. This renders $s^{max}$ for all $C$ as the maximal solution.

Next, we plug the solution $s^{max}$ into $f(s)$ and calculate the optimal front with the inequality constraint '$g_p(s) \leq 0$' of condition (2) and across all number of dimensions $k$ and range of the solution counter $j \in \{0, k\}$:

$$\max_{k, j} \left\{ f(s^{max}) \mid s_p^{(j)} \geq 0 \wedge s_p^{(k-j)} \geq 0 \wedge j \in \{0, k\} \right\}$$

$$= \max_k \left\{ \sum_{p=1}^{k} s_p^{(0)}(1 - s_p^{(0)})(C + s_p^{(0)}) \mid s_p^{(0)} \geq 0 \right\}$$

$$= \max_k \left\{ \sum_{p=1}^{k} \tfrac{1}{k}(1 - \tfrac{1}{k})(C + \tfrac{1}{k}) \mid \tfrac{1}{k} \geq 0 \right\}$$

$$= \max_k \left\{ C + \tfrac{1-C}{k} - \tfrac{1}{k^2} \right\}$$

$\Big($ for $k = \frac{2}{1-C}$ the term $C + \frac{1-C}{k} - \frac{1}{k^2}$ is maximal for which we need the derivative to be zero: $\frac{d}{dk}(C + \frac{1-C}{k} - \frac{1}{k^2}) = \frac{C-1}{k^2} + \frac{2}{k^3} = 0 \Big)$

$$= C + \tfrac{1}{2}(1 - C)^2 - \tfrac{1}{4}(1 - C)^2$$

$$= \tfrac{C^2}{4} + \tfrac{C}{2} + \tfrac{1}{4} = 0.25(C + 1)^2$$

Thus, we conclude that $f(s^{max})$ is equal to or below the convex hull $0.25(C + 1)^2$ for any solution counter $j$ and any number of classes $k$.

$\square$

Note: In this proof, we assumed $k \in \mathbb{R}_+$, however, we can restrict the number of classes $k$ even further: $k \in \mathbb{N}$ and $k \leq K$. Yet, this restriction does not have much impact on the bound on $f$ for a reasonable $C, K$: Now, we only have $K$ possible maxima ($s_p^{max} = \{1, 1/2, \dots, 1/K\}$) where for a given $C$ only one of these maxima are dominant. This also means that our $0.25(C+1)^2$-bound is a convex hull and only matches the maxima in a few selected points. However, already for little $K$ does the maximum come considerably close to the hull as shown in Fig. 11.

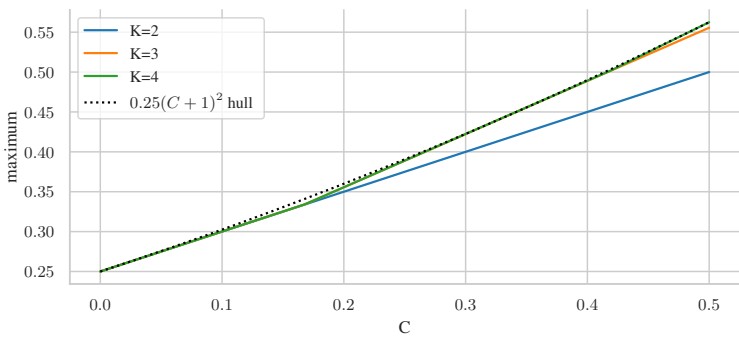

Figure 11: Precise maximum of $f(s^{max})$ per constant $C$ and restricted, discretized number of classes $k \leq K, k \in \mathbb{N}$ versus convex hull of the maximum of $f(s^{max})$ across all number of classes $k \in \mathbb{R}_+$.

