# OpenReview forum: "Private Blind Model Averaging – Distributed, Non-interactive, and Convergent"
_ICLR.cc/2025/Conference — Submitted to ICLR 2025_

### Official Review · Reviewer_fS8P · 2024-10-19

**Soundness:** 2
**Presentation:** 2
**Contribution:** 2
**Rating:** 3
**Confidence:** 4

**Summary:**

This papers addresses the problem of privacy protection in blind model averaging for convex and smooth empirical risk minimization. The performance of blind model averaging is discussed in problems of SVM and Softmax regression. The privacy approach is differential privacy.

**Strengths:**

The discussed problem of privacy protection is interesting and important.

**Weaknesses:**

1. The results are limited to convex and smooth objective functions. Given that many machine learning problems are essentially nonconvex and nonsmooth, it would be important to relax this assumption.

2. The applicability of the approach also seems limited. The paper only discusses SVM and Softmax regression applications. It will be interesting to know if the approach also applies to other applications, particularly deep learning applications.

3. The design of differential privacy seems straightforward. It is unclear if there are any challenges.

**Questions:**

see above.

---

> ### Author Response · Authors · 2024-11-22
> **Rebuttal**
>
> We thank the reviewers for their insightful comments. Below, we will respond to some specific comments and refer to the shared comment above for a summary of our changes in an updated paper version.
>
> ## Limited Applicability
> > The results are limited to convex and smooth objective functions. Given that many machine learning problems are essentially nonconvex and nonsmooth, it would be important to relax this assumption.
> > The applicability of the approach also seems limited. The paper only discusses SVM and Softmax regression applications. It will be interesting to know if the approach also applies to other applications, particularly deep learning applications.
>
> Blind Averaging is designed for convex models which -- as described in Section 6 (previously Section 7) -- achieve over 83% ImageNet performance, thanks to pre-training. It is as of yet unclear whether large-scale neural networks can be trained from the ground up with meaningful utility and meaningful differential privacy guarantees. We strongly believe that the paradigm of pre-training + convex model is a very promising direction for differentially private learning.
>
> Technically, you can apply our work to non-linear neural networks, yet these have no provably data-independent Lipschitzness, smoothness, strong convexity, and sensitivity which makes output perturbation with little noise impossible and a path to strong utility bounds is unclear: if a local model has multiple (local) optima, averaging these most probably won't result in strong utility.
>
> ## Challenges
>
> > The design of differential privacy seems straightforward. It is unclear if there are any challenges.
>
> We solved one major privacy challenge described in Section 3 (previously in Section 4): how a softmax regression can be output private. Without such a bound on the sensitivity, differential privacy does not hold. Due to the length of the proof we postponed most of it in the appendix but emphasized in the main body how the proof works and what has to be shown.
>
> Apart from that, we also provide a non-interactive variant of SecAgg (Section 5, "Non-interactive protocol"). We show how secure summation impacts the privacy bound (Theorem 5.1). We show how privacy amplification by averaging works and provides privacy (Lemma E.1 and E.2, similarly in Jayaraman et al. (2018)). We give a user-level sensitivity of BlindAvg (Corollary 5.2). We show how a learner with a randomized sensitivity as used in Wu et al. (2017) can still provide DP and is compatible with many other DP building blocks as most DP proofs assume a deterministic sensitivity (Lemma J.2 and Theorem J.3).

---

> > ### Comment · Reviewer_fS8P · 2024-11-22
> >
> > I read the response and retain my opinion.

---

### Official Review · Reviewer_bBFP · 2024-10-30

**Soundness:** 2
**Presentation:** 1
**Contribution:** 2
**Rating:** 3
**Confidence:** 3

**Summary:**

This paper presents a method called Blind Model Averaging (BlindAvg) to improve the scalability and privacy of distributed machine learning, particularly for differentially private settings. Unlike traditional gradient-averaging methods, BlindAvg allows each client to train a model independently and submit it for secure averaging without any online synchronization. This approach is shown to work effectively for convex, smooth empirical risk minimization (ERM) tasks like Support Vector Machines (SVM) and Softmax regression.

**Strengths:**

Scalability and Efficiency: By proposing a non-interactive learning framework, the paper addresses the high communication and synchronization costs that typically come with FL.

Enhanced Privacy Guarantees: The BlindAvg approach integrates DP at both the data point and user levels,

**Weaknesses:**

1) Limited Applicability to Non-Convex Models: The theoretical convergence guarantees provided in the paper apply primarily to convex models like SVMs and smooth ERM-based models. The lack of results for non-convex models, such as deep neural networks, restricts BlindAvg’s applicability in modern machine learning.

2) **Fatal Presentation Problem**: The paper suffers from a significant presentation issue, as numerous variables are introduced and used without clear, standalone definitions, which considerably hinders readability. For example, variables such as $d$ (dimensionality of the input) and $c$ (input clipping bound) in line #240 are used without clear definitions, making it challenging to understand their roles in the algorithms and theoretical analysis. Beyond these examples, many other variables/names/functions/algorithms lack clarity, compounding the difficulty. This lack of clarity in variable presentation is one of the worst I’ve encountered, severely impacting the paper’s accessibility.


Presentation:
1) You should define all variables before use in your algorithm. Variable $x$ was not defined in algorithm 1. You only define $D={(x_j, y_j)}$ in the algorithm 1. If you want to loop through every $x_j$, you must specify the loop in the algorithm. Otherwise, it is ambiguous and unclear.
2) Algorithm 2. "run the client code of secure summation..." and "Run the server protocol of $\pi$" are unclear illustrations. I can see your illustration in #378-#381 and appendix D.5. However, who sent what information to whom? Who did what calculation on which variables? e.g. you can write $F(n^(i)/w\cdot f_{priv}_1, n^(i)/w\cdot f_{priv}_1)$ for secure summation. and "send $f$ to server", "receive $f$ from client". Even though you want to hide the detail of secure summation in the appendix, a certain level of abstraction should be provided in the algorithm, instead of a plain unclear illustration.
3) Figure 5. why there is "?" after "distributed"? I can only recognize it as a typo. Please use clear expressions in the figure, instead of indication. Besides, in this figure your global model and SVM-SGD both use a black solid line, which is confusing. Please be professional.

Minor:
1) Gauss mechanism -> **Gaussian** mechanism.
2) #224, for m **in** 1...M.

**Questions:**

N/A

---

> ### Author Response · Authors · 2024-11-22
> **Rebuttal (I)**
>
> We thank the reviewers for their insightful comments. Below, we will respond to some specific comments and refer to the shared comment above for a summary of our changes in an updated paper version.
>
> ## Limited Applicability
> > 1. Limited Applicability to Non-Convex Models: The theoretical convergence guarantees provided in the paper apply primarily to convex models like SVMs and smooth ERM-based models. The lack of results for non-convex models, such as deep neural networks, restricts BlindAvg’s applicability in modern machine learning.
>
> Blind Averaging is designed for convex models which -- as described in Section 6 (previously Section 7) -- achieve over 83% ImageNet performance, thanks to pre-training. It is as of yet unclear whether large-scale neural networks can be trained from the ground up with meaningful utility and meaningful differential privacy guarantees. We strongly believe that the paradigm of pre-training + convex model is a very promising direction for differentially private learning.
>
> Technically, you can apply our work to non-linear neural networks, yet these have no provably data-independent Lipschitzness, smoothness, strong convexity, and sensitivity which makes output perturbation with little noise impossible and a path to strong utility bounds is unclear: if a local model has multiple (local) optima, averaging these most probably won't result in strong utility.

---

> ### Author Response · Authors · 2024-11-22
> **Rebuttal (II)**
>
> ## Presentation
>
> > 2. Fatal Presentation Problem: The paper suffers from a significant presentation issue, as numerous variables are introduced and used without clear, standalone definitions, which considerably hinders readability. For example, variables such as $d$ (dimensionality of the input) and $c$ (input clipping bound) in line #240 are used without clear definitions, making it challenging to understand their roles in the algorithms and theoretical analysis. Beyond these examples, many other variables/names/functions/algorithms lack clarity, compounding the difficulty. This lack of clarity in variable presentation is one of the worst I’ve encountered, severely impacting the paper’s accessibility.
>
> Thanks for your honest feedback!
> In our revised paper we refactored our configuration definition (formally Def. 3.1) to a table together with other important notation (including the dimensionality and input clipping bound). We hope that this adds more clarity to our notation.
>
> In l240, we did not introduce $c$ as it is described two lines above. In Algorithm 1 and 3 as well as below Theorem 2.1 (previously within Theorem 3.1) we explicitly note how we clip using the clipping bound $c$.
>
>
> > 1. You should define all variables before use in your algorithm. Variable $x$ was not defined in algorithm 1. You only define $D=(x_j, y_j))$ in the algorithm 1. If you want to loop through every $x_j$, you must specify the loop in the algorithm. Otherwise, it is ambiguous and unclear.
>
> Thanks for pointing that out. We made it clear in Algorithms 1 and 3 that we loop through every data point $x_j$.
>
> > Algorithm 2. "run the client code of secure summation..." and "Run the server protocol of $\pi$" are unclear illustrations. I can see your illustration in #378-#381 and appendix D.5. However, who sent what information to whom? Who did what calculation on which variables? e.g. you can write $F(n(i)/w\cdot f_{priv}1, n(i)/w\cdot f{priv}_1)$ for secure summation. and "send $f$ to server", "receive $f$ from client". Even though you want to hide the detail of secure summation in the appendix, a certain level of abstraction should be provided in the algorithm, instead of a plain unclear illustration.
>
> All clients work together with the server to perform the MPC. The details of who sends which message to whom depend on the MPC protocol in question. As BlindAvg works independently of the concrete MPC protocol, we abstracted from a concrete MPC protocol. In our case, we extend SecAgg (Bell et al., 2020) with computation servers (cf. Section 6, "Non-interactive protocol"). Here, each user sends a share of their model to each of the computation servers. The computation servers then perform one secure summation step.
>
> We have clarified in Section 5 (previously Section 6), "Non-interactive protocol", that we can use this specific SecAgg extension as $\pi_{\text{SecSum}}$ in Algorithm 2.
>
> > Figure 5. why there is "?" after "distributed"? I can only recognize it as a typo. Please use clear expressions in the figure, instead of indication. Besides, in this figure your global model and SVM-SGD both use a black solid line, which is confusing. Please be professional.
>
> We polished Figure 5. For all Subfigures, black (global model) and green lines (averaged model) always use the same learner. Here, the line style or caption indicates which learner is used.
>
> SVM-SGD and HingeSVM have the same line style (solid) across the subfigures because they are very similar with two distinctions: 1) SVM-SGD has an upper bound on the learning rate $\frac{1}{\beta}$. Due to our knowledge that does not impact convergence. 2) SVM-SGD uses a smooth relaxation of the hinge loss: the Huber loss. The Huber loss uses the relaxation parameter $h$ to describe how close it resembles the hinge loss, for $h=0$ both are the same. In our experiments, we used $h=0.1$.

---

> > ### Comment · Reviewer_bBFP · 2024-11-24
> > **Thank you.**
> >
> > Thank you for your comment. It appears that a major revision has been made to the submission. I have decided to keep my original score.

---

### Official Review · Reviewer_n1Ay · 2024-10-31

**Soundness:** 2
**Presentation:** 2
**Contribution:** 2
**Rating:** 3
**Confidence:** 3

**Summary:**

This paper proposes and analyzes a privacy-preserving model averaging technique based on output perturbation. Privacy guarantees are established for Support Vector Machines and Softmax Regression. Empirical results are also presented.

**Strengths:**

The proposed method requires only one round of model aggregation, due to the use of output perturbation. The privacy analysis is provided and seems to be correct.

**Weaknesses:**

(1) The proposed method has limited applicability. For complex models, the sensitivity of model parameters is often unbounded.

(2) The advantages of using output perturbation over other private optimization techniques, such as DP-SGD or objective perturbation, for optimizing local models are not clear to me.

(3) Finally, the paper's organization and notation are somewhat disorganized.
> For instance
> - 1. In the Contributions section, the order of Phase 1 and Phase 2 is reversed.
> - 2. Some notations are unclear. For example, on line 343, the meaning of "_" is ambiguous. Additionally, the definition of the inner product between functions on line 330 should be formally stated.
> - 3. Definition D.8 is not well-articulated
> - 4. What is the meaning of ADP in Lemma K.6?

**Questions:**

(1) If the primary contribution is reducing communication time, could the authors elaborate on the rationale for using output perturbation to privately train local models? How does output perturbation compare to other methods, such as objective perturbation or NoisySGD, for training local models?

(2) Can the proposed method be applied to a broader range of models, beyond SVMs and Softmax classifiers?

---

> ### Author Response · Authors · 2024-11-22
> **Rebuttal**
>
> We thank the reviewers for their insightful comments. Below, we will respond to some specific comments and refer to the shared comment above for a summary of our changes in an updated paper version.
>
> ## Limited Applicability
>
> > (1) The proposed method has limited applicability. For complex models, the sensitivity of model parameters is often unbounded.
> > (2) Can the proposed method be applied to a broader range of models, beyond SVMs and Softmax classifiers?
>
> Blind Averaging is designed for convex models which -- as described in Section 6 (previously Section 7) -- achieve over 83% ImageNet performance, thanks to pre-training. It is as of yet unclear whether large-scale neural networks can be trained from the ground up with meaningful utility and meaningful differential privacy guarantees. We strongly believe that the paradigm of pre-training + convex model is a very promising direction for differentially private learning.
>
> Technically, you can apply our work to non-linear neural networks, yet these have no provably data-independent Lipschitzness, smoothness, strong convexity, and sensitivity which makes output perturbation with little noise impossible and a path to strong utility bounds is unclear: if a local model has multiple (local) optima, averaging these most probably won't result in strong utility.
>
> ## Output perturbation vs gradient vs objective perturbation
>
> > (2) The advantages of using output perturbation over other private optimization techniques, such as DP-SGD or objective perturbation, for optimizing local models are not clear to me.
> > (1) If the primary contribution is reducing communication time, could the authors elaborate on the rationale for using output perturbation to privately train local models? How does output perturbation compare to other methods, such as objective perturbation or NoisySGD, for training local models?
>
> We agree that generally gradient perturbation performs slightly better than objective perturbation which performs slightly better than output perturbation (McKenna et al., 2021). Yet, in a distributed setting, all but output perturbation have a significant disadvantage: they are highly interactive and require a large communication overhead. We emphasize this in Table 3 (previously Table 1), our related work in Section 7 (previously Section 2), and our extended related work in Appendix C.
>
> The reason is as follows: To protect all intermediary updates (objective function or gradients) in a distributed setting, the training has to be performed in expensive protocols like MPC or homomorphic encryption (or noise would have to be added at every step, like in DP-FL). Other gradient perturbation methods that leak intermediary gradient Bassily et al. (2014); Abadi et al. (2016); Yu et al. (2021) need one MPC invocation per iteration. In contrast, we only need one synchronization update of the trained noisy model.
>
> Moreover, objective perturbation also does not perform well with the number of classes $K$ as it is still open whether it works with tight composition bounds in $\mathcal{O}(\sqrt{K})$ (cf. "AMP" in Figure 9 of Appendix G).
>
> [McKenna et al., 2021]  A Practitioners Guide to Differentially Private Convex Optimization, By Ryan McKenna, Hristo Paskov, Kunal Talwar. In: TPDP 2021.
>
> ## Organization / Notation
>
> > 1. In the Contributions section, the order of Phase 1 and Phase 2 is reversed.
>
> That's a good point. We decided to reverse the order in the contribution section for a concise top-down description of BlindAvg: we motivate the technical concepts in Phase I by the convergence we show in Phase II. For ease of readability, we first talk about where we want to go, and then how we get there. If you have strong reservations about this, we are open to changing the order.
>
>
> > 2. Some notations are unclear. For example, on line 343, the meaning of "\_" is ambiguous. Additionally, the definition of the inner product between functions on line 330 should be formally stated.
>
> We refactored the "\_" notation out of our paper. It was intended to notate that this parameter is not relevant.
> In l330 and everywhere else in the paper, we do not use an inner product between functions. $f$ notates the SVM parameter. Our paper now emphasizes more frequently that $f$ notates the model parameter.
>
>
> > 3. Definition D.8 is not well-articulated
>
> This is a standard definition of strong convexity, which we are very familiar with. We have articulated the meaning of the terms to make it easier to read directly above Definition D.8.
>
> > 4. What is the meaning of ADP in Lemma K.6?
>
> We refactored our paper to use a consistent notation. By ADP we mean approximate differential privacy, i.e. the main differential privacy notion denoted by $(\varepsilon,\delta$)-DP. This is in contrast to PDP (probabilistic differential privacy), which also has $\varepsilon$ and $\delta$.

---

> ### Comment · Reviewer_n1Ay · 2024-11-22
> **Reply to rebuttal**
>
> Thanks for your detailed answers.
>
> > "It is as of yet unclear whether large-scale neural networks can be trained from the ground up with meaningful utility and meaningful differential privacy guarantees. We strongly believe that the paradigm of pre-training + convex model is a very promising direction for differentially private learning."
>
> Pre-trained features can enhance the accuracy of DP learning, even when the model is non-convex, as demonstrated in [1] and [2]. In your paper, the feature representations are got from a pre-trained model (SimCLR), yet the model is constrained to being convex with a strongly convex regularizer. This limitation suggests a narrower scope for the proposed approach.
>
> [1] De, Soham, et al. "Unlocking high-accuracy differentially private image classification through scale." arXiv preprint arXiv:2204.13650 (2022).
>
> [2] Yu, Da, et al. "Differentially private fine-tuning of language models." arXiv preprint arXiv:2110.06500 (2021).
>
> > "The reason is as follows: To protect all intermediary updates (objective function or gradients) in a distributed setting ... In contrast, we only need one synchronization update of the trained noisy model."
>
> My question is that: what if first train each local model using DP-(S)GD, then aggregating model parameters across all local users? I think this approach also only requires one synchronization update. And it would be interesting to see empirical comparisions.
>
> > Regarding definition D.8 "This is a standard definition of strong convexity, which we are very familiar with. We have articulated the meaning of the terms to make it easier to read directly above Definition D.8."
>
> My question was regarding the $\sup_{f, f^\prime}$ term on the left-hand side of this inequality. Specifically, I find it unclear how a bivariate supremum ($\sup_{f, f^\prime}$) applies to a univariate function ($q(f)$). Furthermore, since the arbitrariness of $f$ and $f^\prime$ is already assumed in the definition, are you perhaps aiming to convey something beyond strong convexity by introducing $\sup_{f, f^\prime}$?

---

> > ### Author Response · Authors · 2024-11-25
> > **Author Reply**
> >
> > Thanks for the follow-up questions and clarifications!
> >
> > ## Limited Applicability
> >
> > > Pre-trained features can enhance the accuracy of DP learning, even when the model is non-convex, as demonstrated in [1] and [2]. In your paper, the feature representations are got from a pre-trained model (SimCLR), yet the model is constrained to being convex with a strongly convex regularizer. This limitation suggests a narrower scope for the proposed approach.
> >
> > Thank you for pointing out the references [1,2]. We agree that in [1] multi-layer fine-tuning is mostly better than last-layer fine-tuning (albeit for smaller $\varepsilon$ this advantage seems to shrink). Yet, with better-performing pertaining methods, we do not think that there is a fundamental reason why last-layer fine-tuning should always be inferior to more-layer fine-tuning. Especially with more layers, the noise that DP adds will also increase, leading to a natural shift to small-parameter fine-tuning like last-layer fine-tuning or LoRA [2]. In particular [1] used a regular (supervised) model and not specific pre-training networks like SimCLR: the authors of SimCLR also chose a linear prediction head thus their model is designed to work well with a convex prediction head. In Figure 10 (Appendix H), we illustrate the embedding space of SimCLR on CIFAR-10 indicating clearly separated classes and thus most likely a good linear separability in the original 6144-dimensional space.
> >
> > ## Output perturbation vs gradient perturbation
> >
> > > My question is that: what if first train each local model using DP-(S)GD, then aggregating model parameters across all local users? I think this approach also only requires one synchronization update. And it would be interesting to see empirical comparisons.
> >
> > This approach would require similar to DP-FL $\sqrt{w}$ more noise for $w$ many users while having a worse accuracy than DP-FL (which we already evaluate) as the models are trained separable and gradients are not synchronized. The reason is as follows: For DP-SGD-based training, we do not have an output sensitivity, thus we can not apply privacy amplification by averaging on the models themselves. The noise in DP-SGD is added to the gradients and not the output: whether we use MPC for the averaging or not, we do have to release each gradient update per user. In a centralized setting of DP-SGD, we would only release one gradient update across all users which leads for averaged DP-SGD trained models to $w$ more total gradients released and by sequential composition a $\sqrt w$ worse $\varepsilon$.
> >
> > ## Organization / Notation
> >
> > > My question was regarding the $\sup_{f,f'}$ term on the left-hand side of this inequality. Specifically, I find it unclear how a bivariate supremum ($\sup_{f,f'}$) applies to a univariate function ($q(f)$). Furthermore, since the arbitrariness of $f$ and $f'$ is already assumed in the definition, are you perhaps aiming to convey something beyond strong convexity by introducing $\sup_{f,f'}$?
> >
> > Good point, we fixed that in our paper! The supremum was redundant: as you pointed out, we already assumed the arbitrariness of $f$ and $f'$ in the definition.

---

> > > ### Comment · Reviewer_n1Ay · 2024-11-26
> > > **Reply to authors**
> > >
> > > Thanks for answering my question. As other reviewers pointed out, it seems a made major revisions has been made to the submission. Thus I will keep my score.

---

### Official Review · Reviewer_8mnr · 2024-11-04

**Soundness:** 3
**Presentation:** 2
**Contribution:** 3
**Rating:** 8
**Confidence:** 3

**Summary:**

The paper tackles the reduction of communication and synchronization overhead in distributed, private learning.
Focusing on differentially private (DP) empirical risk minimization for convex and smooth functions, the authors analyze a private blind model averaging technique with provable utility.
Blind model averaging is a non-interactive technique where each user only sends a single message, and a minimal amount of communication is therefore used.
The paper provides bounds and empirical results for SVMs and softmax regressions.
Supported by their theoretical findings, the authors observe that blindly averaged softmax regression outperforms blindly averaged SVMs when dealing with a large number of classes, while blindly averaged SVMs outperform gradient averaging (such as federated learning) when dealing with a large number of users.
The authors therefore conclude that, under their assumptions, blind model averaging can provide a good privacy-utility trade-off.

**Strengths:**

The paper tackles a relevant problem, as distributed learning is essential in privacy-preserving ML.
The overall separation of the paper's content and structure in two "phases", that is, the individual training and the averaging in the models, is conveniently presented and helps readability.
The proof of differential privacy for softmax aggregation trained with SGD follows from the proof that the objective function is strongly convex, smooth, and Lipschitz.
While details were not checked, the derivation seems sound and convincing.
The experimental evaluation supports the theoretical findings and is reasonably extensive.
I think this is overall a good manuscript which, if improved in presentation/form (see weaknesses), can be a very well rounded contribution .

**Weaknesses:**

**Structure**

While the derivation of the results is quite streamlined, the many aspects of the organization of the paper should be improved.
Specifically, the introduction presents the overview of the method using symbols which have not been defined yet.
While I understand the usefulness of describing the approach early on, I feel like this may be confusing.
On a similar note, the "main result" is presented in Figure 2, in the introduction.
Similarly, a summary of the comparison of the proposed approach against related work is part of Section 2 (related work).
In this case as well, I feel like moving the discussion of results to the experiments section would benefit readability.
I think that moving the related work section to the bottom of the manuscript could also be a possibility.
The subdivision of the work in several named paragraphs (that is, paragraphs which begin with bold text) does not help readability, in my opinion.
Maybe using subsections for this could help to visually and conceptually separate the different parts of the contribution.
Bold text is also used in figures where, I feel, it is not necessary to highlight several lines of text.


**Presentation of empirical results**

Following up on my previous comment, empirical results are difficult to parse.
Firstly, the plots are very small with very tiny labels, across the whole manuscript.
Secondly, the experimental results section discusses 4 research questions which have, however, not clearly been introduced before.


**Imprecise language**

The terminology used is at times imprecise (e.g., "spread" the dataset in line 103, of "perfectly converge" in line 481), or not introduced (e.g., "local DP" and "group privacy" are never introduced as concepts).
Local DP and group privacy should, in my opinion, definitely be at least informally introduced in the main body.
The results in figure 6 are, in this sense, pretty much impossible to parse by referencing the main text only.
In fact (see for instance Section 6), the concepts "local DP" and "privacy with local aggregator" are not compatible but can be easily confused.


**Limitations and discussion**

I think the limitations and discussion section should be improved.
While I understand that page limits may be tricky, I feel like the limitations of the contribution are unclear from the main body of the text and at least a few lines should be dedicated to it.
A more thorough discussion should also mention possibile improvements and future work directions.



**Minor points**

* line 27 "we also conclude the first" -> "we also derive", maybe?
* line 146 (and others) "noised" -> "noisy"
* lines 74 and 81, are "phase I" and "phase II" inverted?

**Questions:**

* Can you make the connection between group-DP and local DP clearer (i.e., expand on the discussion for figure 6)?

---

> ### Author Response · Authors · 2024-11-22
> **Rebuttal**
>
> We thank the reviewers for their insightful comments. Below, we will respond to some specific comments and refer to the shared comment above for a summary of our changes in an updated paper version.
>
> ## Structure
>
> > Specifically, the introduction presents the overview of the method using symbols which have not been defined yet. While I understand the usefulness of describing the approach early on, I feel like this may be confusing.
>
> Thanks, we now introduce the symbols $\Lambda$, $\beta$, and $L$ in our introduction.
>
> > On a similar note, the "main result" is presented in Figure 2, in the introduction. Similarly, a summary of the comparison of the proposed approach against related work is part of Section 2 (related work). In this case as well, I feel like moving the discussion of results to the experiments section would benefit readability. I think that moving the related work section to the bottom of the manuscript could also be a possibility.
>
> Great idea, we have moved the related work section to the end.
>
> We believe that showing our main result directly with the third contribution gives a graphical intuition of our achievements.
>
> > The subdivision of the work in several named paragraphs (that is, paragraphs which begin with bold text) does not help readability, in my opinion. Maybe using subsections for this could help to visually and conceptually separate the different parts of the contribution.
>
> That's a valid point. We felt compelled to use this style to condense our material. We are open to changing this for the camera-ready version if you find it hinders readability; in that case, we will cut some material or move it to the appendix (suggestions welcome).
>
> > Bold text is also used in figures where, I feel, it is not necessary to highlight several lines of text.
>
> Thanks, we fixed that.
>
> ## Imprecise language
> > The terminology used is at times imprecise (e.g., "spread" the dataset in line 103, of "perfectly converge" in line 481), or not introduced (e.g., "local DP" and "group privacy" are never introduced as concepts). Local DP and group privacy should, in my opinion, definitely be at least informally introduced in the main body. The results in figure 6 are, in this sense, pretty much impossible to parse by referencing the main text only. In fact (see for instance Section 6), the concepts "local DP" and "privacy with local aggregator" are not compatible but can be easily confused.
>
> Thanks, we fixed all points here.
>
> ## Limitations and discussion
>
> > I think the limitations and discussion section should be improved. While I understand that page limits may be tricky, I feel like the limitations of the contribution are unclear from the main body of the text and at least a few lines should be dedicated to it. A more thorough discussion should also mention possibile improvements and future work directions.
>
> Good point! We rewrote the limitations and discussion section and have added a comment on possible future work.
>
> ## Group DP / local DP
>
> > Can you make the connection between group-DP and local DP clearer (i.e., expand on the discussion for figure 6)?
>
> Good point! We refactored the caption of Figure 6 and the motivation for user-level privacy in the paragraph before Corollary 5.2 (previously Corollary 6.2). By user-level privacy, we mean group DP which protects all data points of a user.

---

> > ### Comment · Reviewer_8mnr · 2024-11-26
> >
> > Thank you very much for your rebuttal and for addressing my questions. I think the significant revisions do improve upon readability which was, personally, my main concern. I will keep my score as it is.

---

### Official Review · Reviewer_gYEj · 2024-11-04

**Soundness:** 1
**Presentation:** 1
**Contribution:** 1
**Rating:** 1
**Confidence:** 3

**Summary:**

This paper considers *blind model averaging* for differentially private federated learning. Here, each "user" holds data for many individuals, whose privacy we wish to protect. Each user trains a model locally with privacy. These models are averaged to produce a final model.

**Strengths:**

This is a topic of practical and theoretical interest. I am not deeply acquainted with this literature, but to the best of my knowledge the questions considered in this submission are not fully answered by prior work.

**Weaknesses:**

I found this paper hard to read and do not believe it meets the necessary level of rigor.

I was often confused by the organization and informal discussion. Here are a couple issues I observed.
- Figure 1 lists "Phase I" as "local ERM training" for SVM or softmaxReg and "Phase II" as the averaging, but Section 4 is "PHASE I: DIFFERENTIALLY PRIVATE SOFTMAXREG" and doesn't include any analysis of SVMs. Then Section 5, titled "PHASE II: NON-INTERACTIVE BLIND MODEL AVERAGING (BLINDAVG)", only appears to address SVMs. Then Section 6 is "SYSTEM DESIGN OF BLINDAVG", which I would think also describes Phase II.
- Some crucial concepts appear undefined, like "honest" users or the version of SGD used.
- Definition 3.1 is where we define the "configuration," ie learning setting, but it has a number of strange features. It introduces the notation $I$ for the identity matrix and $K_{\mathrm{comp}}$ for "the number of compositions," but neither are used in the notation $\zeta(\cdots)$ that denotes the configuration. "$i$" is used, but that confuses me, since it appears to just index into the set $\lbrace 1,\ldots, w\rbrace$. "$n$" also appears but is undefined.
- Theorem 3.2 mentions parameters $c$ and $R$ in the hypothesis but not in the conclusion; these are only tied to $\Lambda, L,$ and $\beta$ later in the text.

On a technical level, I will focus on Theorem 5.2, which concerns the convergence of an average of SVMs, each trained on a subset of the data, to the "global" model trained on all the data. Here are a few issues around this.
1. The theorem and proof don't address the noise for privacy. I don't understand how these results relate to claims about the privately trained model.
1. The proof establishes that, for a sufficiently large $\ell_2$ penalty, the average (over the best local models) equals the best global model (ie, over all data). It's not clear why this is interesting: we can obtain an approximate version of this claim "at a glance" by noting that sufficiently high regularization will push all weights to zero. It's not clear why we should care about such a regime.
2. The theorem mentions "projected subgradient descent using weighted averaging," but it doesn't show up in the algorithms and is only mentioned, but not described, in the proof. Algorithm 3 uses projected SGD but doesn't specify what that entails.
4. I'm confused by what it means to take the average of the models, even in Definition 3.1: I don't understand the "type" of $T$, so it's not clear why $\sum_i T(D^{(i)})$ is well-defined. In the proof of Corollary K.2 it looks like we sum models in the dual space, but Algs 2 and 3 look to me like we're working in the weight space. I think this distinction matters when we talk about adding noise.
5. The theorem relies on Lemma 5.1, which (in its proof) assumes that the datasets are disjoint, i.e., no two data points are repeated across users. I don't see why this is a justified assumption, as differential privacy needs to hold for worst-case datasets.
1. Two sections in the appendix are titled "Proof of Thm 5.2", J.5.2 and K.5.2, only the latter actually contains a proof.

This submission would need substantial revisions before I would be comfortable accepting it.

**Questions:**

No questions.

---

> ### Author Response · Authors · 2024-11-22
> **Rebuttal (I)**
>
> We thank the reviewers for their insightful comments. Below, we will respond to some specific comments and refer to the shared comment above for a summary of our changes in an updated paper version.
>
> > This submission would need substantial revisions before I would be comfortable accepting it.
>
> Thank you for the harsh (but fair) examination of our work. We believe that we can counter your criticism of Theorem 5.2 and we revised the presentation to a point that it might convince you.
>
> ## Presentation
>
> > Figure 1 lists "Phase I" as "local ERM training" for SVM or softmaxReg and "Phase II" as the averaging, but Section 4 is "PHASE I: DIFFERENTIALLY PRIVATE SOFTMAXREG" and doesn't include any analysis of SVMs. Then Section 5, titled "PHASE II: NON-INTERACTIVE BLIND MODEL AVERAGING (BLINDAVG)", only appears to address SVMs. Then Section 6 is "SYSTEM DESIGN OF BLINDAVG", which I would think also describes Phase II.
>
> As Phase I for SVMs is related work, we decided to not include it in Section 3 (SoftmaxReg, previously Section 4). We can see now that this made for a more confusing read, especially since we contribute a novel analysis of SVMs in Phase II in Section 4 (previously Section 5).
>
> You're right that Section 5 (System Design, previously Section 6) belongs semantically to Phase II. We have changed the section title to reflect this.
> We further updated Figure 1 to better reflect the organization of our paper and our contributions.
>
> > Some crucial concepts appear undefined, like "honest" users [...]
>
> Thanks for the feedback! Our threat model now clearly defines honest users (Section 5 "System Design").
>
> > [...] or the version of SGD used.
>
> We updated Algorithms 1 and 3 to make the SGD algorithm explicit.
>
>
> > Definition 3.1 is where we define the "configuration," ie learning setting, but it has a number of strange features. It introduces the notation $I$ for the identity matrix and $K_{comp}$ for "the number of compositions," but neither are used in the notation $\zeta(\dots)$ that denotes the configuration. "$i$" is used, but that confuses me, since it appears to just index into the set $\{1,\dots,w\}$. "$n$" also appears but is undefined.
>
> In the revised paper version, we refactored our configuration definition to a table (cf. Table 1) and added other important notation as well. We hope that this adds more clarity to our notation.
>
> > Theorem 3.2 mentions parameters $c$ and $R$ in the hypothesis but not in the conclusion; these are only tied to $\Lambda$, $L$, and $\beta$ later in the text.
>
> We agree that we used more assumptions than necessary in Theorem 3.2 (now Theorem 2.1) and thus refactored our paper accordingly. In particular, $c$ and $R$ are recommended to reach a finite data-independent Lipschitzness $L$, yet a Lipschitzness $L$ suffices for this Theorem to hold. We note that each different ERM has a different Lipschitzness $L$ and smoothness $\beta$ as seen in logistic regression (cf. Example 2.6), SVMs (cf. Example 2.5), and SoftmaxReg (cf. Theorem 3.1).
>
> ## Theorem 5.2
>
> > 1. The theorem and proof don't address the noise for privacy. I don't understand how these results relate to claims about the privately trained model.
>
> The theorem and proof consider the case without noise.
> With differential privacy, BlindAvg does not converge. The error introduced by Gaussian noise $\mathcal{N}(0,\tilde\sigma)$ on a trained model $f$ amounts to $E[\lvert(f + \mathcal{N}(0,\tilde\sigma)) - f\rvert] =  E[\lvert \mathcal{N}(0,\tilde\sigma) \rvert] = \frac{\tilde\sigma \sqrt2}{\sqrt\pi} \approx 0.8\tilde\sigma$ since we add noise once after finishing training.
>
> We had not included the statement as we considered it less interesting, but have now added a statement for completeness.
>
> > 2. The proof establishes that, for a sufficiently large $\ell_2$ penalty, the average (over the best local models) equals the best global model (ie, over all data). It's not clear why this is interesting: we can obtain an approximate version of this claim "at a glance" by noting that sufficiently high regularization will push all weights to zero. It's not clear why we should care about such a regime.
>
> Thanks for highlighting this point. Our proof actually shows more than this trivial case: the blind average of ERMs has the union of the local dual coefficients as dual coefficients $\alpha_j$. Thus, we can describe the error introduced by blind averaging with the difference in the averaged and global dual coefficients. Previously this insight was hidden in our Corollary in the Appendix, we have now highlighted this in the main body as Corollary 4.1.
>
> We also show that for non-zero weights, we still converge. The reason is that $\Lambda$ has to be sufficiently high but since the inputs are bounded by $c$ due to DP, we will always reach a finite $\Lambda$ that fulfills our requirements.

---

> ### Author Response · Authors · 2024-11-22
> **Rebuttal (II)**
>
> > 3. The theorem mentions "projected subgradient descent using weighted averaging," but it doesn't show up in the algorithms and is only mentioned, but not described, in the proof. Algorithm 3 uses projected SGD but doesn't specify what that entails.
>
>
> In short: for technical reasons, we had to use a slightly different loss function for our utility analysis (Section 4) and our privacy proof (Section 5). For the utility analysis, we use the non-smooth hinge loss but for the privacy analysis, we have to use a smooth loss. Therefore we use the Huber loss, which approximates the hinge loss with relaxation parameter $h$. For $h=0$ the Huber loss equals the hinge loss. In our experiments, we used $h=0.1$.
>
> In more detail, on why we use projected subgradient descent using weighted averaging (SGDWA) in Theorem 5.2 (now Theorem 4.3):
> As indicated in the previous paragraph, for the utility analysis we use a non-smooth loss function (for our proof) which has a slower convergence rate with the number of iterations $M$ than the smooth counterpart. In particular, for a non-smooth loss function under SGD, we converge with rate $\mathcal{O}(\log(M)/M)$ (Shalev-Shwartz et al., 2011) but for the smooth Huber loss under projected SGD (the one from our experiments), we converge with rate $\mathcal{O}(1/M)$ (Juditsky et al., 2009).
> Thus, we used for our utility analysis the SGD variation SGDWA to also reach a convergence rate of $\mathcal{O}(1/M)$ (Lacost-Julien et al., 2012).
>
> Projected gradient descent is defined in Algorithm 1 (last line) and Algorithm 3 (last line). After each SGD update, it projects all parameters outside an $R$-sized sphere, to this sphere.
>
> [Juditsky et al., 2009] "Stochastic Approximation approach to Stochastic Programming" by A. Juditsky, G. Lan, A. Nemirovski, A. Shapiro. In: SIAM Journal on Optimization, 2009.
>
> [Shalev-Shwartz et al., 2011] "Pegasos: primal estimated sub-gradient solver for SVM" by S. Shalev-Shwartz, Y. Singer, N. Srebro, A. Cotter. In: Math. Program, 2011.
>
>
> > 4. I'm confused by what it means to take the average of the models, even in Definition 3.1: I don't understand the "type" of $T$, so it's not clear why $\sum_i T(D^{(i)})$ is well-defined. In the proof of Corollary K.2 it looks like we sum models in the dual space, but Algs 2 and 3 look to me like we're working in the weight space. I think this distinction matters when we talk about adding noise.
>
> We took your feedback to refine our paper. In particular, we now use more clearly the model $f$ as the output of the learner $T : \text{Data space} \mapsto \text{Parameter space}$, defined as $f := T(D)$. With $\operatorname{avg}(T)$ we abbreviate the average of locally trained models, i.e., the average of the model parameters.
>
> In our paper, we locally train with $T$ (e.g. $T=$SVM-SGD or $T=$SoftmaxReg) and then noise and average the model parameters which we call blind model averaging (BlindAvg). Since for SVMs and SoftmaxReg, the representer theorem holds (cf. Appendix D.4), we can always represent the model parameters $f$ in the dual space: $f = \sum_{x_i \in D} \alpha_i x_i$. Whether we average the parameters in dual space or primal space, we get the same result. This is also formulated in Corollary 4.1 (moved from the Appendix into the main body).
>
>
> > 5. The theorem relies on Lemma 5.1, which (in its proof) assumes that the datasets are disjoint, i.e., no two data points are repeated across users. I don't see why this is a justified assumption, as differential privacy needs to hold for worst-case datasets.
>
> We agree that disjoint distributed datasets are an interesting problem for differentially private federated learning. In our work, we use the definition of federated disjoint datasets to represent local aggregators which is common in the literature on privacy-preserving federated learning.
>
> As our neighboring relation for DP, we consider that the datasets differ in one element in total, i.e., one element in one of the local datasets (similar to the case where the dataset is global).
> There are other neighboring relations like in group DP which differ in more data points in one local dataset or one data point in several local datasets. Even here, the group size is a necessary parameter that directly influences the privacy guarantees (and is never assumed worst-case).
>
> > 6. Two sections in the appendix are titled "Proof of Thm 5.2", J.5.2 and K.5.2, only the latter actually contains a proof.
>
> We agree that this raises some confusion. We replaced our summary of all statements with proofs (Appendix J) with a proper table of contents in Appendix K (now Appendix J).

---

> > ### Comment · Reviewer_gYEj · 2024-11-22
> >
> > I have read the other reviews and your response. It seems that you have made major revisions to the submission. I retain my opinion of the paper.

---

### Author Response · Authors · 2024-11-22
**Summary of Rebuttal**

We thank all reviewers for their insightful comments and answer each reviewer individually.
We updated our paper to address many of your comments on the presentation and to clarify our argument.

In particular, we
- reworked Section 4 (previously 5) such that it emphasizes the insights for non-HingeSVMs in Corollary 4.1 (previously this Corollary was in the Appendix),
- refactored our Configuration (previously Def. 3.1) into a table and also added other important notation,
- clarified the SGD variant in Algorithms 1 and 3,
- updated Figure 1 to clarify better which Section belongs to which part and what learners (e.g. SVM, SoftmaxReg) are covered in each part,
- added the error of BlindAvg with noise in Section 4 (previously Section 5),
- clarified the hypothesis of Theorem 2.1 (previously Theorem 3.2),
- better organized our proofs in the Appendix incl. a table of contents,
- improved the limitation section and added a discussion about future work,
- clarified user-level privacy, local DP, and group DP throughout our paper incl. the caption of Figure 6, and
- clarified our threat model s.t. it now clearly defines honest users.

---

> ### Author Response · Authors · 2024-11-27
> **Re: Summary of Rebuttal**
>
> For transparency, we only made (minor) presentational changes and clarifications/bug fixes to our statements and did not change anything substantial in the paper. Thus, we consider our changes a minor revision.
>
> In particular, section number changes were solely due to a reordering, and almost all changes were based on the valuable feedback of the reviewers.

---

### Meta-Review · Area_Chair_3mDE · 2024-12-20

**Metareview:**

This paper analyzes a "blind model averaging" for distributed and differentially-private model training. Each user holds a dataset, trains a model, and submits their model for averaging. The authors analyze privacy and utility guarantees for Support Vector Machines and Softmax Regression, together with empirical results.

The reviewers recognized the topic's relevance, appreciated the privacy analysis, and acknowledged the value in addressing the costs associated with privacy-ensuring federated learning. However, they also noted several weaknesses. Reviewers noted that the paper lacks rigor and needs a significantly improved presentation (gYEj and n1Ay) -- including "fatal" presentation problems involving the theoretical development (reviewer bBFP) -- and the results have limited applicability (n1Ay and fS8P). I side with these critiques and agree that the paper needs significant improvements prior to acceptance.

**Additional Comments On Reviewer Discussion:**

The authors significantly revised their manuscript after the first round of reviews were released. Despite the changes in the paper, the reviewers stood by their initial assessment.

---

### Decision · Program_Chairs · 2025-01-22

Reject